# Learning to Reduce Search Space for Generalizable Neural Routing Solver

## Abstract

Constructive neural combinatorial optimization (NCO) has attracted growing research attention due to its ability to solve complex routing problems without relying on handcrafted rules. However, existing NCO methods face significant challenges in generalizing to large-scale problems due to high computational complexity and inefficient capture of structural patterns. To address this issue, we propose a novel learning-based search space reduction method that adaptively selects a small set of promising candidate nodes at each step of the constructive NCO process. Unlike traditional methods that rely on fixed heuristics, our selection model dynamically prioritizes nodes based on learned patterns, significantly reducing the search space while maintaining solution quality. Experimental results demonstrate that our method, trained solely on 100-node instances from a uniform distribution, generalizes remarkably well to large-scale Traveling Salesman Problem (TSP), Capacitated Vehicle Routing Problem (CVRP), and Capacitated Vehicle Routing Problem with Time Windows (CVRPTW) instances with up to 1 million nodes from the uniform distribution and over 80K nodes from other distributions.

## 1 Introduction

The Vehicle Routing Problem (VRP) is one of the core problems in Operations Research with significant practical implications in domains such as logistics, supply chain management, and express delivery (Tiwari & Sharma, 2023; Sar & Ghadimi, 2023). Efficient routing optimization is crucial for enhancing delivery performance and reducing operational costs. Traditional heuristic algorithms, such as LKH3 (Helsgaun, 2017) and HGS (Vidal, 2022), have demonstrated strong capabilities in solving VRPs with diverse constraints. However, these methods face two fundamental limitations: (1) their design requires extensive domain expertise to craft problem-specific rules, and (2) their computational complexity scales poorly with instance size due to the NP-hard nature of VRPs. These challenges are particularly acute for large-scale instances (e.g., more than 10,000 nodes), where existing algorithms often fail to provide practical solutions with a reasonable runtime.

In recent years, neural combinatorial optimization (NCO) has emerged as a promising paradigm for solving complex problems like VRPs, which eliminates the need for time-consuming and handcrafted algorithm design by experts (Bengio et al., 2021; Li et al., 2022; Wu et al., 2024). These methods automatically learn problem-specific patterns through training frameworks such as supervised learning (SL) (Vinyals et al., 2015; Luo et al., 2023; Drakulic et al., 2023; Xiao et al., 2024b; Joshi et al., 2019; Hudson et al., 2022) or reinforcement learning (RL) (Bello et al., 2016; Khalil et al., 2017; Kool et al., 2019; Zhou et al., 2024a). A well-trained NCO model can directly construct approximate solutions without explicit search, offering a promising direction for real-time VRP solving. However, SL-based methods face a critical limitation due to the difficulty of obtaining high-quality labeled data (e.g., near-optimal solutions) for large-scale NP-hard problems. In contrast, RL-based methods do not require labeled data and have demonstrated strong performance on small-scale instances (e.g., 100 nodes) (Kwon et al., 2020; Kim et al., 2022; Xiao et al., 2025). Nevertheless, their effectiveness diminishes significantly on large-scale instances (e.g., 10,000 nodes), primarily due to the exponentially growing search space and the challenge of sparse reward.

To address the scalability challenges, search space reduction (SSR) has gained increasing attention as a scalable strategy. As listed in Table 1, existing SSR techniques can be broadly categorized into two

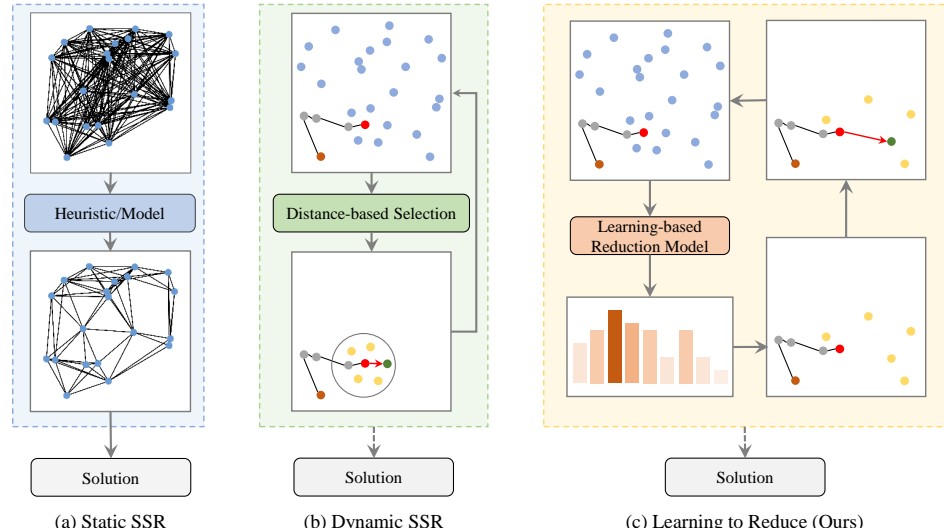

(a) Static SSR      (b) Dynamic SSR      (c) Learning to Reduce (Ours)

Figure 1: **Different Search Space Reduction (SSR) Methods:** (a) **Static Search Space Reduction** prunes the whole search space only once at the beginning of optimization process; (b) **Dynamic Search Space Reduction** reduces the search space to a small set of candidate nodes based on the distance to the last visited node at each construction step; (c) **Learning to Reduce (Ours)** builds reinforcement learning based model to adaptively reduce the search space and then select the next node for solution construction. We provide detailed descriptions for each reduction in Appendix C.

types: static and dynamic. Static SSR performs a one-time pruning at the beginning of the optimization process, offering computational efficiency. However, it often requires additional search procedures (e.g., Monte Carlo Tree Search for TSP) to achieve high-quality solutions (Fu et al., 2021; Qiu et al., 2022; Sun & Yang, 2023). In contrast, dynamic SSR (Fang et al., 2024; Gao et al., 2024; Wang et al., 2024) adaptively adjusts the candidate node set at each construction step based on real-time problem states, enabling more effective search space reduction for constructive NCO methods. Despite their advantages, existing dynamic SSR methods are fundamentally constrained by their reliance on distance-based node selection, which struggles to generalize to large-scale instances, particularly those with non-uniform node distributions.

Table 1: Comparison of L2R and classical neural routing solvers with search space reduction.

| Neural Routing Solver | Static SSR | Dynamic SSR | Training Scale | Generalizable Scale |
|---|---|---|---|---|
| MLPR (Sun et al., 2021) | ✓ | × | 100 | 2K |
| Att-GCN+MCTS (Fu et al., 2021) | ✓ | × | 50 | 10K |
| DIMES (Qiu et al., 2022) | ✓ | × | 10K | 10K |
| DIFUSCO (Sun & Yang, 2023) | ✓ | × | 10K | 10K |
| T2T (Li et al., 2023) | ✓ | × | 1K | 1K |
| BQ (Drakulic et al., 2023)‡ | × | Distance-based | 100 | 1K |
| ELG (Gao et al., 2024) | × | Distance-based | 100 | 7K |
| DAR (Wang et al., 2024) | × | Distance-based | 500 | 11K |
| INViT (Fang et al., 2024) | × | Distance-based | 100 | 10K |
| L2R (Ours) | ✓ | Learning-based | 100 | **10M** |

‡ BQ (Drakulic et al., 2023) limits the sub-graph to the 250 nearest neighbors of the current node when facing large-scale instances.

As shown in Figure 1, unlike existing SSR approaches, this work proposes a novel *Learning to Reduce (L2R)* framework that adaptively combines distance-based priors with context-dependent features learned by the network. Our contributions can be summarized as follows:

- We comprehensively analyze the key limitations of existing distance-based SSR methods and then propose a novel learning-based framework with hierarchical static and dynamic SSR for solving large-scale VRPs.

- In particular, we develop an RL-based approach to reduce the search space and select candidate nodes at each construction step, significantly reducing computational overhead without compromising solution quality.

- We conduct comprehensive experiments to demonstrate that our proposed method, trained only on 100-node instances from a uniform distribution, can generalize remarkably well to instances with up to 1 million nodes from a uniform distribution and over 80k nodes from other distributions.

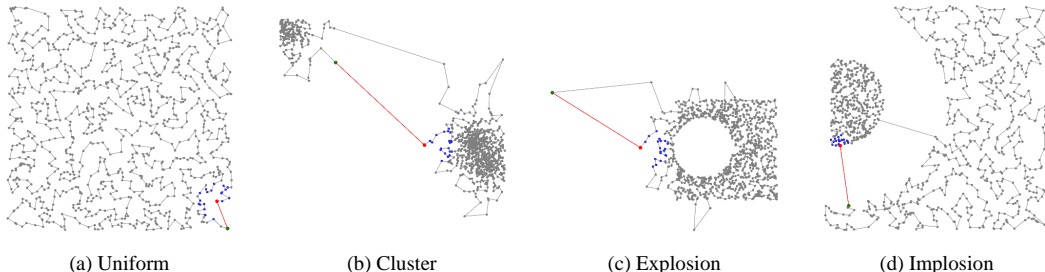

|     |     |     |     |
| --- | --- | --- | --- |
| (a) Uniform | (b) Cluster | (c) Explosion | (d) Implosion |

Figure 2: **Impacts of distance-based search space reduction on solution optimality.** (a)-(d) Near-optimal solutions for TSP1K instances under four distribution patterns: uniform, cluster, explosion, and implosion. Restricting the search space to the k-nearest neighbors ($k = 20$) will lead to suboptimal routes. The red, green, and blue nodes indicate the current node, the next visiting node, and $k$ nearest neighbors, respectively.

## 2 RELATED WORK

### 2.1 NCO WITHOUT SEARCH SPACE REDUCTION

Most NCO models are trained on small-scale instances (e.g., 100 nodes) without search space reduction and achieve strong performance on instances of similar size. However, their effectiveness significantly diminishes when applied to larger instances (e.g., those exceeding 1,000 nodes) (Nazari et al., 2018; Xiao et al., 2024a; Xin et al., 2021; Hottung & Tierney, 2020; Chen & Tian, 2019). Some approaches incorporate additional search procedures, such as 2-opt (Deudon et al., 2018) and active search (Bello et al., 2016; Hottung et al., 2022), to address this limitation. While improving solution quality, they are still computationally expensive for large-scale instances. Another line of research focuses on training NCO models directly on larger-scale instances (e.g., up to 500 nodes) to enhance generalization (Wang et al., 2024; Zhou et al., 2023; 2024a). However, this approach incurs prohibitive computational costs due to the exponentially growing search space. Alternatively, some methods simplify large-scale VRPs through decomposition policies (Kim et al., 2021; Li et al., 2021; Hou et al., 2022; Pan et al., 2023; Zheng et al., 2024). Although effective, they often overlook the dependency between decomposition and repair policies, resulting in suboptimal solutions. Moreover, the reliance on expert-designed policies limits their practicality for real-world applications.

### 2.2 NCO WITH STATIC SEARCH SPACE REDUCTION

To address the scalability challenges, static SSR has been proposed as a computationally efficient approach. These methods perform one-time pruning at the beginning of the optimization process, significantly reducing the problem size. For example, Sun et al. (2021) develop a static problem reduction technique to eliminate unpromising edges in large-scale TSP instances. Recently, heatmap-based approaches have gained popularity for solving large-scale TSPs, where models are trained to predict the probability of each edge belonging to the optimal solution. To handle large-scale instances, these methods often incorporate graph sparsification (Qiu et al., 2022; Sun & Yang, 2023; Li et al., 2023; 2024) or pruning strategies (Xing & Tu, 2020; Fu et al., 2021; Min et al., 2023) to reduce the search space. While static SSR is computationally efficient, it typically requires additional well-designed search procedures (e.g., Monte Carlo Tree Search for TSP) to achieve high-quality solutions, which might be more important for the optimization process (Xia et al., 2024).

### 2.3 NCO WITH DYNAMIC SEARCH SPACE REDUCTION

Dynamic SSR has also been explored as a promising approach to address the scalability challenges of NCO methods. These methods adaptively prune the search space to a small set of candidate nodes at each construction step, typically based on the distance to the last visited node. The final node selection can be guided by either the original policy augmented with auxiliary distance information (Wang et al., 2024) or a well-designed local policy (Gao et al., 2024). Additionally, recent work by Fang et al. (2024) and Drakulic et al. (2023) directly selects the next node from the candidate set using NCO

models. While dynamic SSR can more efficiently reduce the search space for constructive NCO, its reliance on distance-based node selection is inappropriate for large-scale instances, especially those from non-uniform distributions.

## 3 Shortcomings of Distance-based Search Space Reduction

The fundamental limitation of distance-based search space reduction lies in its tendency to prune globally optimal nodes during solution generation. As demonstrated in Figure 2, restricting candidate nodes to the $k$-nearest neighbors forces the algorithm to ignore critical long-range connections necessary for optimal routing. This over-pruning effect accumulates systematically throughout the solving process, ultimately compromising solution quality. In this section, we systematically analyze these shortcomings through the impact on the optimality gap of the classic solver and the impact on constructive NCO methods across diverse architectures.

**Impact on Optimality Gap** To evaluate the impact of distance-based search space reduction on final optimization performance, we adopt the widely-used LKH-3 algorithm (Helsgaun, 2017) as our benchmark solver and conduct experiments on TSPLIB (Reinelt, 1991). Given that LKH-3 is not a constructive heuristic, we restrict the search space for each instance by pruning the original fully connected graph into a sparse topology. Specifically, only connections to the $k$ nearest nodes are retained for each node in our experiments.

The results in Table 2 illustrate the performance across TSPLIB instances of varying scales under different levels of search space

Table 2: Optimality gap comparison of LKH-3 on unreduced and reduced TSPLIB instances.

| Instance | Scale | w/o D-SSR | w/ $k = 10$ | w/ $k = 20$ |
|---|---|---|---|---|
| dsj1000 | 1,000 | 0.00% | 7.27% | 6.85% |
| pr1002 | 1,002 | 0.00% | 3.14% | 1.06% |
| d1291 | 1,291 | 0.00% | 26.71% | 9.59% |
| fl1400 | 1,400 | 0.02% | 64.72% | 54.29% |
| fl1577 | 1,577 | 0.01% | 48.14% | 38.19% |
| d1655 | 1,655 | 0.00% | 20.41% | 8.74% |
| rl1889 | 1,889 | 0.00% | 27.94% | 8.45% |
| d2103 | 2,103 | 0.01% | 17.62% | 6.10% |
| fl3795 | 3795 | 0.06% | 56.87% | 53.38% |
| rl5915 | 5,915 | 0.03% | 17.99% | 2.58% |
| rl5934 | 5,934 | 0.03% | 27.24% | 6.40% |
| rl11849 | 11,849 | 0.00% | 11.01% | 1.90% |
| usa13509 | 13,509 | 0.01% | 20.06% | 5.87% |
| Avg.gap | | 0.01% | 26.86% | 15.64% |

reduction ($k$). A key observation is that while LKH-3 achieves near-optimal solutions in an unreduced search space, aggressive pruning (i.e., using small $k$ values) substantially degrades performance. This degradation is particularly pronounced in instances where optimal routes inherently depend on non-local node selections, which may be eliminated by such aggressive pruning strategies. We provide detailed solution visualizations comparing three different results in Appendix D.1.

**Impact on Constructive NCOs** For constructive NCO methods, each construction step should consider all feasible nodes to guarantee optimality. However, the search space of VRPs grows exponentially with problem size. For large-scale instances, directly obtaining high-quality solutions remains extremely challenging in an unreduced search space. While distance-based SSR methods (Drakulic et al., 2023; Fang et al., 2024) improve computational efficiency, they also introduce critical limitations in their failure to include a small but essential subset of nodes within their candidate set. To evaluate the impact of the search space reduction in NCO performance, we use the well-known LEHD (Luo et al., 2023) and POMO (Kwon et al., 2020) as representative examples, restricting the search space to the $k$ nearest nodes from the last visited node at each construction step. A key observation (see Appendix D.2 for detailed results) is the existence of a critical threshold $k^*$ for each problem size, beyond which LEHD and POMO with $k \geq k^*$ achieve a 0% performance gap with the original models, indicating no loss in solution quality with SSR. However, these critical thresholds vary significantly across problem sizes, and using a small $k < k^*$ leads to substantial performance degradation. We also find that while reducing the search space significantly decreases inference time, employing an excessively small $k$ based on pairwise distance information results in substantial performance degradation. These findings highlight the potential of search space reduction while underscoring the need for advanced reduction methods that can efficiently handle large-scale problems with a small $k$.

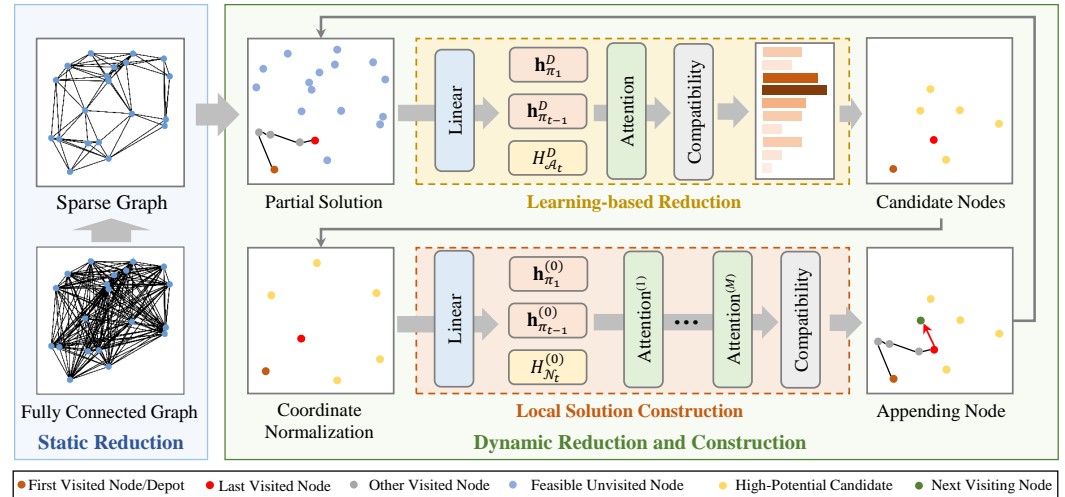

Figure 3: The pipeline of our proposed L2R framework for solving large-scale VRP instances.

# 4 LEARNING TO REDUCE (L2R)

In this section, we propose *Learning to Reduce (L2R)*, a hierarchical neural framework to address the scalability limitations of search space reduction in vehicle routing problems. As illustrated in Figure 3, our framework introduces three complementary stages: 1) static reduction, 2) learning-based reduction, and 3) local solution construction.

## 4.1 STATIC REDUCTION

The static reduction stage initiates our hierarchical framework by pruning the original fully connected graph $G = (\Omega, E)$ into a sparse topology $G' = (\Omega, E')$. For each node $i \in \Omega$, we compute pairwise Euclidean distances $\{d_{ij}\}_{j=1}^N$ and eliminate connections to nodes in the farthest $\gamma$-percentile. Formally, the sparse edge set is defined as

$$E' = \bigcup_{i \in \Omega} \{e_{ij} \mid \mathrm{rank}(d_{ij}) \leq (1 - \gamma)|\Omega|\}, \quad \gamma \in [0, 1] \tag{1}$$

where $\mathrm{rank}(d_{ij})$ denotes the ascending order of distances from $v_i$ (i.e., $\mathrm{rank}(d_{ij}) = 1$ for the closest neighbor). Through empirical analysis across diverse scales and node distributions, we use $\gamma = 10\%$ as the threshold in this work, which efficiently reduces computational overhead without compromising solution optimality (see Appendix G for further analysis). Crucially, this stage operates as a one-time preprocessing step, incurring no runtime overhead during subsequent solving phases.

## 4.2 LEARNING-BASED REDUCTION

The static reduction stage conducts partial search space pruning. However, there is still a large number of non-optimal edges remaining in the pruned graph $G'$, which imposes prohibitive computational costs for NCO in large-scale scenarios. However, as discussed in the previous section, further distance-based search reduction with a large reduced rate will lead to a poor optimality ratio and poor performance.

To mitigate the negative impact of reduction on optimality, we develop a learning-based model to dynamically evaluate the potential of feasible nodes and adaptively reduce the search space at each construction step. Our proposed model ensures efficiency through a lightweight structure containing only an embedding layer and an attention layer. Their implementations are detailed in the following.

**Embedding Layer** Given an instance $S = \{\mathbf{s}_i\}_{i=1}^n$ with $n$ node features $\mathbf{s}_i \in \mathbb{R}^{d_x}$ (e.g., city coordinates in TSP), we first project these features into $d$-dimensional embeddings through a shared linear transformation for each node:

$$\mathbf{h}_i^D = W^{(e)}\mathbf{s}_i + \mathbf{b}^{(e)}, \quad \forall i \in \{1, \ldots, N\} \tag{2}$$

where $W^{(e)} \in \mathbb{R}^{d_x \times d}$ and $\mathbf{b}^{(e)} \in \mathbb{R}^d$ are learnable parameters. In other words, we obtain a set of embeddings $H^D = \{\mathbf{h}_i^D\}_{i=1}^n \in \mathbb{R}^{n \times d}$ for all nodes in the instance $S$.

At $t$-th step with partial solution $(\pi_1, \dots, \pi_{t-1})$, we adopt the setting from Kool et al. (2019) to represent the current partial solution using the initial node embedding $\mathbf{h}_{\pi_1}^D$ and latest node embedding $\mathbf{h}_{\pi_{t-1}}^D$. In addition, following Kwon et al. (2020), we define a context embedding of the current partial solution:

$$\mathbf{h}_{C_D}^t = W_{\text{first}} \mathbf{h}_{\pi_1}^D + W_{\text{last}} \mathbf{h}_{\pi_{t-1}}^D, \tag{3}$$

where $W_{\text{first}} \in \mathbb{R}^{d \times d}$ and $W_{\text{last}} \in \mathbb{R}^{d \times d}$ are two learnable matrices. In addition, we denote $\mathcal{A}_t$ as the set of all feasible nodes (e.g., unvisited cities with a valid edge in $E'$ connected to the current city) at the $t$-th step. The embeddings of $\mathcal{A}_t$ are denoted by $H_{\mathcal{A}_t}^D = \{\mathbf{h}_i^D | i \in \mathcal{A}_t\} \in \mathbb{R}^{|\mathcal{A}_t| \times d}$.

**Attention Layer** To calculate potential scores for all feasible nodes in $\mathcal{A}_t$, we process the context embedding $\mathbf{h}_{C_D}^t$ and node embeddings $H_{\mathcal{A}_t}^D$ through an attention mechanism. First, we project the node embeddings into key-value pairs:

$$K_{\mathcal{A}_t}^D = W^K H_{\mathcal{A}_t}^D, \quad V_{\mathcal{A}_t}^D = W^V H_{\mathcal{A}_t}^D \tag{4}$$

where $W^K, W^V \in \mathbb{R}^{d \times d}$ are learnable projection matrices. The context embedding $\mathbf{h}_{C_D}^t$ then interacts with these projections through the attention operator:

$$\hat{\mathbf{h}}_{C_D}^t = \text{Attention}\left(\mathbf{h}_{C_D}^t, K_{\mathcal{A}_t}^D, V_{\mathcal{A}_t}^D\right) \tag{5}$$

**Compatibility Calculation** Finally, similar to previous work (Kool et al., 2019; Kwon et al., 2020; Zhou et al., 2024a), we compute the compatibilities $\mathbf{u}^R = \{u_{t,i}^R | i \in \mathcal{A}_t\}$:

$$u_{t,i}^R = \begin{cases} \xi \cdot \tanh\left(\dfrac{\hat{\mathbf{h}}_{C_D}^t (\mathbf{h}_i^D)^{\mathsf{T}}}{\sqrt{d_k}} + a_{t-1,i}^R\right) & \text{if } i \in \mathcal{A}_t \\ -\infty & \text{otherwise} \end{cases}, \tag{6}$$

where $\xi$ is the clipping parameter, $d_k$ is the dimension for matrix $K_{\mathcal{A}_t}^D$, and $a_{t-1,i}^R$ is the adaptation bias between each node $i \in \mathcal{A}_t$ and the current node $\pi_{t-1}$, following Zhou et al. (2024a) (see Appendix E for more details). Finally, the potential scores $\mathbf{o} = \{o_i | i \in \mathcal{A}_t\}$ are computed as $\mathbf{o} = \text{softmax}(\mathbf{u}^R)$. The score $o_i$ quantifies the potential of node $i \in \mathcal{A}_t$. Then, we retain the top-$k$ nodes with the highest potential scores as the candidate nodes and denote the set of all candidate nodes as $\mathcal{N}_t$. In this way, we conduct a learning-based dynamic search space reduction at each construction step. More details about search space reduction can be found in Appendix H.

### 4.3 Local Solution Construction

In this subsection, we develop a learning-based local solution construction model to select one final node from the candidate set $\mathcal{N}_t$ to construct the partial solution at construction step $t$. Similar to previous work (Kool et al., 2019; Kwon et al., 2020; Drakulic et al., 2023; Luo et al., 2023), we also use the initial node $\pi_1$ and the latest node $\pi_{t-1}$ to represent the current partial solution. First of all, the embedding of each node can be obtained by a shared linear transformation $\mathbf{h}_i^{(0)} = W^{(0)} \mathbf{s}_i + \mathbf{b}^{(0)}, \quad \forall i \in \{\pi_1, \pi_{t-1}\} \cup \mathcal{N}_t$ where $W^{(0)}$ and $b^{(0)}$ are learnable parameters. Then we obtain the embedding of the partial graph:

$$\widetilde{H}^{(0)} = [W_1 \mathbf{h}_{\pi_1}^{(0)}, W_2 \mathbf{h}_{\pi_{t-1}}^{(0)}, H_{\mathcal{N}_t}^{(0)}] \in \mathbb{R}^{(2+|\mathcal{N}_t|) \times d} \tag{7}$$

where $[\cdot, \cdot]$ denotes the vertical concatenation operator, $H_{\mathcal{N}_t}^{(0)} = \{\mathbf{h}_i^{(0)} | i \in \mathcal{N}_t\} \in \mathbb{R}^{|\mathcal{N}_t| \times d}$ are the embeddings for all candidate nodes in $\mathcal{N}_t$, $W_1 \in \mathbb{R}^{d \times d}$ and $W_2 \in \mathbb{R}^{d \times d}$ are two learnable matrices. This embedding is the initial input to a sequence of attention layers.

In our local solution construction model, one attention layer consists of two sub-layers: an attention sub-layer and a Feed-Forward (FF) sub-layer, both of which use Layer Normalization (Ba et al., 2016) and skip-connection (He et al., 2016). The detailed process can be found in Appendix I.3. After $M$ attention layers, the final embedding $\widetilde{H}^{(M)} = [\mathbf{h}_{\pi_1}^{(M)}, \mathbf{h}_{\pi_{t-1}}^{(M)}, H_{\mathcal{N}_t}^{(M)}] \in \mathbb{R}^{(2+|\mathcal{N}_t|) \times d}$ contains advanced feature representations of the initial, last, and $k$ candidate nodes.

Finally, we compute the probabilities for selecting each node in $\mathcal{N}_t$ via another compatibility module different from Equation (6):

$$
u_{t,i}^L = \begin{cases} \xi \cdot \tanh \left( \dfrac{\hat{\mathbf{h}}_{(C)}^t \left( \mathbf{h}_i^{(M)} \right)^{\mathrm{T}}}{\sqrt{d_k}} + a_{t-1,i}^L \right) & \text{if } i \in \mathcal{N}_t \\ -\infty & \text{otherwise} \end{cases}, \tag{8}
$$

where $\hat{\mathbf{h}}_{(C)}^t = \mathbf{h}_{\pi_1}^{(M)} + \mathbf{h}_{\pi_{t-1}}^{(M)}$ and $a_{t-1,i}^L$ is the adaptation bias between each node $i \in \mathcal{N}_t$ and $\pi_{t-1}$. The probabilities $\mathbf{p} = \{p_i | i \in \mathcal{N}_t\}$ of selecting the candidate node $i$ can be calculated as $\mathbf{p} = \text{softmax}(\mathbf{u}^L)$. More details are provided in Appendix I.

### 4.4 TRAINING

Our proposed L2R framework has a learning-based reduction model and a local solution construction model, which can be trained by a joint training scheme. We denote the learnable parameters as $\boldsymbol{\theta}_R$ and $\boldsymbol{\theta}_L$ for the reduction model and local solution construction model, respectively. Hence $\boldsymbol{\theta} = \{\boldsymbol{\theta}_R, \boldsymbol{\theta}_L\}$ contains all learnable parameters in the L2R framework.

L2R samples the node $\pi_t$ by probabilities $\mathbf{p} = \{p_i | i \in \mathcal{N}_t\}$ and add it to current partial solution, a complete solution $\pi = (\pi_1, \dots, \pi_n)$ for an instance $S$ is constructed by $\boldsymbol{\theta}_L$ with $n$ decoding steps. Since the reduction model is only responsible for selecting $k$ candidate nodes and does not participate in the actual node selection process, we use the total reward $\mathcal{R}(\pi \mid S, \boldsymbol{\theta}_L)$ (e.g., the negative value of tour length) of instance $S$ given a specific solution $\pi$ generated by $\boldsymbol{\theta}_L$ for the training of both $\boldsymbol{\theta}_R$ and $\boldsymbol{\theta}_L$ simultaneously, and we define $\tau = (\tau_1, \dots, \tau_n)$ as the set of sampled nodes by $\boldsymbol{\theta}_R$. Following previous NCO works (Kool et al., 2019; Fang et al., 2024; Xin et al., 2020), L2R is trained by the REINFORCE (Williams, 1992) gradient estimator:

$$
\mathcal{L}_{\text{Joint}}(\boldsymbol{\theta}) = \mathcal{L}_R(\boldsymbol{\theta}_R) + \mathcal{L}_L(\boldsymbol{\theta}_L), \tag{9}
$$

$$
\nabla_{\boldsymbol{\theta}_R} \mathcal{L}_R(\boldsymbol{\theta}_R) = \mathbb{E}_{o(\tau|S,\boldsymbol{\theta}_R)} \left[ (\mathcal{R}(\pi|S,\boldsymbol{\theta}_L) - b(S)) \nabla_{\boldsymbol{\theta}_R} \log o(\tau|S) \right], \tag{10}
$$

$$
\nabla_{\boldsymbol{\theta}_L} \mathcal{L}_L(\boldsymbol{\theta}_L) = \mathbb{E}_{p(\pi|S,\boldsymbol{\theta}_L)} \left[ (\mathcal{R}(\pi|S,\boldsymbol{\theta}_L) - b(S)) \nabla_{\boldsymbol{\theta}_L} \log p(\pi|S) \right], \tag{11}
$$

where $o(\tau \mid S) = \prod_{t=2}^n o(\tau_t \mid S, \pi_{1:t-1})$, $p(\pi \mid S) = \prod_{t=2}^n p(\pi_t \mid S, \pi_{1:t-1})$, $b(S)$ is the greedy rollout baseline. The detailed training process is provided in Appendix J.

## 5 EXPERIMENTS

In this section, we comprehensively evaluate our proposed model against classical and learning-based solvers on synthetic and real-world TSP, CVRP and CVRPTW instances[1]. Notably, for each problem type (TSP, CVRP, and CVRPTW), the corresponding model is trained solely on 100-node instances from the uniform distribution. We assess its generalization performance on: (1) scalability to problem sizes up to 1 million nodes and (2) robustness to varying node distributions.

### 5.1 EXPERIMENTAL SETUP

**Problem Setting** For all problems, we generate synthetic instances following the methodology outlined in Kool et al. (2019). Specifically, we construct TSP and CVRP test datasets with uniformly distributed nodes at six scales: 1K, 5K, 10K, 50K, 100K, and 1M. Following Fu et al. (2021), the TSP1K test set consists of 128 instances, while the larger TSP datasets each contain 16 instances. For the CVRP(TW), we adhere to the capacity constraints specified in Hou et al. (2022). For CVRP, each dataset includes 100 instances, except for CVRP10K and CVRP1M, which contain 16 instances. For CVRPTW, we generate instances following MVMoE (Zhou et al., 2024b) and limit the test scale to 10K since existing methods consistently run Out of Memory (OOM) on instances where $N \geq 5,000$. Each dataset contains 16 instances. The near-optimal solutions for the TSP and CVRP(TW) instances are obtained using LKH3 (Helsgaun, 2017) and HGS (Vidal, 2022), respectively. To evaluate cross-distribution generalization performance, we test on the TSP/CVRP5K instances obtained from INViT (Fang et al., 2024). Additionally, we evaluate L2R on three benchmark datasets, they are: (1)

---

[1]We provide detailed definitions and formulations of TSP, CVRP and CVRPTW in Appendix A.

symmetric EUC_2D instances from TSPLIB (Reinelt, 1991);(2) CVRPLIB Set-XXL (Arnold et al., 2019), and (3) 8th DIMACS Implementation Challenge (Johnson & McGeoch, 2002)[2].

**Model & Training Setting**  For all experiments, we use an embedding dimension of 128 and a feed-forward layer dimension of 512. To enhance geometric pattern recognition in VRPs, we integrate scale and distance information into the attention mechanism (see Appendix F for implementation details). The local construction model employs 6 attention layers. Consistent with Kool et al. (2019), we set the clipping parameter $\xi = 10$ in Equation (6) and Equation (8). The hyperparameter $k$ is configured as 20 for TSP, 50 for CVRP ,and 15 for CVRPTW. All experiments are conducted on a single NVIDIA GeForce RTX 3090 GPU (24GB memory).

Our model is exclusively trained on uniformly distributed instances with 100 nodes. We employ the Adam optimizer (Kingma & Ba, 2014) with an initial learning rate of $\eta = 10^{-4}$ and a learning rate decay of 0.98 per epoch. Training spans 100 epochs with 2,500 batches per epoch. Due to memory constraints, batch sizes differ across problems—180 for TSP, 60 for CVRP , and 128 for CVRPTW. The same pre-trained model is used in all experimental evaluations for each problem. Additional implementation details are provided in Appendix J (pseudocode for training) and Appendix K (training and model settings).

**Baseline**  We compare L2R with the following methods: (1) **Classical Solver**: Concorde (Applegate et al., 2006), LKH3 (Helsgaun, 2017), HGS (Vidal, 2022); (2) **Constructive NCO**: POMO (Kwon et al., 2020), Omni_VRP (Zhou et al., 2023), ELG (Gao et al., 2024), BQ (Drakulic et al., 2023), LEHD (Luo et al., 2023), INViT (Fang et al., 2024), SIL (Luo et al., 2024)[3], MTPOMO (Liu et al., 2024), MVMoE (Zhou et al., 2024b) and ReLD-MTL (Huang et al., 2025); (3) **Two-Stage NCO**: TAM (Hou et al., 2022) and GLOP (Ye et al., 2024).

**Metrics and Inference**  We report the average objective value (Obj.), optimality gap (Gap), and average inference time (Time) for each method. The optimality gap quantifies the discrepancy between the generated solutions and the near-optimal solutions. For most NCO baseline methods, we execute the source code provided by the authors using default settings. Results marked with an asterisk (*) are directly obtained from the corresponding papers. Some methods fail to produce feasible solutions within a reasonable time limit (e.g., several days), which is denoted by 'N/A'. The notation 'OOM' indicates that the memory consumption exceeds the available memory limits. For L2R, we report two types of results, which are obtained by greedy trajectory (greedy) and those derived from Parallel local ReConstruction (PRC) under different numbers of iterations (Luo et al., 2024). The parallel approach demonstrates promising results by effectively trading computing time for improved solution quality. For PRC, initial solutions are generated using the greedy trajectory. PRC100 refers to 100 iterations, with the longest destruction length per iteration set to 1,000 to balance speed and effectiveness. Further details about PRC are available in (Luo et al., 2024).

## 5.2 Performance Evaluation

**Cross-Size Generalization**  We conduct experiments on large-scale routing instances with uniform distribution, and the experimental results are reported in Table 3 (TSP/CVRP1K−100K), Table 4(TSP/CVRP1M), and Appendix L.1 (CVRPTW1K-10K). Benefiting from an efficient search space reduction scheme, our method consistently delivers superior inference performance across various problem instances. While it does not surpass SL-based LEHD and BQ on TSP/CVRP1K, RL-based L2R achieves shorter runtime compared to other methods, such as LEHD (e.g., 25 sec vs. 3.3 min on TSP1K instances), and L2R can outperform all comparable RL-based methods. For larger-scale instances, our proposed L2R, trained only on 100-node instances, significantly improves the scalability of NCO methods and outperforms other methods (including BQ and LEHD). Compared to classic heuristics, L2R achieves a 4.79% optimality gap and a 4× speed-up over LKH-3 on 16 TSP1M instances. Additionally, L2R successfully tackles CVRP1M instances that exceed the computational limits of both LKH3 and HGS. To the best of our knowledge, L2R is the first neural

---

[2]We also assess performance on small-scale instances ($N = 100$ and $N = 500$). Please see Appendix L.2 for detailed results and analysis.

[3]Due to the fundamental difference in training strategy (i.e., SIL trains separate models for each target scale), we will discuss the comparison results with SIL separately. For detailed results, please refer to Appendix L.3.

Table 3: Comparison of TSP and CVRP instances with uniform distribution.

| Method | TSP1K Obj. (Gap) | Time | TSP5K Obj. (Gap) | Time | TSP10K Obj. (Gap) | Time | TSP50K Obj. (Gap) | Time | TSP100K Obj. (Gap) | Time |
|---|---|---|---|---|---|---|---|---|---|---|
| LKH3 | 23.12 (0.00%) | 1.7m | 50.97 (0.00%) | 12m | 71.78 (0.00%) | 33m | 159.93 (0.00%) | 10h | 225.99 (0.00%) | 25h |
| Concorde | 23.12 (0.00%) | 1m | 50.95 (-0.05%) | 31m | 72.00 (0.15%) | 1.4h | N/A | N/A | N/A | N/A |
| GLOP (more revisions) | 23.78 (2.85%) | 10.2s | 53.15 (4.26%) | 1.0m | 75.04 (4.39%) | 1.9m | 168.09 (5.10%) | 1.5m | 237.61 (5.14%) | 3.9m |
| LEHD greedy | 23.84 (3.11%) | 0.8s | 58.85 (15.46%) | 1.5m | 91.33 (27.24%) | 11.7m | OOM | | OOM | |
| LEHD RRC1,000 | 23.29 (0.72%) | 3.3m | 54.43 (6.79%) | 8.6m | 80.90 (12.5%) | 18.6m | OOM | | OOM | |
| BQ greedy | 23.65 (2.30%) | 0.9s | 58.27 (14.31%) | 22.5s | 89.73 (25.02%) | 1.0m | OOM | | OOM | |
| BQ bs16 | 23.43 (1.37%) | 13s | 58.27 (10.7%) | 24s | OOM | | OOM | | OOM | |
| POMO aug×8 | 32.51 (40.6%) | 4.1s | 87.72 (72.1%) | 8.6m | OOM | | OOM | | OOM | |
| ELG aug×8 | 25.74 (11.33%) | 0.8s | 60.19 (18.08%) | 21s | OOM | | OOM | | OOM | |
| INViT-3V greedy | 24.67 (6.84%) | 0.4s | 54.46 (6.84%) | 12.7s | 76.87 (7.09%) | 34.9s | 171.42 (7.18%) | 4.9m | 242.26 (7.20%) | 18.8m |
| L2R greedy | 24.16 (4.49%) | 0.05s | 53.36 (4.69%) | 1.8s | 75.24 (4.82%) | 4.1s | 167.70 (4.86%) | 35.5s | 236.81 (4.79%) | 1.8m |
| L2R PRC100 | 23.62 (2.18%) | 2.5s | 52.41 (2.82%) | 20.1s | 73.95 (3.03%) | 26.3s | 165.16 (3.27%) | 1.8m | 234.36 (3.70%) | 3.2m |
| L2R PRC500 | 23.54 (1.82%) | 12.6s | 52.25 (2.50%) | 1.6m | 73.73 (2.72%) | 2.0m | 164.61 (2.92%) | 6.6m | 233.24 (3.21%) | 8.8m |
| L2R PRC1,000 | 23.52 (1.72%) | 24.9s | 52.20 (2.40%) | 3.1m | 73.66 (2.62%) | 3.8m | 164.41 (2.80%) | 12.4m | 232.77 (3.00%) | 15.5m |

| Method | CVRP1K Obj.(Gap) | Time | CVRP5K Obj.(Gap) | Time | CVRP10K Obj.(Gap) | Time | CVRP50K Obj.(Gap) | Time | CVRP100K Obj.(Gap) | Time |
|---|---|---|---|---|---|---|---|---|---|---|
| HGS | 41.2 (0.00%) | 5m | 126.2 (0.00%) | 5m | 227.3 (0.00) | 5m | 1081 (0.00%) | 4h | 2087.5 (0.00%) | 6h |
| GLOP-G (LKH-3) | 45.9 (11.4%) | 1.1s | 140.6(11.4%) | 4.0s | 256.4 (11.1%) | 6.2s | OOM | | OOM | |
| LEHD greedy | 44 (6.80%) | 0.8s | 138.2 (9.51%) | 1.4m | 256.3 (12.76%) | 12m | OOM | | OOM | |
| LEHD RRC1,000 | 42.4 (2.91%) | 3.4m | 132.7 (5.15%) | 10m | 243.8 (7.28%) | 51.6m | OOM | | OOM | |
| BQ greedy | 44.2 (7.28%) | 1s | 139.9 (10.86%) | 18.5s | 262.2 (15.35%) | 2m | OOM | | OOM | |
| BQ bs16 | 43.1 (4.61%) | 14s | 136.4 (8.08%) | 2.4m | OOM | | OOM | | OOM | |
| POMO aug × 8 | 101 (145.15%) | 4.6s | 632.9 (401.51%) | 11m | OOM | | OOM | | OOM | |
| ELG aug × 8 | 46.4 (12.62%) | 10.3s | OOM | | OOM | | OOM | | OOM | |
| INViT-3V greedy | 48.3 (17.23%) | 1s | 146.2 (15.85%) | 7s | 262.5 (15.54%) | 1.2m | 1331.1 (23.1%) | 5.6m | 2683.4 (28.55%) | 22m |
| L2R greedy | 45.8 (11.27%) | 0.1s | 136.0 (7.74%) | 0.5s | 236.7 (4.18%) | 4.4s | 1075.0 (-0.56%) | 37s | 2055.1 (-1.55%) | 2m |
| L2R PRC100 | 44.7 (8.53%) | 4.0s | 132.7 (5.14%) | 14.4s | 233.2 (2.62%) | 39.9s | 1072.6 (-0.78%) | 3.5m | 2051.5 (-1.73%) | 7m |
| L2R PRC500 | 44.4 (7.75%) | 20.0s | 131.2 (3.97%) | 1.2m | 230.9 (1.65%) | 3.0m | 1071.2 (-0.90%) | 17m | 2051.1 (-1.74%) | 33m |
| L2R PRC1,000 | 44.3 (7.42%) | 40.3s | 130.6 (3.52%) | 2.3m | 230.1 (1.26%) | 6.0m | 1070.5 (-0.98%) | 33m | 2050.9 (-1.75%) | 1.1h |

solver capable of effectively solving TSP and CVRP instances with one million nodes, and it is also the first constructive neural solver to surpass HGS on large-scale CVRP instances ($N \geq 50K$) when trained solely on 100-node uniform instances without any post-processing operations.

**Cross-Distribution Generalization** We evaluate the cross-distribution performance on TSP5K/CVRP5K instances from three distinct distributions: cluster, explosion, and implosion. As shown in Table 5, L2R consistently achieves the best performance among all comparable methods across these distributions, which further highlights the robust generalization capabilities of our L2R.

**Results on Benchmark Dataset** We further assess the generalization performance of L2R on CVRPLIB Set-XXL (Arnold et al., 2019) and TSPLIB (Reinelt, 1991). As demonstrated in Table 6, L2R maintains its position as the best-performing model across instances of varying scales, underscoring its practical applicability in real-world scenarios. Detailed results are provided in Appendix M.2. In addition, we also evaluate L2R on challenging large-scale TSP

Table 4: Comparison of TSP and CVRP instances with one million nodes.

| Method | TSP1M Obj. | Gap | Time | CVRP1M Obj. | Gap | Time |
|---|---|---|---|---|---|---|
| Concorde | N/A | N/A | N/A | – | | |
| HGS | | | | OOM | | |
| LKH3 | 713.97 | 0.00% | 4.8h | N/A | N/A | N/A |
| POMO | | OOM | | | OOM | |
| ELG | | OOM | | | OOM | |
| INViT-3V greedy | N/A | N/A | N/A | N/A | N/A | N/A |
| L2R greedy | 748.18 | 4.79% | 1.2h | 17231.85 | 0.00% | 1.4h |

instances from the DIMACS challenge ($10K \leq N \leq 10M$). As shown in Appendix M.3, L2R can consistently produce high-quality solutions for unprecedented instances with up to **10 million** nodes, which significantly outperforms other NCO methods. Notably, this capability is achieved through generalization from the small-scale 100-node training instances with uniform distributions.

## 5.3 ADDITIONAL ANALYSES

**L2R vs. Leading Heuristics** We evaluate L2R alongside current leading heuristics (i.e., LKH3 and HGS). As shown in Appendix N.1, each solver has distinct strengths, which represent complementary advantages rather than unilateral dominance. L2R is a strong contender that outperforms LKH3 on CVRP while demonstrating massive-scale solving capability that HGS cannot match.

**L2R vs. D-SSR** To validate the effectiveness of our proposed L2R, we conduct a comparative experiment against a dynamic distance-based SSR model, ensuring identical experimental settings. We systematically analyze the advantages of L2R, mainly including: (1) overall performance on real-world instances, (2) optimality gap and optimality ratio per instance, (3) impacts of different candidate node selection strategies, and (4) solution visualizations comparing the two approaches. As detailed in Appendix N.2, L2R consistently demonstrates better candidate selection on real-world problems

Table 5: Comparison on cross-distribution generalization.

| Method | TSP5K, Cluster | | TSP5K, Explosion | | TSP5K, Implosion | | CVRP5K, Cluster | | CVRP5K, Explosion | | CVRP5K, Implosion | |
|---|---|---|---|---|---|---|---|---|---|---|---|---|
| | Gap | Time | Gap | Time | Gap | Time | Gap | Time | Gap | Time | Gap | Time |
| Near-optimal | 0.00% | – | 0.00% | – | 0.00% | – | 0.00% | – | 0.00% | – | 0.00% | – |
| ELG multi-greedy‡ | 22.83% | 1.6m | 20.71% | 1.6m | 17.55% | 1.6m | 18.14% | 2.4m | 13.21% | 2.2m | 7.50% | 2.2m |
| Omni_VRP multi-greedy‡ | 54.53% | 1.1m | 51.09% | 1.1m | 50.20% | 1.1m | 22.05% | 1.3m | 33.09% | 1.4m | 40.20% | 1.3m |
| INViT-3V greedy | 8.20% | 11.3s | 11.48% | 11.3s | 8.52% | 11.3s | 9.05% | 29.4s | 8.44% | 29.3s | 8.77% | 23.2s |
| L2R greedy | 6.14% | 1.5s | 9.54% | 1.5s | 6.17% | 1.5s | 2.65% | 1.8s | 4.38% | 1.8s | 3.15% | 1.8s |
| L2R PRC1,000 | **3.16%** | 2.6m | **3.52%** | 2.6m | **2.87%** | 2.6m | **0.99%** | 4.2m | **1.14%** | 4.2m | **0.73%** | 4.2m |

† All datasets are obtained from INViT(Fang et al., 2024) and each contains 20 instances.

‡ The instance augmentation technique is not employed for comparable methods to prevent methods from exceeding memory limits.

Table 6: Comparison on large-scale TSPLIB (Reinelt, 1991) instances ($5,000 \leq N \leq 85,900$) and CVRPLIB (Arnold et al., 2019) instances ($3,000 \leq N \leq 30,000$).

| | TSPLIB | | | | CVRPLIB-XXL | | | |
|---|---|---|---|---|---|---|---|---|
| Method | $1K \leq N \leq 5K$ (23 instances) | $5K < N \leq 100K$ (10 instances) | All (33 instances) | Solved# | $3K \leq N \leq 7K$ (4 instances) | $7K < N \leq 30K$ (6 instances) | All (10 instances) | Solved# |
| GLOP | 6.16% | 7.69% | 6.62% | 33/33 | 17.07% | 21.32% | 19.62% | 10/10 |
| TAM-LKH3* | – | – | – | – | 20.44% | – | – | – |
| BQ bs16 | 10.65% | 30.58%† | – | 26/33 | 20.20% | OOM | – | 4/10 |
| LEHD greedy | 11.14% | 39.34%† | – | 30/33 | 22.22% | 32.80%† | – | 6/10 |
| LEHD RRC1000 | 4.00% | 18.46%† | – | 30/33 | 14.06% | 21.52%† | – | 6/10 |
| POMO aug×8 | 62.81% | OOM | – | 23/33 | 331.24%† | OOM | – | 2/10 |
| ELG aug×8 | 11.34% | OOM | – | 23/33 | 16.82% † | OOM | – | 2/10 |
| INViT-3V greedy | 11.49% | 10.03% | 11.05% | 33/33 | 20.74% | 26.64% | 24.28% | 10/10 |
| L2R greedy | 9.16% | 7.36% | 8.61% | 33/33 | 12.05% | 11.22% | 11.55% | 10/10 |
| L2R PRC100 | **3.60%** | **4.58%** | **3.89%** | 33/33 | **8.62%** | **9.34%** | **9.05%** | 10/10 |

† Some instances are skipped due to the OOM issue.

and significantly outperforms the dynamic D-SSR baseline. These findings further underscore the superior practical applicability of L2R in real-world scenarios.

**Complexity Analysis** We present the time and space complexity analyses for three different constructive models: L2R with learning-based SSR, INViT with distance-based SSR, and LEHD without SSR. The results in Appendix N.3 demonstrate that our local solution construction model achieves significantly lower computational complexity than both LEHD and INViT. This advantage is particularly evident in the substantially reduced runtime compared to existing representative models.

**Training and Inference Costs** We have conducted a detailed analysis of training and inference costs. As shown in Appendix N.4, L2R achieves strong performance without introducing excessive computational overhead. Meanwhile, the resource requirements of L2R are modest, and L2R can efficiently scale with problem size. This highlights the practicality of L2R for large-scale applications.

## 5.4 ABLATION STUDY

We conduct a detailed ablation study to demonstrate the effectiveness and robustness of L2R, mainly including: (1) **L2R vs. Distance-based SSR with Larger Search Space** (see Appendix O.1); (2) **Effects of Adaptation Bias in Compatibility** (see Appendix O.2); (3) **Comparison between different attention mechanisms** (see Appendix O.3); (4) **Effects of PRC under different initial solution generations** (see Appendix O.4); (5) **Effects of static reduction**(see Appendix O.5); (6) **Effects of different training scale** (see Appendix O.6).

## 6 CONCLUSION, LIMITATION, AND FUTURE WORK

In this work, we propose a novel RL-based Learning-to-Reduce (L2R) framework for solving large-scale vehicle routing problems. Our approach adaptively selects a small set of candidate nodes at each construction step, enabling efficient search space reduction while maintaining solution quality. Extensive experiments demonstrate that L2R, trained exclusively on 100-node instances from the uniform distribution, achieves remarkable performance on TSP and CVRP instances with up to 1 million nodes from the uniform distribution and over 80K nodes from other distributions.

**Limitation and Future Work** While L2R achieves impressive performance on large-scale TSP and CVRP instances, it currently relies on a simple combined loss for training. In future work, we aim to develop a more effective loss function to further improve the training efficiency of L2R. In addition, we plan to extend the L2R framework's applicability to tackle routing problems with complex constraints without compromising solution quality.

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

# A    DEFINITIONS AND FORMULATIONS OF TSP AND CVRP

In this section, we provide the definitions and mathematical formulations of the Traveling Salesman Problem (TSP) and the Capacitated Vehicle Routing Problem (CVRP), thereby enabling readers to obtain a clearer and more profound understanding of the fundamental nature of these problems.

## A.1    TRAVELING SALESMAN PROBLEM (TSP)

The TSP is one of the most representative combinatorial optimization problems, and we focus on the 2D Euclidean TSP in this paper (Bello et al., 2016; Kool et al., 2019). Given an instance $S = \{\mathbf{s}_i\}_{i=1}^n$ with node features $\mathbf{s}_i \in \mathbb{R}^2$, represented as a sequence of $n$ cities in a two-dimensional space. We are concerned with finding a permutation of the nodes $\pi = (\pi_1, \pi_2, \pi_3, \ldots, \pi_n)$, termed a tour, that visits each city once and has the minimum total length. We define the length of a tour defined by a permutation $\pi$ as

$$\text{minimize} \quad L(\pi|S) = \|\mathbf{s}_{\pi_n} - \mathbf{s}_{\pi_1}\| + \sum_{i=1}^{n-1} \|\mathbf{s}_{\pi_i} - \mathbf{s}_{\pi_{i+1}}\|, \tag{12}$$

where $\|\cdot\|$ represents $\ell_2$ norm.

## A.2    CAPACITATED VEHICLE ROUTING PROBLEM (CVRP)

A general formulation for CVRP with maximum vehicle number constraints $\epsilon_m$ Toth & Vigo (2002); Hou et al. (2022) is introduced in this section. Specifically, given a complete graph $G = (\Omega, E)$, binary variables $z_{ij} \in \{0, 1\}$ are defined to represent whether a vehicle traverses arc $(i, j)$ in the solution. $c_i$ and $\delta_i$ denote the remaining vehicle capacity after visiting customer $i$ and the demand of customer $i$, respectively. The maximum capacity of each vehicle is denoted as $C$, respectively. Given these variables, the formulation is given as follows.

$$\text{minimize} \quad \sum_{i \in \Omega, j \in \Omega, i \neq j} d_{ij} z_{ij} \tag{13}$$

$$\text{subject to} \quad \sum_{j \in \Omega, i \neq j} x_{ij} = 1, \quad \forall i \in \Omega \setminus \{0\} \tag{14}$$

$$\sum_{i \in \Omega, i \neq j} x_{ij} = 1, \quad \forall j \in \Omega \setminus \{0\} \tag{15}$$

$$c_j = \begin{cases} c_i - \delta_j, & \text{if } z_{ij} = 1, \quad \forall (i,j) \in \Omega \\ c_j, & \text{if } z_{ij} = 0, \quad \forall (i,j) \in \Omega \end{cases} \tag{16}$$

$$c_j - c_i \geq \delta_j - C(1 - z_{ij}), \quad \forall i, j \in \Omega \setminus \{0\}, i \neq j, d_i + d_j \leq C \tag{17}$$

$$0 \leq c_i \leq C - \delta_i, \quad i \in \Omega \setminus \{0\} \tag{18}$$

$$\sum_{j \in \Omega \setminus \{0\}} z_{j0} = \epsilon \tag{19}$$

$$\sum_{i \in \Omega \setminus \{0\}} z_{i0} = \epsilon \tag{20}$$

$$\epsilon \leq \epsilon_m \tag{21}$$

$$z_{ij} \in \{0, 1\}, \quad \forall i, j \in \Omega, i \neq j \tag{22}$$

where Equation (13) is objective function, $d_{ij}$ is the Euclidean distance from customer $i$ to customer $j$. Equation (14) and Equation (15) ensure that each customer is visited exactly once. Equation (16) is the updating process of remaining vehicle capacity and Equation (17)−Equation (18) define the capacity and connectivity constraints. Equation (19)−Equation (21) ensure that the number of vehicles departing from the depot equals the total number returning to the depot, i.e., each vehicle must start and terminate its route at the depot.

### A.3 CAPACITATED VEHICLE ROUTING PROBLEM WITH TIME WINDOWS

Consistent with the MVMoE (Zhou et al., 2024b), we set time windows of the depot to $[e_0, l_0]$=[0,3] and its service time to 0. The service time $s_i$ for each customer node is set to 0.2. For the time windows for each customer node, we generate them following the methodology outlined in MVMoE (Zhou et al., 2024b).

## B DISTANCE-BASED DYNAMIC SEARCH SPACE REDUCTION

In this section, we formally define the concept of distance-based search space reduction and its mathematical formulation (Fang et al., 2024; Drakulic et al., 2023; Gao et al., 2024; Wang et al., 2024). The "distance-based search space reduction" strategy directly applies the $k$-Nearest Neighbors (KNN) principle to select nodes. At each construction step, it selects $k$ candidates from all unvisited nodes based solely on the ascending order of Euclidean distance to the current node. Let $\Omega$ be the set of all nodes and $\pi_{1:t-1}$ be the set of visited nodes in the partial solution. At construction step $t$, for each node with coordinates $\mathbf{s}_i = \{x_i, y_i\}$, the distance $d_{t-1,i}$ and the set of candidate $\mathcal{N}_t$ can be calculated as:

$$d_{t-1,i} = \begin{cases} \infty & \text{if } i \in \pi_{1,t-1} \\ \sqrt{(x_i - x_{\pi_{t-1}})^2 + (y_i - y_{\pi_{t-1}})^2} & \text{otherwise} \end{cases}, \quad (23)$$

$$\mathcal{N}_t = \bigcup_{i \in \Omega} \{\mathbf{s}_i \mid \text{rank}(d_{t-1,i}) \leq k\} \quad (24)$$

where $\text{rank}(d_{t-1,i})$ and $k$ are the ascending order of distances and a hyperparameter defining the number of candidates.

## C DETAILED DESCRIPTIONS OF DIFFERENT SEARCH SPACE REDUCTIONS

In this section, we provide detailed descriptions of different search space reduction methods:

**Static SSR:** This is a one-time, upfront operation performed at the beginning of the solving process. It reduces the fully connected graph to a sparse one by directly removing unpromising edges based on a heuristic rule or a pretrained model. Static SSR is a practical technique for improving computational efficiency, a benefit also validated in recent studies (Sun et al., 2021; Fu et al., 2021). However, it cannot adapt to the dynamic state of the partial solution during construction.

**Dynamic SSR:** At each step of solution construction, the dynamic SSR model (Fang et al., 2024; Drakulic et al., 2023; Gao et al., 2024; Wang et al., 2024) selects a small subset of candidate nodes to consider for the next visit. This approach is adaptive, which leverages the dynamic state of the current partial solution to guide decisions. However, existing dynamic SSR methods typically rely on a simple $k$-nearest-neighbor rule based on distance ascending order, which can be problematic, as it may prune crucial long-distance connections necessary for optimal routing.

**Why Both Are Better Together:** Unlike traditional distance-based dynamic SSR, our proposed learning-based dynamic SSR can effectively mitigate the negative impact of dynamic reduction on solution optimality. However, this advantage comes at a cost: at each construction step, the reduction model must evaluate the potential of all unvisited nodes, which becomes computationally expensive for large-scale instances (e.g., N=100K). To address this challenge, L2R employs a hierarchical approach where static SSR first acts as a coarse, global filter to ensure computational tractability, followed by dynamic SSR that applies a fine-grained, adaptive policy for candidate selection. This framework allows us to leverage the benefits of learned dynamic reduction while maintaining computational efficiency.

## D  FURTHER ANALYSIS OF DISTANCE-BASED SEARCH SPACE REDUCTION

### D.1  IMPACT ON OPTIMALITY GAP

In this subsection, we provide detailed solution visualizations across TSPLIB instances of varying scales under different levels of search space reduction ($k$). As shown in Figure 5, this degradation is primarily attributed to the elimination of non-local node selections that are crucial for optimal routes, and the cumulative effect of over-pruning during the solution construction process leads to significant deviations from the optimal route in subsequent node selections, ultimately compromising the overall solution quality obtained by D-SSR.

### D.2  IMPACTS ON CONSTRUCTIVE NCOs

In this subsection, regarding the impacts of distance-based search space reduction across diverse constructive NCO architectures, we conduct further experimental analyses, which mainly include: (1) the impact for constructive NCO methods with a heavy decoder, (2) the impact for constructive NCO methods with a heavy encoder, and (3) the impact for the retrained constructive NCO method.

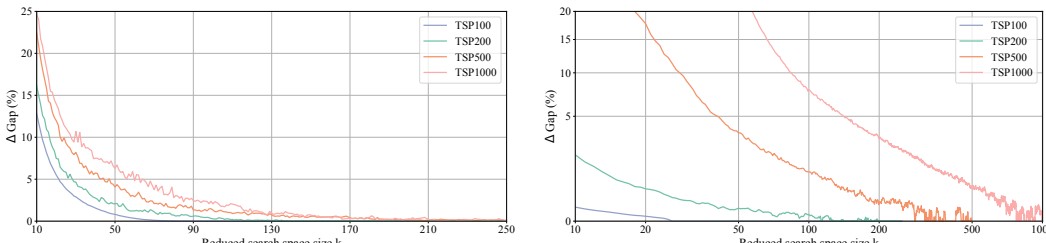

(a) LEHD with Light Encoder and Heavy Decoder.    (b) POMO with Heavy Encoder and Light Decoder.

Figure 4: The effect of distance-based search space reduction on the gap in diverse constructive NCO architectures (Left: LEHD (Luo et al., 2023); Right: POMO (Kwon et al., 2020)). Here $\Delta Gap = Gap_{\text{reduced}} - Gap_{\text{original}}$.

### D.2.1  IMPACT ON PRETRAINED CONSTRUCTIVE NCO

**Impact on Constructive NCO with Heavy Decoder**  To evaluate the impact of search space reduction on NCO performance, we use the well-known LEHD method (Luo et al., 2023) as a representative example, restricting the search space to the $k$ nearest nodes from the last visited node at each construction step. Without loss of generality, we use the LEHD-greedy without any RRC strategy. We quantify the impact by measuring the performance gap $\Delta Gap = Gap_{\text{reduced}} - Gap_{\text{original}}$, defined as the difference between the optimality gap obtained by LEHD with a reduced search space ($Gap_{\text{reduced}}$) and the original LEHD without search space reduction ($Gap_{\text{original}}$).

The results in Figure 4a illustrate the performance on TSP instances of varying sizes under different search space reduction levels ($k$). A key observation is the existence of a critical threshold $k^*$ for each problem size, beyond which LEHD with $k \geq k^*$ achieves a $0\%$ performance gap with the original LEHD, indicating no loss in solution quality with SSR. However, these critical thresholds vary significantly across problem sizes, and using a small $k < k^*$ leads to substantial performance degradation.

**Impact on Constructive NCO with Heavy Encoder**  In addition to SL-based LEHD with a heavy decoder architecture, we also evaluate the RL-based POMO model featuring a heavy encoder architecture, and the instance augmentation technology is not used in our experiment. As shown in Figure 4b, our experimental results demonstrate a consistent phenomenon with the SL-based LEHD case: The critical thresholds $k^*$ exhibit significant variations across different problem sizes, and using a small $k < k^*$ results in considerable performance deterioration.

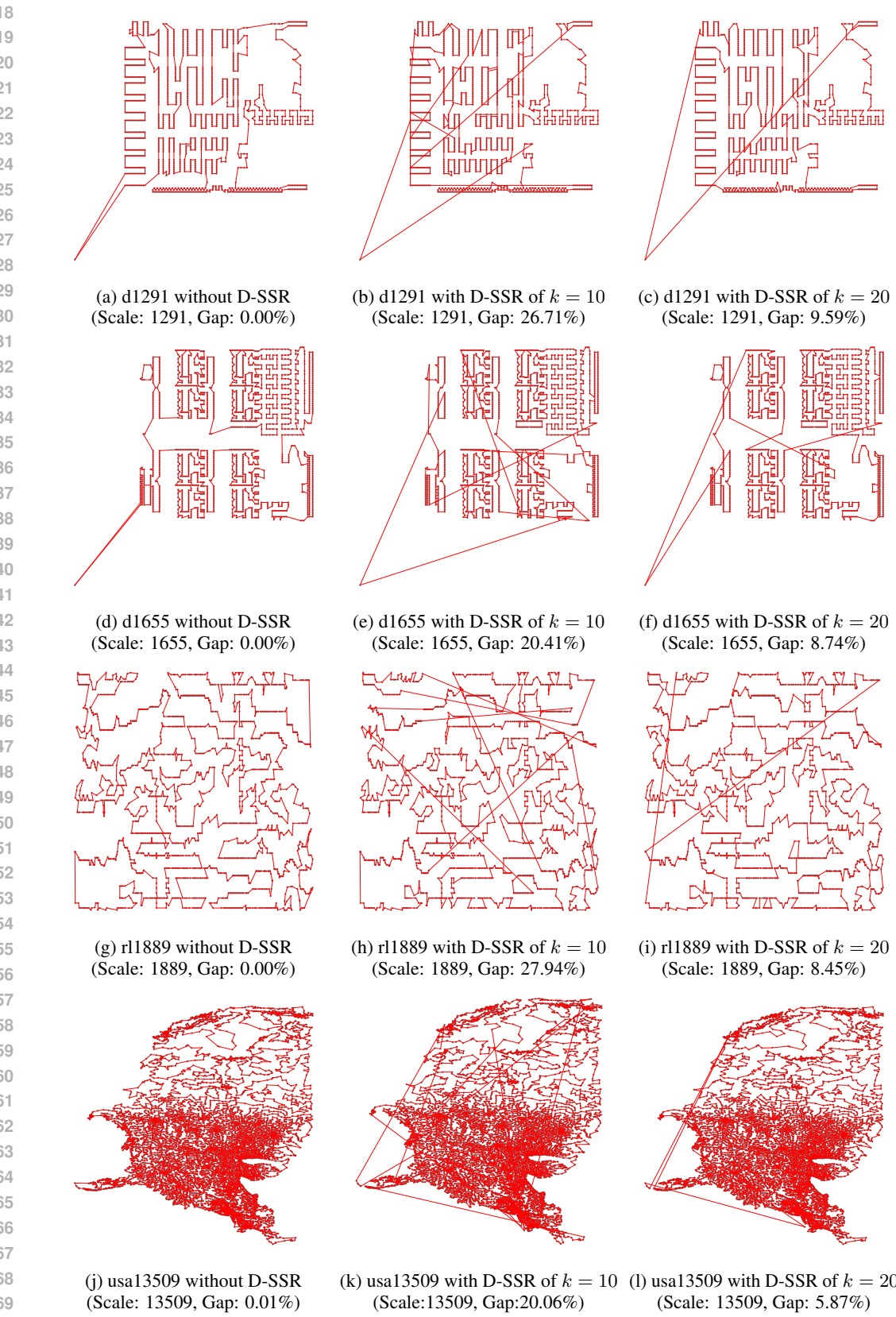

(a) d1291 without D-SSR
(Scale: 1291, Gap: 0.00%)

(b) d1291 with D-SSR of $k = 10$
(Scale: 1291, Gap: 26.71%)

(c) d1291 with D-SSR of $k = 20$
(Scale: 1291, Gap: 9.59%)

(d) d1655 without D-SSR
(Scale: 1655, Gap: 0.00%)

(e) d1655 with D-SSR of $k = 10$
(Scale: 1655, Gap: 20.41%)

(f) d1655 with D-SSR of $k = 20$
(Scale: 1655, Gap: 8.74%)

(g) rl1889 without D-SSR
(Scale: 1889, Gap: 0.00%)

(h) rl1889 with D-SSR of $k = 10$
(Scale: 1889, Gap: 27.94%)

(i) rl1889 with D-SSR of $k = 20$
(Scale: 1889, Gap: 8.45%)

(j) usa13509 without D-SSR
(Scale: 13509, Gap: 0.01%)

(k) usa13509 with D-SSR of $k = 10$
(Scale:13509, Gap:20.06%)

(l) usa13509 with D-SSR of $k = 20$
(Scale: 13509, Gap: 5.87%)

Figure 5: The solution visualizations of TSPLIB instances under different levels of search space. All solutions are obtained by the LKH-3 algorithm (Helsgaun, 2017).

### D.2.2 IMPACT ON RETRAINED CONSTRUCTIVE NCO

Considering that the inconsistency between training and inference strategies may introduce a deviation between the model's learned perception and its actual decision-making process, we retrain three reduced variants of LEHD with different search space reduction configurations. As illustrated in Table 7, while reducing the search space significantly decreases inference time, employing an excessively small $k$ based on pairwise distance information results in substantial performance deterioration.

Table 7: The effect of distance-based search space reduction on the gap in the LEHD model. Note that all LEHD models with search space reduction are retrained with the same specifications.

| Method | TSP100 | | TSP200 | | TSP500 | | TSP1000 | |
|---|---|---|---|---|---|---|---|---|
| | Gap | Time | Gap | Time | Gap | Time | Gap | Time |
| LKH3 | 0.00% | 56m | 0.00% | 4m | 0.00% | 32m | 0.00% | 8.2h |
| LEHD w/ $k = 20$ | 3.10% | **10.7s** | 5.21% | **1.0s** | 7.95% | **2.1s** | 11.09% | **3.7s** |
| LEHD w/ $k = 30$ | 1.71% | 15s | 3.03% | 1.1s | 6.01% | 2.2s | 9.44% | 4s |
| LEHD w/ $k = 50$ | 0.87% | 18.7s | 1.74% | 1.28s | 4.03% | 2.6s | 7.41% | 4.9s |
| LEHD w/o SSR | **0.58%** | 24s | **0.86%** | 3s | **1.56%** | 18s | **3.17%** | 1.6m |

These findings highlight the potential of search space reduction while underscoring the need for advanced reduction methods capable of efficiently handling large-scale problems with a small $k$.

# E   COMPATIBILITY WITH ADAPTATION BIAS

As defined in Equation (6), the adaptation bias between each feasible node $i \in \mathcal{A}_t$ and the current node $\pi_{t-1}$ (selected by the local construction model) is given by $a^R_{t-1,i} = -\alpha \cdot \log_2 N \cdot d_{t-1,i}$, following Zhou et al. (2024a). Here, $N$ denotes the total number of nodes (i.e., problem size), and $\alpha > 0$ is a learnable parameter with a default value of 1 that enables the model to adaptively learn information-specific weights. This function incorporates both distance $d_{t-1,i}$ and scale information $\log_2 N$ to better capture varying geometric variations with different scales. Accurately identifying candidate nodes within a large search space remains challenging when relying solely on a lightweight network architecture. Unlike existing distance-based reduction methods, our approach addresses the trade-off between learning difficulty and solution quality by incorporating a distance-assisted reduction model to enhance neighborhood selection performance.

Additionally, we extend this adaptation bias to the compatibility computation in the local construction model. In Equation (8), the bias between each candidate node $i \in \mathcal{N}_t$ and the current node $\pi_{t-1}$ is defined as $a^L_{t-1,i} = -\alpha \cdot \log_2 |\mathcal{N}_t| \cdot d_{t-1,i}$. Here, $|\mathcal{N}_t|$ denotes the total number of candidate nodes.

To evaluate the effectiveness of two adaptation biases, we conduct an ablation study with different component configurations, and the detailed results are provided in Appendix O.2.

# F   ADAPTATION ATTENTION FREE MODULE

Following Zhou et al. (2024a), we implement $\text{Attention}(\cdot)$ using a scale-distance adopted adaptation attention free module (AAFM) to enhance geometric pattern recognition for routing problems. Given the input $X$, AAFM first transforms it into $Q$, $K$, and $V$ through corresponding linear projection operations:

$$Q = XW^Q, \quad K = XW^K, \quad V = XW^V, \tag{25}$$

where $W^Q$, $W^K$, and $W^V$ are learnable matrices. The AAFM computation is then expressed as:

$$\text{Attention}(Q, K, V, A) = \sigma(Q) \odot \frac{\exp(A)(\exp(K) \odot V)}{\exp(A)\exp(K)}, \tag{26}$$

where $\sigma$ denotes the sigmoid function, $\odot$ represents the element-wise product, and $A = \{a_{ij}\}$ denotes the pair-wise adaptation bias. For more details about $a_{ij}$, please see Appendix E.

Compared to multi-head attention (MHA) (Vaswani et al., 2017), AAFM enables the model to explicitly capture instance-specific knowledge by updating pair-wise adaptation biases while exhibiting lower computational overhead. Further details are provided in the related work section mentioned above. For a detailed discussion of the comparison of AAFM and MHA in the L2R framework, please refer to Appendix O.3.

## G FURTHER ANALYSIS OF STATIC REDUCTION

Static search space reduction is a practical technique for improving computational efficiency, a benefit consistently validated in related studies. Our approach builds upon prior work, including MLPR (Sun et al., 2021), which employs model predictions to eliminate unpromising edges in large-scale COPs. We also draw inspiration from recent advancements, such as DIMES (Qiu et al., 2022), DIFUSCO (Sun & Yang, 2023), and T2T (Li et al., 2023), which utilize graph sparsification to enhance solving efficiency. In our L2R framework, the static search space reduction serves as a conservative, one-time preprocessing step that filters out clearly non-optimal edges (typically the longest edges) without compromising the search space for subsequent learning-based dynamic reduction.

To determine an appropriate threshold $\gamma$, we have conducted a statistical analysis of the optimality ratio, defined as the proportion of construction steps where the (near-)optimal next-visit node resides within the $k$-nearest candidates. We generate TSP instances at three scales ($N = 100, N = 1,000, N = 10,000$) across four distinct distributions (uniform, clustered, explosion, implosion). Specifically, the dataset comprises 10,000 instances per setting for $N = 100$, and 1,000 instances per setting for the larger scales ($N = 1,000$ and $10,000$). We use LKH3 to generate near-optimal solutions as ground truth.

As illustrated in Figure 6, the optimality ratio increases monotonically with the neighbor size $k$, and each distribution exhibits a critical threshold $k^*$ such that all optimal nodes are preserved when $k \leq k^*$. We observe that the edges belonging to the farthest 10% (long-distance connections) almost never appear in the optimal solutions across all tested distributions. Therefore, we set $\gamma = 10\%$ (retaining the top 90% of edges) as a safe boundary. This ensures that it preserves the potential for global optimality while reducing memory and computational overhead for the subsequent steps.

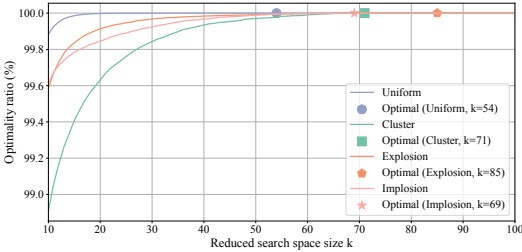

(a) TSP100

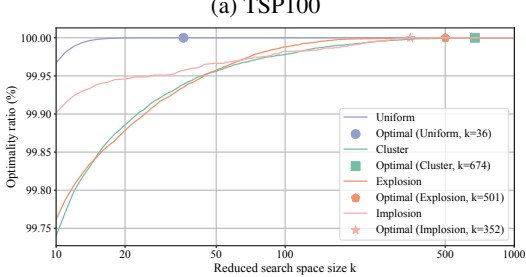

(b) TSP1,000

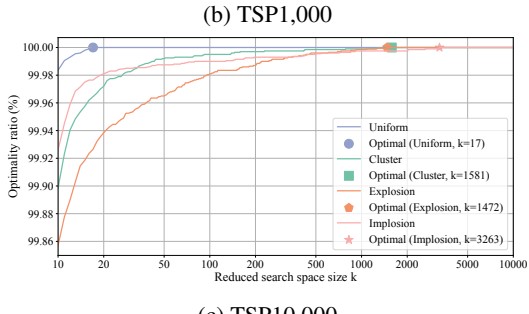

(c) TSP10,000

Figure 6: The effects of distance-based search space reduction on optimality ratio with different sizes and different distributions.

## H LEARNING-BASED REDUCTION MODEL

We develop a learning-based reduction model to dynamically evaluate the potential of feasible nodes and adaptively reduce the search space at each construction step. Our proposed model ensures efficiency through a lightweight architecture consisting of only an embedding layer and an attention layer. For CVRPs, we introduce a specialized treatment for each node to address the unique demands of the problem. In this section, we provide a detailed description of this specialized treatment for CVRPs in the reduction model.

**Context Embedding** The context embedding used in our model is mathematically justified by the structure of the routing problem, which is naturally modeled as a Markov Decision Process. For a routing problem, the transition probability depends only on the current state and the remaining unvisited nodes, rather than the full set of visited nodes along the route. Given the Hamiltonian cycle constraint, the current state is sufficiently defined by the current node ($\pi_{t-1}$) and the origin/target node ($\pi_1$). This formulation is widely adopted in current representative neural routing solvers (e.g.,

INViT (Fang et al., 2024) and BQ-NCO (Drakulic et al., 2023)). A general context embedding is often defined as $\mathbf{h}_{C_D}^t = W_c[\mathbf{h}_{\pi_1}^D, \mathbf{h}_{\pi_{t-1}}^D]$, where $W_c \in \mathbb{R}^{d \times 2d}$ is a learnable matrix. The approach used in our paper is mathematically equivalent to the standard practice mentioned above. For computational efficiency, we decompose $W_c$ into $W_{\text{first}} \in \mathbb{R}^{d \times d}$ and $W_{\text{last}} \in \mathbb{R}^{d \times d}$, resulting in our formulation $\mathbf{h}_{C_D}^t = W_{\text{first}}\mathbf{h}_{\pi_1}^D + W_{\text{last}}\mathbf{h}_{\pi_{t-1}}^D$.

**Embedding Layer for CVRPs**   Given an instance $S = \{\mathbf{s}_i\}_{i=1}^{N+1}$ with node features $\mathbf{s}_i \in \mathbb{R}^{d_x}$ (i.e., node coordinates and demand), we feed depot and nodes into a unified linear layer. The detailed calculation is expressed as

$$\mathbf{h}_i^D = W^{(e)}[x_i, \, y_i, \, \delta_i] + \mathbf{b}^{(e)}, \, i = 1, \ldots, N + 1, \tag{27}$$

where $x_i$ and $y_i$ denote the node coordinates, $\delta_i$ is the demand of node $i$, we regard the demand of depot as 0 (i.e., $\delta_0 = 0$), $W^{(e)} \in \mathbb{R}^{3 \times d}$ and $\mathbf{b}^{(e)} \in \mathbb{R}^d$ are learnable parameters. This process yields a set of embeddings $H^D = \{\mathbf{h}_i^D\}_{i=1}^{N+1}$ for all nodes in the instance $S$. At the $t$-th step, with a partial solution following Kwon et al. (2020), we define a context embedding for the current partial solution:

$$\mathbf{h}_{(C_D)}^t = W_{\text{last}}[\mathbf{h}_{\pi_{t-1}}^D, Q_{\text{remain}}], \tag{28}$$

where $W_{\text{last}} \in \mathbb{R}^{(1+d) \times d}$ is a learnable matrix, and $Q_{\text{remain}}$ represents the current remaining capacity.

**Embedding Layer for CVRPTWs**   Given an instance $S = \{\mathbf{s}_i\}_{i=1}^{N+1}$ with node features $\mathbf{s}_i \in \mathbb{R}^{d_5}$ (i.e., node coordinates, demand, earliest arrival time, and latest arrival time). For depot, the demand, earliest arrival time, and latest arrival time are zero-padded. We feed the depot and nodes into a unified linear layer. The detailed calculation is expressed as

$$\mathbf{h}_i^D = W^{(e)}[x_i, \, y_i, \, \delta_i, e_i, l_i] + \mathbf{b}^{(e)}, \, i = 1, \ldots, N + 1, \tag{29}$$

where $x_i$ and $y_i$ denote the node coordinates, $\delta_i, e_i, l_i$ are the demand, earliest arrival time, and latest arrival time of node $i$, respectively. For depot, the demand, earliest arrival time, and latest arrival time are zero-padded. $W^{(e)} \in \mathbb{R}^{5 \times d}$ and $\mathbf{b}^{(e)} \in \mathbb{R}^d$ are learnable parameters. This process yields a set of embeddings $H^D = \{\mathbf{h}_i^D\}_{i=1}^{N+1}$ for all nodes in the instance $S$. At the $t$-th step, with a partial solution following Kwon et al. (2020), we define a context embedding for the current partial solution:

$$\mathbf{h}_{(C_D)}^t = W_{\text{last}}[\mathbf{h}_{\pi_{t-1}}^D, Q_{\text{remain}}], \tag{30}$$

where $W_{\text{last}} \in \mathbb{R}^{(1+d) \times d}$ is a learnable matrix, and $Q_{\text{remain}}$ represents the current remaining capacity.

**Masking**   The static reduction stage performs partial search space reduction to obtain the pruned graph $G'$. In particular, we do not allow nodes to be visited if their remaining demand is either 0 (indicating the node has already been visited) or exceeds the vehicle's remaining capacity and time windows. We denote $\mathcal{A}_t$ as the set of all feasible nodes, and the embeddings of $\mathcal{A}_t$ are represented by $H_{\mathcal{A}_t}^D = \{\mathbf{h}_i^D \mid i \in \mathcal{A}_t\} \in \mathbb{R}^{|\mathcal{A}_t| \times d}$.

**The Advantage of L2R**   Standard attention mechanisms typically require interactions between the current state and every feasible node at each construction step. While effective at small scales, this approach becomes computationally prohibitive for large-scale problems (i.e., $N \geq 10K$). To address this, we employ a lightweight reduction model. However, identifying promising candidates within a large search space using only a lightweight network remains a challenging task. Therefore, we designed a distance-assisted reduction model, which can adaptively combine distance-based priors with context-dependent features learned by the network to generate potential scores for unvisited nodes. For further results and analysis, please see Section O.2.

The key contribution of L2R is its ability to perceive and utilize node attributes beyond simple spatial proximity (e.g., demands and time windows), enabling it to effectively apply to more complex routing problems, such as CVRP and CVRPTW, where the optimal solution depends not only on geometric distance but also heavily on customer demands and time window information. Extensive experiments demonstrate that L2R significantly outperforms methods that rely solely on distance rules (such as INViT (Fang et al., 2024)). Moreover, L2R can also demonstrate unprecedented generalization capabilities. To the best of our knowledge, it is the first constructive neural solver to surpass HGS on large-scale CVRP instances ($N \geq 50K$) when trained solely on 100-node uniform instances without any post-processing operations.

# I LOCAL SOLUTION CONSTRUCTION MODEL

The proposed local solution construction model includes four components: 1) coordinate normalization, 2) embedding layer, 3) attention layer, and 4) compatibility calculation, which are detailed in the following subsections.

## I.1 COORDINATE NORMALIZATION

We introduce a coordinate normalization operation to ensure that each extracted sub-graph $G'_{\text{sub}}$ adheres to a similar distribution (Fu et al., 2021; Ye et al., 2024; Fang et al., 2024). This operation not only simplifies the input feature space but also significantly enhances the stability and homogeneity of model inputs across different sub-graphs. Following Fu et al. (2021) and Fang et al. (2024), the coordinate transformation is formulated as

$$x^{\min} = \min_{i \in \mathcal{N}_t} x_i, \quad x^{\max} = \max_{i \in \mathcal{N}_t} x_i,$$
$$y^{\min} = \min_{i \in \mathcal{N}_t} y_i, \quad y^{\max} = \max_{i \in \mathcal{N}_t} y_i,$$

(31)

$$r = \frac{1}{\max(x^{\max} - x^{\min}, y^{\max} - y^{\min})},$$

(32)

$$x_i^{new} = r \times (x_i - x^{\min}) \quad \forall i \in \{\pi_1, \pi_{t-1}\} \cup \mathcal{N}_t,$$
$$y_i^{new} = r \times (y_i - y^{\min}) \quad \forall i \in \{\pi_1, \pi_{t-1}\} \cup \mathcal{N}_t,$$

(33)

where $x_i$ and $y_i$ denote the node coordinates, and $\mathcal{N}_t$ represents the $k$ candidate nodes generated based on $\pi_{t-1}$ and the dynamic reduction model. To ensure that the first visited node (or depot) $\pi_1$ remains within the boundary (i.e., $0 \le x_i^{new} \le 1$), we apply $x_i^{new} = \max(0, \min(x_i^{new}, 1))$, and the same operation is applied to $y_i^{new}$. Equations equation 31–equation 33 effectively simplify the input feature space and improve the stability and homogeneity of model inputs across different sub-graphs. Subsequently, the sub-graph $G'_{\text{sub}}$ is transformed into a new graph $G''_{\text{sub}}$.

## I.2 EMBEDDING LAYER

**Traveling Salesman Problem**   Given the converted sub-graph $G''_{\text{sub}}$, the embedding layer first transforms the coordinates into initial embeddings using a shared linear layer with learnable parameters $[W^{(0)} \in \mathbb{R}^{d_x \times d}; b^{(0)} \in \mathbb{R}^d]$. The embeddings of the $k$ candidate nodes $\mathcal{N}_t$ are denoted by $H_{\mathcal{N}}^{(0)} = \{\mathbf{h}_i^{(0)} \mid i \in \mathcal{N}_t\} \in \mathbb{R}^{k \times d}$. Here, the first node $\pi_1$ and the last node $\pi_{t-1}$ are used to represent the current partial solution. Therefore, their initial embeddings require special treatment (Drakulic et al., 2023; Luo et al., 2023). Specifically, additional learnable matrices $W_1 \in \mathbb{R}^{d \times d}$ and $W_2 \in \mathbb{R}^{d \times d}$ are applied to $\mathbf{h}_{\pi_1}^{(0)}$ and $\mathbf{h}_{\pi_{t-1}}^{(0)}$, respectively. Accordingly, we define the initial graph node embeddings $\widetilde{H}^{(0)} \in \mathbb{R}^{(2+|\mathcal{N}_t|) \times d}$ as

$$\widetilde{H}^{(0)} = [W_1 \mathbf{h}_{\pi_1}^{(0)}, \ W_2 \mathbf{h}_{\pi_{t-1}}^{(0)}, \ H_{\mathcal{N}_t}^{(0)}],$$

(34)

where $[\cdot, \cdot]$ denotes the vertical concatenation operator. Next, $\widetilde{H}^{(0)}$ is passed through the $L$ attention layers sequentially.

**Capacitated Vehicle Routing Problem**   For CVRP, due to the demands and capacity constraints, we introduce a specialized treatment for the initial embedding of each node. Specifically, the node demands are normalized as $\delta_i = \{\delta_i / Q_{\text{remain}} \mid i \in \mathcal{N}_t\}$, where $Q_{\text{remain}}$ represents the remaining capacity. We separate the coordinates and demands and feed them into distinct linear layers. The detailed calculation is expressed as

$$\mathbf{h}_{i_{\text{coor}}}^{(0)} = W^{(0)}[x_i, y_i] + \mathbf{b}^{(0)} \quad \forall i \in \{\pi_1, \pi_{t-1}\} \cup \mathcal{N}_t,$$
$$\mathbf{h}_i^{(0)} = \mathbf{h}_{i_{\text{coor}}}^{(0)} + W_{\text{demand}} \delta_i \quad \forall i \in \mathcal{N}_t,$$
$$\mathbf{h}_{\pi_1}^{(0)} = W_1 \mathbf{h}_{\pi_{1_{\text{coor}}}}^{(0)} + W_{\text{load}} Q_{\text{remain}},$$
$$\mathbf{h}_{\pi_{t-1}}^{(0)} = W_2 \mathbf{h}_{\pi_{t-1_{\text{coor}}}}^{(0)} + W_{\text{load}} Q_{\text{remain}},$$

(35)

where $W_{\text{demand}} \in \mathbb{R}^{1 \times d}$ and $W_{\text{load}} \in \mathbb{R}^{1 \times d}$ are learnable matrices used for demand and remaining capacity, respectively.

**Capacitated Vehicle Routing Problem with Time Windows**    For CVRPTW, due to the time windows and capacity constraints, we introduce a specialized treatment for the initial embedding of each node. Specifically, the node demands are normalized as $\delta_i = \{\delta_i / Q_{\text{remain}} \mid i \in \mathcal{N}_t\}$, where $Q_{\text{remain}}$ represents the remaining capacity and we add the end time of serving the current node to first and last node embeddings. We separate the coordinates and demands and feed them into distinct linear layers. The detailed calculation is expressed as

$$
\begin{aligned}
\mathbf{h}^{(0)}_{i_{\text{coor}}} &= W^{(0)}[x_i, y_i] + \mathbf{b}^{(0)} \quad \forall i \in \{\pi_1, \pi_{t-1}\} \cup \mathcal{N}_t, \\
\mathbf{h}^{(0)}_i &= \mathbf{h}^{(0)}_{i_{\text{coor}}} + W_{\text{demand\_time}}[\delta_i, e_i, l_i] \quad \forall i \in \mathcal{N}_t, \\
\mathbf{h}^{(0)}_{\pi_1} &= W_1 \mathbf{h}^{(0)}_{\pi_{1\,\text{coor}}} + W_{\text{load\_time}}[Q_{\text{remain}}, T_{\text{current}}], \\
\mathbf{h}^{(0)}_{\pi_{t-1}} &= W_2 \mathbf{h}^{(0)}_{\pi_{t-1\,\text{coor}}} + W_{\text{load\_time}}[Q_{\text{remain}}, T_{\text{current}}],
\end{aligned}
\tag{36}
$$

where $W_{\text{demand}} \in \mathbb{R}^{1 \times d}$ and $W_{\text{load}} \in \mathbb{R}^{1 \times d}$ are learnable matrices used for demand and remaining capacity, respectively. $T_{\text{current}}$ is the end time of serving the current node.

### I.3  ATTENTION LAYER

Inspired by BQ (Drakulic et al., 2023) and LEHD (Luo et al., 2023), our attention layer consists of two sub-layers: an attention sub-layer and a feed-forward (FF) sub-layer. Both sub-layers incorporate Layer Normalization (Ba et al., 2016) and skip connections (He et al., 2016).

Let $\widetilde{H}^{(\ell-1)} = [\mathbf{h}^{(\ell-1)}_{\pi_1}, \mathbf{h}^{(\ell-1)}_{\pi_{t-1}}, H^{(\ell-1)}_{\mathcal{N}_t}]$ denote the input to the $\ell$-th attention layer for $\ell = 1, \ldots, M$. The outputs for the $i$-th node are computed as follows:

$$
\hat{\mathbf{h}}^{(\ell)}_i = \text{LN}^{(\ell)}\left(\mathbf{h}^{(\ell-1)}_i + \text{Attention}^{(\ell)}\left(\mathbf{h}^{(\ell-1)}_i, \widetilde{H}^{(\ell-1)}\right)\right),
\tag{37}
$$

$$
\mathbf{h}^{(\ell)}_i = \text{LN}^{(\ell)}\left(\hat{\mathbf{h}}^{(\ell)}_i + \text{FF}^{(\ell)}\left(\hat{\mathbf{h}}^{(\ell)}_i\right)\right),
\tag{38}
$$

where $\text{LN}(\cdot)$ denotes layer normalization (Ba et al., 2016), which mitigates potential value overflows caused by exponential operations; $\text{Attention}$ in Equation (37) represents the adopted attention mechanism (see Appendix F for details), and $\text{FF}(\cdot)$ in Equation (38) corresponds to a fully connected neural network with ReLU activation.

After $M$ attention layers, the final node embeddings $\widetilde{H}^{(M)} = [\mathbf{h}^{(M)}_{\pi_1}, \mathbf{h}^{(M)}_{\pi_{t-1}}, H^{(M)}_{\mathcal{N}_t}] \in \mathbb{R}^{(2+|\mathcal{N}_t|) \times d}$ encapsulate the advanced feature representations of the first node, last node, and $k$ candidate nodes.

### I.4  PADDING

To address the variability in the number of candidate nodes when solving a batch of instances, we pad the number of candidate nodes $|\mathcal{N}_t|$ to the maximum length by adding an all-zero tensor. The maximum length is computed as $\max\{\min\{k, |\mathcal{A}_{i,t}|\} \mid i \in \{1, \ldots, B\}\}$, where $k$ is the predefined search space size, $|\mathcal{A}_{i,t}|$ denotes the number of feasible unvisited nodes at the current step $t$ for instance $S_i$, and $B$ represents the batch size. An attention mask is then applied to mask out the padded zeros during computation.

## J    PSEUDOCODE FOR TRAINING

Following Kool et al. (2019), we employ an exponential baseline ($\beta = 0.8$) during the first epoch to stabilize initial learning. The baseline parameters $\boldsymbol{\theta}^{BL}$ are updated only if the improvement is statistically significant, as determined by a paired t-test ($\alpha = 5\%$) conducted on 10,000 separate evaluation instances at the end of each epoch. If the baseline policy is updated, new evaluation instances are sampled to prevent overfitting. Additionally, at the first construction step for TSP, $\pi_1$ is randomly selected as the initial partial solution and is treated as both the initial and last nodes for the first decoding steps.

---

**Algorithm 1** L2R Training with REINFORCE Algorithm

---

1: **Input:** number of epochs $E$, steps per epoch $T$, batch size $B$, significance $\alpha$, the percentage $\gamma$ of static reduction.
2: **Output:** The trained model with parameters $\boldsymbol{\theta} = \{\boldsymbol{\theta}_R, \boldsymbol{\theta}_L\}$, which are reduction model $\boldsymbol{\theta}_R$ and local construction model $\boldsymbol{\theta}_L$.
3: Init $\boldsymbol{\theta}_R$, $\boldsymbol{\theta}_R^{BL} \leftarrow \boldsymbol{\theta}_R$ and $\boldsymbol{\theta}_L$, $\boldsymbol{\theta}_L^{BL} \leftarrow \boldsymbol{\theta}_L$.
4: **for** epoch $= 1, \ldots, E$ **do**
5:    **for** step $= 1, \ldots, T$ **do**
6:       $G_i \leftarrow$ RandomInstance() $\forall i \in \{1, \ldots, B\}$
7:       $G_i' \leftarrow$ StaticReduction($\gamma$) $\forall i \in \{1, \ldots, B\}$
8:       *// For $\boldsymbol{\theta}$, we use a decoding strategy of sampling according to the output probabilities.*
9:       **while** not done **do**
10:          *// $\mathcal{N}_{i,t}$ denotes the $k$ high-potential candidate nodes at step $t$ given sparse graph $G_i'$ generated by $\boldsymbol{\theta}_R$.*
11:          *// This selection $\tau_{i,t}$ is only used for the loss calculation of the reduction model.*
12:          $\mathcal{N}_{i,t}, \tau_{i,t} \leftarrow$ DynamicReduction($G_i'$, $\pi_{t-1}$, $\boldsymbol{\theta}_R$) $\forall i \in \{1, \ldots, B\}$
13:          *// This selection $\pi_{i,t}$ generated by $\boldsymbol{\theta}_L$ is used as the node selection for the current step $t$.*
14:          $\pi_{i,t} \leftarrow$ LocalConstruction($\pi_1$, $\pi_{t-1}$, $\mathcal{N}_{i,t}$, $\boldsymbol{\theta}_L$) $\forall i \in \{1, \ldots, B\}$
15:       **end while**
16:       *// For $\boldsymbol{\theta}^{BL}$, we use a greedy decoding strategy according to the output probabilities.*
17:       **while** not done **do**
18:          $\mathcal{N}_{i,t}^{BL} \leftarrow$ DynamicReduction($G_i'$, $\pi_{t-1}^{BL}$, $\boldsymbol{\theta}_R^{BL}$) $\forall i \in \{1, \ldots, B\}$
19:          *// For $\boldsymbol{\theta}^{BL}$, we only obtain the current node selection $\pi_{i,t}^{BL}$ of $\boldsymbol{\theta}_L^{BL}$ for calculating the reward.*
20:          $\pi_{i,t}^{BL} \leftarrow$ LocalConstruction($\pi_1^{BL}$, $\pi_{t-1}^{BL}$, $\mathcal{N}_{i,t}^{BL}$, $\boldsymbol{\theta}_L^{BL}$) $\forall i \in \{1, \ldots, B\}$
21:       **end while**
22:       $\nabla_{\boldsymbol{\theta}_R} \mathcal{L}_R(\boldsymbol{\theta}_R) \leftarrow \frac{1}{B} \sum_{i=1}^{B} \left( \mathcal{R}(\pi_i) - \mathcal{R}(\boldsymbol{\pi}_i^{BL}) \right) \nabla_{\boldsymbol{\theta}_R} \log o(\tau_i)$
23:       $\nabla_{\boldsymbol{\theta}_L} \mathcal{L}_L(\boldsymbol{\theta}_L) \leftarrow \frac{1}{B} \sum_{i=1}^{B} \left( \mathcal{R}(\pi_i) - \mathcal{R}(\boldsymbol{\pi}_i^{BL}) \right) \nabla_{\boldsymbol{\theta}_L} \log p(\pi_i)$
24:       $\mathcal{L}_{\text{Joint}}(\boldsymbol{\theta}) \leftarrow \mathcal{L}_R(\boldsymbol{\theta}_R) + \mathcal{L}_L(\boldsymbol{\theta}_L)$
25:       $\boldsymbol{\theta} \leftarrow$ Adam($\boldsymbol{\theta}, \nabla \mathcal{L}_{\text{Joint}}$)
26:    **end for**
27:    **if** OneSidedPairedTTest($p_{\boldsymbol{\theta}}, p_{\boldsymbol{\theta}^{\text{BL}}}$) $< \alpha$ **then**
28:       $\boldsymbol{\theta}_R^{BL} \leftarrow \boldsymbol{\theta}_R$
29:       $\boldsymbol{\theta}_L^{BL} \leftarrow \boldsymbol{\theta}_L$
30:    **end if**
31: **end for**

---

We chose Reinforcement Learning (RL) based on the specific challenges of broad routing problems and the core philosophy of NCO.

**Comparison with Supervised Learning (SL):**    SL relies heavily on high-quality ground truth labels. However, for complex routing variants, obtaining a large amount of near-optimal solutions is computationally prohibitive. This often requires (computationally) expensive exact solvers or extensive runs of leading heuristics, which severely limits the practical applicability of SL-based methods.

**Comparison with Self-Supervised Learning (SSL):** While SSL can reduce the need for extensive (nearly) optimal solutions, effective SSL in the NCO community typically requires designing advanced search strategies to generate high-quality pseudo-labels for self-training. Developing these strategies demands substantial domain-specific expert knowledge and careful fine-tuning.

Our primary objective aligns with the fundamental goal of NCO: to minimize reliance on domain expertise. RL enables the model to learn rules by repeatedly interacting with the environment, without requiring any labeled data or manually well-designed search strategies. This enables L2R to be effectively applied across multiple routing problems, such as TSP, CVRP, and CVRPTW, with outstanding large-scale generalization performance, which significantly facilitates the practical application of the model.

## K  TRAINING AND MODEL SETTINGS

Here, we provide the hyperparameter settings for the model and training process, and the detailed information can be found in Table 8.

Table 8: Training settings in our experiments.

| Hyperparameter | TSP | CVRP | CVRPTW |
|---|---|---|---|
| Optimizer | | Adam | |
| Initial learning rate of $\boldsymbol{\theta}_R$ | | $10^{-4}$ | |
| Learning rate decay of $\boldsymbol{\theta}_R$ | | 0.98 per epoch | |
| Initial learning rate of $\boldsymbol{\theta}_L$ | | $10^{-4}$ | |
| Learning rate decay of $\boldsymbol{\theta}_L$ | | 0.98 per epoch | |
| Batch size | 180 | 60 | 128 |
| Batches of each epoch | | $2,500$ | |
| Training scale | | 100 | |
| Epochs | | 100 | |
| The number of attention layer of $\boldsymbol{\theta}_L$ | | 6 | |
| Embedding dimension | | 128 | |
| Feed forward dimension | | 512 | |
| Clipping parameter $\xi$ | | 10 | |
| Training capacity | $-$ | 50 | 50 |
| Maximum search space size $k$ | 20 | 50 | 15 |
| Percentage $\gamma$ of static reduction | | 10% | |
| Gradient clipping | | max_norm=1.0 | |
| Training baseline (epoch $= 1$) | | Exponential baseline ($\beta = 0.8$) | |
| Training baseline (epoch $>1$) | | Greedy rollout baseline | |

# L  FURTHER EXPERIMENTS

## L.1  DETAILED RESULTS ON CVRPTW

According to the results in Table 9, our proposed L2R consistently achieves the best solution quality, complemented by remarkably fast inference times, across various problem scales. The advantages of L2R over existing neural solvers are significant. These results confirm that our learning-based reduction successfully captures complex, non-spatial constraints that simple distance-based heuristics cannot, proving the broad applicability of the L2R framework beyond simple routing tasks.

Table 9: Comparison on CVRPTW instances. Note that "Time" indicates the total running time. We limit the test scale to 10,000 nodes, as existing NCO methods consistently run out of Memory (OOM) on instances where $N \geq 5,000$.

| Method | CVRPTW1K Obj.(Gap) | Time | CVRPTW5K Obj.(Gap) | Time | CVRPTW10K Obj.(Gap) | Time |
|---|---|---|---|---|---|---|
| HGS-PyVRP | 159.35(0.00%) | 20m | 787.25(0.00%) | 3.3h | 1375.84 (0.00%) | 5h |
| MTPOMO aug×8 | 219.22(37.57%) | 1.7m | OOM | | OOM | |
| MVMoE aug×8 | 233.53(46.55%) | 1.7m | OOM | | OOM | |
| ReLD-MTL aug×8 | 183.31(15.036%) | 1.6m | OOM | | OOM | |
| L2R greedy | **174.31 (9.39%)** | **8s** | **849.83 (7.95%)** | **40s** | **1497.30 (8.83%)** | **1.4m** |

## L.2  PERFORMANCE ON SMALL-SCALE INSTANCES

Table 10: Experimental results on varying-scale TSP and CVRP instances with uniform distribution. All models are trained on instances of size N = 100. Note that "Time" indicates the total running time, and all methods employ greedy decoding, without utilizing instance augmentation or additional search strategies (e.g., RRC or PRC).

| Method | TSP100 (128 ins.) Obj. | Time | TSP500 (128 ins.) Obj. | Time | TSP50K (16 ins.) Obj. | Time | TSP100K (16 ins.) Obj. | Time |
|---|---|---|---|---|---|---|---|---|
| LKH3 | 7.73 | 1m | 16.52 | 32m | 159.93 | 10h | 225.99 | 25h |
| BQ | 7.76 | 2s | **16.72** | 46s | OOM | | OOM | |
| LEHD | 7.78 | 1s | 16.78 | 16s | OOM | | OOM | |
| POMO | **7.76** | 1s | 20.50 | 9s | OOM | | OOM | |
| ELG | 7.78 | 1s | 17.82 | 17s | OOM | | OOM | |
| INViT | 8.10 | 2s | 17.55 | 12s | 171.42 | 1.3h | 242.26 | 5h |
| L2R | 7.84 | 2s | 17.03 | 3s | **167.70** | 9.5m | **236.81** | 29.4 m |

| Method | CVRP100 (128 ins.) Obj. | Time | CVRP500 (128 ins.) Obj. | Time | CVRP50K (16 ins.) Obj. | Time | CVRP100K (16 ins.) Obj. | Time |
|---|---|---|---|---|---|---|---|---|
| HGS | 15.52 | 3.5m | 36.56 | 4h | 1081.00 | 1.3d | 2087.51 | 4d |
| BQ | 16.03 | 2s | 38.43 | 47s | OOM | | OOM | |
| LEHD | 16.16 | 1s | **38.41** | 17s | OOM | | OOM | |
| POMO | **15.80** | 1s | 48.22 | 10s | OOM | | OOM | |
| ELG | 15.94 | 1s | 39.67 | 23s | OOM | | OOM | |
| INViT | 16.77 | 2s | 43.44 | 12s | 1331.10 | 1.5h | 2683.40 | 5.8h |
| L2R | 16.12 | 2s | 39.31 | 4s | **1074.98** | 10m | **2055.14** | 31m |

Assessing performance on small-scale instances is crucial for a comprehensive evaluation, especially in terms of the training distribution (N = 100). To address this, we have conducted new experiments on both TSP and CVRP instances with sizes $N = 100$ and $N = 500$. We directly compare L2R with current representative NCO solvers. For small-scale CVRP, the capacity settings follow the setup in LEHD (Luo et al., 2023).

As shown in Table 10, our L2R achieves highly competitive performance on these small-scale instances compared to mainstream NCO solvers, such as BQ, LEHD, and ELG. Furthermore, the results also highlight L2R's superior scalability. These comprehensive experiments confirm that L2R's promising large-scale generalization ability does not come at the cost of performance degradation on small-scale instances. The new experiments validate that L2R is also a competitive solver on small-scale instances.

Moreover, as instance size increases, most NCO solvers (BQ, LEHD, POMO, ELG) fail due to Out-of-Memory (OOM) errors, while L2R's performance advantage becomes more significant. It is particularly noteworthy that on large-scale CVRP (e.g., CVRP100K), our L2R finds a higher-quality solution than the powerful heuristic HGS (which has been refined for decades), and does so with a dramatic speedup (31 minutes vs. 4 days on CVRP100K instances). This represents a significant milestone in NCO: the first constructive neural solver to achieve superior performance over HGS on large-scale CVRP instances ($N \geq 50K$) that only trained on 100-node uniform instances without any post-processing operations.

## L.3 COMPARISON WITH SIL

Table 11: Performance comparison with SIL on large-scale TSP and CVRP instances. Note that "Time" is the average inference time. Each dataset includes 16 instances.

| Method | TSP50K Obj.(Gap) | Time | TSP100K Obj.(Gap) | Time | CVRP50K Obj.(Gap) | Time | CVRP100K Obj.(Gap) | Time |
|---|---|---|---|---|---|---|---|---|
| Near-optimal | 159.93(0.00%) | 10h | 225.99(0.00%) | 25h | 1081.00(0.00%) | 4h | 2087.51(0.00%) | 6h |
| SIL greedy | 178.99 (11.92%) | 8.6m | 266.63 (17.98%) | 36.2m | 1108.28 (2.52%) | 9.2m | 2153.78 (3.17%) | 45.8m |
| L2R greedy | **167.70 (4.86%)** | **35.5s** | **236.81 (4.79%)** | **1.8m** | **1074.98 (-0.56%)** | **37s** | **2055.14 (-1.55%)** | **2m** |
| SIL PRC1000 | 167.01 (4.43%) | 1.6h | 241.68 (6.94%) | 3.0h | 1094.63(1.26%) | 1.6h | 2140.02 (2.52%) | 3.5h |
| L2R PRC1000 | **164.41 (2.80%)** | **12.4m** | **232.77 (3.00%)** | **15.5m** | **1070.49 (-0.98%)** | **33m** | **2050.91 (-1.75%)** | **1.1h** |

The SIL (Luo et al., 2024) is a powerful state-of-the-art method for large-scale routing problems. However, we do not include it in our main experiment. The primary reason for excluding SIL from the initial comparison is the fundamental difference in training strategy. SIL trains separate models for each target scale (e.g., a model trained on 10K-node instances is used to solve the 10K-node problem). The better "raw results" you mentioned are achieved by models trained directly on those large-scale instances. In contrast, our L2R is trained exclusively on small-scale instances (100 nodes) and generalizes in zero-shot to large-scale instances (up to 10M nodes). Comparing a model trained on 100 nodes against a model trained on 50K/100K nodes would be an unfair comparison, as it evaluates training capability rather than generalization capability.

To provide a fair assessment of the generalization performance, we have conducted additional experiments comparing L2R (trained on 100 nodes) with SIL (trained on 1,000 nodes). We selected the SIL-1K model because it is the smallest official model provided by SIL, yet it still benefits from a training size 10x larger than L2R's. We evaluate both L2R and SIL on large-scale TSP and CVRP instances (50K and 100K nodes). As shown in Table 11, even with a disadvantage in training data scale (100 vs. 1K), L2R significantly outperforms SIL in both solution quality (Gap) and inference speed, highlighting the efficiency of our search space reduction strategy. These results further confirm that the proposed L2R framework possesses robust generalization capabilities and practical efficiency.

## M CASE-BY-CASE RESULTS ON BENCHMARK DATASETS

### M.1 L2R VS. ALGORITHMIC METHODS

Taking the TSPLIB dataset as an example, we add the results of algorithmic methods (i.e., LKH3 and Concorde). As shown in Table 12, LKH3 and Concorde can outperform L2R on large-scale real-world instances, but they require a much longer runtime to obtain high-quality solutions. In addition, it should be emphasized that L2R can outperform all the other NCO methods with fast runtime on all comparisons. Detailed results are provided in Table 13.

Table 12: Comparison between L2R and algorithmic methods (i.e., LKH-3 and Concorde) on large-scale TSPLIB instances ($5,000 \leq N \leq 85,900$). Here, "BKS" denotes the best-known solution.

| Instance | Scale | BKS | LKH-3 | | Concorde | | L2R Greedy | | L2R PRC100 | |
|---|---|---|---|---|---|---|---|---|---|---|
| | | | Gap | Time | Gap | Time | Gap | Time | Gap | Time |
| rl5915 | 5,915 | 565,530 | 0.00% | 1.2h | 0.29% | 1.1h | 10.04% | 29s | 6.37% | 4.6m |
| rl5934 | 5,934 | 556,045 | 0.00% | 1.3h | 0.23% | 1.2h | 12.39% | 29s | 7.20% | 5.2m |
| pla7397 | 7,397 | 23,260,728 | 0.00% | 11.8h | 0.22% | 1.0h | 7.22 % | 36s | 3.25% | 5.2m |
| rl11849 | 11,849 | 923,288 | 0.00% | 4.8h | 0.66% | 1.1h | 6.96% | 58s | 4.49% | 5.4m |
| usa13509 | 13,509 | 19,982,859 | 0.00% | 7.5h | 0.25% | 1.2h | 6.99% | 1.1m | 4.89% | 5.4m |
| brd14051 | 14,051 | 469,385 | 0.00% | 9.1h | 0.39% | 1.1h | 6.52% | 1.1m | 4.16% | 5.5m |
| d15112 | 15,112 | 1,573,084 | 0.00% | 11.4h | 0.26% | 1.1h | 7.55 % | 1.2m | 5.35% | 5.5m |
| d18512 | 18,512 | 645,238 | 0.00% | 13.1h | 0.39% | 1.1h | 5.39% | 1.5m | 3.37% | 5.8m |
| pla33810 | 33,810 | 66,048,945 | 0.00% | 18h | N/A | | 6.37% | 2.8m | 4.35% | 7.4m |
| pla85900 | 85,900 | 142,382,641 | 0.00% | 23h | N/A | | 4.21% | 7.7m | 2.32% | 12.9m |
| Avg.gap & Time | | | 0.00% | 10.1h | 0.34%† | 1.1h | 7.36% | 1.8m | 4.58% | 6.3m |

† The pla33810 and pla85900 instances are skipped due to infeasible solution generation.

## M.2 L2R vs. Classical Neural Solvers

We evaluate the generalization performance on instances from CVRPLIB Set-XXL (Arnold et al., 2019) and TSPLIB (Reinelt, 1991). Detailed results for large-scale instances are provided in Table 13. The results demonstrate that L2R consistently outperforms other models across instances of varying scales, demonstrating its strong applicability in real-world scenarios.

Table 13: The detailed results on large-scale TSPLIB (Reinelt, 1991) instances ($5,000 \leq N \leq 85,900$) and CVRPLIB (Arnold et al., 2019) instances ($3,000 \leq N \leq 30,000$).

| Instance | Scale | GLOP | BQ bs16 | LEHD greedy | LEHD RRC1000 | POMO aug×8 | ELG aug×8 | INViT greedy | L2R greedy | L2R PRC100 |
|---|---|---|---|---|---|---|---|---|---|---|
| rl5915 | 5,915 | 11.56% | 19.58% | 24.17% | 12.38% | OOM | OOM | 14.02% | 10.04% | **6.37%** |
| rl5934 | 5,934 | 11.32% | 24.53% | 24.11% | 11.94% | OOM | OOM | 12.91% | 12.39% | **7.20%** |
| pla7397 | 7,397 | 6.38% | 47.63% | 40.94% | 15.31% | OOM | OOM | 9.45% | 7.22% | **3.25%** |
| rl11849 | 11,849 | 9.55% | OOM | 38.04% | 18.19% | OOM | OOM | 12.71% | 6.96% | **4.49%** |
| usa13509 | 13,509 | 5.81% | OOM | 71.10% | 31.37% | OOM | OOM | 13.44% | 6.99% | **4.89%** |
| brd14051 | 14,051 | 4.87% | OOM | 41.22% | 21.19% | OOM | OOM | 9.31% | 6.52% | **4.16%** |
| d15112 | 15,112 | 4.90% | OOM | 35.82% | 18.83% | OOM | OOM | 7.58% | 7.55% | **5.35%** |
| d18512 | 18,512 | 4.99% | OOM | OOM | OOM | OOM | OOM | 6.62% | 5.39% | **3.37%** |
| pla33810 | 33,810 | 9.53% | OOM | OOM | OOM | OOM | OOM | 7.04% | 6.37% | **4.35%** |
| pla85900 | 85,900 | 7.97% | OOM | OOM | OOM | OOM | OOM | 7.21% | 4.21% | **2.32%** |
| Solved# | | 10/10 | 3/10 | 7/10 | 7/10 | 0/10 | 0/10 | 10/10 | 10/10 | 10/10 |
| Avg.gap | | 7.69% | 30.58%† | 39.34%† | 18.46%† | − | − | 10.03% | 7.36% | **4.58%** |
| Instance | Scale | GLOP-LKH3 | BQ bs16 | LEHD greedy | LEHD RRC1000 | POMO aug×8 | ELG aug×8 | INViT greedy | L2R greedy | L2R PRC100 |
| Leuven1 | 3,000 | 14.95% | 15.39% | 16.60% | 10.71% | 460.32% | 12.12% | 13.71% | 11.49% | **5.67%** |
| Leuven2 | 4,000 | 16.54% | 25.69% | 34.85% | 21.22% | 202.17% | 21.52% | 26.08% | 12.73% | **11.82%** |
| Antwerp1 | 6,000 | 19.08% | 13.64% | 14.66% | 8.91% | OOM | OOM | 15.40% | 11.04% | **5.93%** |
| Antwerp2 | 7,000 | 17.72% | 26.09% | 22.77% | 15.42% | OOM | OOM | 27.75% | 12.95% | **11.07%** |
| Ghent1 | 10,000 | 18.28% | OOM | 27.23% | 17.28% | OOM | OOM | 15.87% | 10.15% | **6.02%** |
| Ghent2 | 11,000 | 16.45% | OOM | 38.36% | 25.77% | OOM | OOM | 30.78% | 11.29% | **11.01%** |
| Brussels1 | 15,000 | 26.17% | OOM | OOM | OOM | OOM | OOM | 18.09% | 12.65% | **8.03%** |
| Brussels2 | 16,000 | 17.56% | OOM | OOM | OOM | OOM | OOM | 32.08% | 12.30% | **11.79%** |
| Flanders1 | 20,000 | 24.02% | OOM | OOM | OOM | OOM | OOM | 23.41% | 7.85% | **6.48%** |
| Flanders2 | 30,000 | 25.42% | OOM | OOM | OOM | OOM | OOM | 39.60% | 13.09% | **12.70%** |
| Solved# | | 10/10 | 4/10 | 6/10 | 6/10 | 2/10 | 2/10 | 10/10 | 10/10 | 10/10 |
| Avg.gap | | 19.62% | 20.20%† | 25.75%† | 16.55%† | 331.24%† | 16.82%† | 24.28% | 11.55% | **9.34%** |

† Some instances are skipped due to the OOM issue.

## M.3 Solving Very Large-Scale TSP Instances

We further evaluate L2R on challenging large-scale TSP instances from the DIMACS challenge and include HDR (Fu et al., 2023) in our comparison. As shown in Table 14, L2R can achieve strong

performance on TSP instances with up to **10M** nodes, which significantly outperforms other NCO methods. While LKH-3 and HDR obtain near-optimal solutions for these instances, they require careful, expert-designed algorithmic components (e.g., the destroy-and-repair operators in HDR).

Our experimental results in Table 14 on the DIMACS benchmark demonstrate that L2R, as a purely learning-based neural solver, can consistently produce high-quality solutions for unprecedented problem instances with up to 10 million nodes. Notably, this capability is achieved through generalization from the small-scale 100-node training instances with uniform distributions, which represents a significant advancement in NCO.

Table 14: Experimental results tested on the 8th DIMACS challenge E series TSP instances (10K $\leq N \leq$ 10M). The results of LKH-3 and HDR are taken from the original HDR paper (Fu et al., 2023).

| Instance | LKH-3* | | HDR* | | LEHD-Greedy | | INViT-Greedy | | L2R-Greedy | | L2R-PRC1000 | |
|---|---|---|---|---|---|---|---|---|---|---|---|---|
| | Gap | Time | Gap | Time | Gap | Time | Gap | Time | Gap | Time | Gap | Time |
| E10k.0 | 0.004% | 4.6h | 0.003% | 5.6h | 24.628% | 12m | 6.636% | 3.8m | 4.562% | 43s | **2.569%** | 45.9m |
| E10k.1 | 0.000% | 4.7h | 0.002% | 5.6h | 26.506% | 12m | 7.080% | 3.8m | 4.591% | 44s | **2.570%** | 46.1m |
| E10k.2 | 0.004% | 4.8h | 0.003% | 5.6h | 24.743% | 12m | 7.382% | 3.8m | 4.766% | 45s | **2.275%** | 45.3m |
| E31k.0 | 0.009% | 17.6h | 0.014% | 17.6h | OOM | | 6.970% | 29.2m | 4.822% | 2.4m | **2.716%** | 49.6m |
| E31k.1 | 0.004% | 17.6h | 0.013% | 17.6h | OOM | | 7.222% | 29.1m | 4.735% | 2.4m | **2.888%** | 48.0m |
| E100k.0 | 0.064% | 55.6h | 0.024% | 55.6h | OOM | | N/A | | 4.711% | 8.2m | **2.967%** | 54.8m |
| E100k.1 | 0.060% | 55.6h | 0.025% | 55.6h | OOM | | N/A | | 4.976% | 8.5m | **3.012%** | 56.1m |
| E316k.0 | 0.109% | 175.7h | 0.026% | 175.7h | OOM | | N/A | | 4.821% | 27.1m | **3.343%** | 1.2h |
| E1M.0 | 0.131% | 278.1h | 0.117% | 277.8h | OOM | | N/A | | **5.022%** | 2.0h | – | – |
| E3M.0 | 0.155% | 279.3h | 0.119% | 277.8h | OOM | | N/A | | **5.047%** | 12h | – | – |
| E10M.0 | 0.169% | 560.4h | 0.136% | 555.7h | OOM | | N/A | | **5.046%** | 80h | – | – |

[†] Some instances are skipped due to the OOM issue.

## N  ADDITIONAL ANALYSES

### N.1  L2R VS. LEADING HEURISTICS

Table 15: Comparison between L2R and classic heuristics on TSP and CVRP instances. "Avg.time" indicates the average time per instance. Note that HGS and LKH3 are running under the given 100 GB of memory.

| Method | TSP1K | | TSP5K | | TSP10K | | TSP1M | |
| --- | --- | --- | --- | --- | --- | --- | --- | --- |
| | Obj. | Avg.time | Obj. | Avg.time | Obj. | Avg.time | Obj. | Avg.time |
| HGS | **23.2** | 5m | 54.3 | 5m | 76.7 | 5m | OOM | |
| L2R greedy | 24.2 | **0.05s** | **53.4** | **1.8s** | **75.2** | **4.1s** | 748.2 | **1.2h** |
| LKH3 | **23.1** | 1.7m | **51.0** | 12m | **71.8** | 33m | **714.0** | 4.8h |
| L2R greedy | 24.2 | **0.05s** | 53.4 | **1.8s** | 75.2 | **4.1s** | 748.2 | **1.2h** |
| Method | CVRP1K | | CVRP5K | | CVRP10K | | CVRP1M | |
| | Obj. | Avg.time | Obj. | Avg.time | Obj. | Avg.time | Obj. | Avg.time |
| HGS | **41.2** | 5m | **126.2** | 5m | **227.2** | 5m | OOM | |
| L2R greedy | 45.8 | **0.1s** | 136.0 | **0.5s** | 236.7 | **4.4s** | 17231.9 | **1.4h** |
| LKH3 | 46.4 | 6.2s | 175.7 | 2.5m | 290.6 | 9.4m | N/A | |
| L2R greedy | **45.8** | **0.1s** | **136.0** | **0.5s** | **236.7** | **4.4s** | **17231.9** | **1.4h** |

We evaluate our method alongside current leading heuristics (i.e., LKH3 and HGS), and the results in Table 15 demonstrate that each approach has distinct strengths, which represent complementary advantages rather than unilateral dominance. Detailed comparisons are as follows:

**L2R vs. LKH3:**  While LKH3 is the undisputed leader for TSP, its performance on CVRP is significantly weaker. As shown in Table 15, our L2R can consistently outperform LKH3 on CVRP instances with different scales, which demonstrates the strength of our learning-based approach in this domain.

**L2R vs. HGS:**  HGS can obtain superior performance on CVRP instances with N≤10K. However, population-based HGS faces critical scalability limitations, which fail to solve million-node instances due to OOM errors. In contrast, our proposed L2R overcomes these limitations through its learning-based search space reduction, which enables effective optimization of massive-scale CVRPs. In addition, while HGS can be technically adapted to solve TSP by setting vehicle capacity to infinity, our results demonstrate that compared with HGS, L2R obtains better performance on large-scale TSP instances. This reinforces that the strength of HGS is highly focused on CVRP, which highlights its narrow specialization compared to our more generalizable approach.

The above comparison shows that our work is highly competitive and even superior in specific, challenging scenarios that are highly relevant to the field (i.e., solving large-scale CVRPs) compared with LKH3 and HGS. Notably, our core contribution is pushing the frontier of NCO in terms of generalization and scalability. This research objective is fundamentally distinct from outperforming long-established and problem-specific heuristics on its specialized problem. Furthermore, direct comparisons reveal L2R as a strong contender that outperforms LKH-3 on CVRP while demonstrating massive-scale solving capability that HGS cannot match.

### N.2  L2R VS. D-SSR

To validate the effectiveness of our proposed L2R, we conduct a comparative experiment against a dynamic distance-based SSR model (i.e., D-SSR), ensuring identical experimental settings. We systematically analyze the advantages of L2R, mainly including: (1) overall performance on real-world instances, (2) optimality gap and optimality ratio per instance, (3) impacts of different candidate node selection strategies, and (4) solution visualizations comparing the two approaches.

### N.2.1 Overall Performance on Real-world Instances

According to the results in Table 16, compared with traditional D-SSR, our proposed L2R can obtain better generalization performance in large-scale TSPLIB and CVRPLIB instances. Notably, this advantage becomes more pronounced when solving complex CVRP instances, where L2R significantly improves the final solution quality compared with D-SSR.

Table 16: Optimality gap comparison between two different candidate node selection approaches. Here, D-SSR denotes distance-based search space reduction.

| | TSPLIB | | | CVRPLIB-XXL | | |
|---|---|---|---|---|---|---|
| Method | $1K \leq N \leq 5K$ (23 instances) | $5K < N \leq 100K$ (10 instances) | All (33 instances) | $3K \leq N \leq 7K$ (4 instances) | $7K < N \leq 30K$ (6 instances) | All (10 instances) |
| D-SSR greedy | 9.65% | 7.47% | 8.98% | 14.38% | 12.19% | 13.06% |
| L2R greedy | **9.16%** | **7.36%** | **8.61%** | **12.05%** | **11.22%** | **11.55%** |

### N.2.2 Optimality Gap vs. Optimality Ratio

We then measure the optimality gap and optimality ratio per instance. The optimality ratio is defined as the proportion of nodes where the optimal next-visit node is located within the $k$ candidate nodes of the current node. As demonstrated in Table 17, L2R achieves superior solution quality over D-SSR on real-world datasets. Although L2R exhibits only marginal improvement in the optimality ratio of next-visit node selection, the cumulative effect of over-pruning during the solution construction process leads to significant deviations from the optimal route in subsequent node selections, ultimately compromising the overall solution quality obtained by D-SSR.

Table 17: Comparison of different reduction approaches on TSPLIB and CVRPLIB instances. All results are obtained with greedy decoding. "#Accuracy" measures the number of construction steps where the optimal next-visit node is contained within the top-$k$ candidates selected by either learning-based or distance-based reduction methods. "Optimality Ratio" measures the proportion of nodes where the optimal next-visit node is located within the $k$ candidate nodes of the current node. (i.e., #Accuracy * 100% / $N$).

| | Instance | Scale | Gap ↓ | | | #Accuracy (Optimality Ratio) ↑ | | |
|---|---|---|---|---|---|---|---|---|
| | | | D-SSR | L2R | ΔGap ↓ | D-SSR | L2R | Δ#Accuracy ↑ |
| TSPLIB | pcb1173 | 1173 | 5.17% | **4.02%** | −1.15% | 1062 (90.54%) | **1067 (90.96%)** | +5 |
| | rl1304 | 1304 | 9.88% | **8.14%** | −1.74% | 1225 (93.94%) | **1234 (94.63%)** | +9 |
| | rl1323 | 1323 | 16.44% | **9.34%** | −7.10% | 1242 (93.88%) | **1252 (94.63%)** | +10 |
| | u1817 | 1817 | 17.60% | **9.54%** | −8.06% | 1614 (88.83%) | **1630 (89.71%)** | +16 |
| | d2103 | 2103 | 20.50% | **8.09%** | −12.41% | 1988 (94.53%) | **2000 (95.10%)** | +12 |
| | pr2392 | 2392 | 10.72% | **8.08%** | −2.64% | 2105 (88.00%) | **2129 (89.01%)** | +24 |
| | fnl4461 | 4461 | 6.85% | **4.22%** | −2.63% | 3871 (86.77%) | **3896 (87.34%)** | +25 |
| | rl11849 | 11849 | 8.13% | **6.96%** | −1.17% | 10851 (91.58%) | **10864 (91.69%)** | +13 |
| | usa13509 | 13509 | 8.97% | **6.99%** | −1.98% | 11886 (87.99%) | **11947 (88.44%)** | +61 |
| | pla33810 | 33810 | 7.57% | **6.37%** | −1.20% | 30263 (89.51%) | **30463 (90.10%)** | +200 |
| CVRPLIB | Leuven1 | 3000 | 12.67% | **11.49%** | −1.18% | 2284 (76.13%) | **2294 (76.47%)** | +10 |
| | Leuven2 | 4000 | 15.71% | **12.73%** | −2.98% | 3287 (82.18%) | **3299 (82.48%)** | +12 |
| | Antwerp1 | 6000 | 13.12% | **11.04%** | −2.08% | 4573 (76.22%) | **4597 (76.62%)** | +24 |
| | Antwerp2 | 7000 | 16.00% | **12.95%** | −3.05% | 5611 (80.16%) | **5638 (80.54%)** | +27 |
| | Ghent1 | 10000 | 11.68% | **10.15%** | −1.53% | 7658 (76.58%) | **7665 (76.65%)** | +7 |
| | Ghent2 | 11000 | 13.55% | **11.29%** | −2.26% | 9024 (82.04%) | **9045 (82.23%)** | +21 |
| | Brussels2 | 16000 | 13.83% | **12.30%** | −1.53% | 13099 (81.87%) | **13109 (81.93%)** | +10 |

### N.2.3 Impacts of Candidate Node Selection Strategy

Figure 7 presents further analysis of candidate node selection across different reduction approaches at identical construction steps. The results in Figure 7a−Figure 7b indicate that while both D-SSR and L2R models successfully include the majority of optimal nodes within their generated neighborhoods, D-SSR exhibits significantly degraded solution quality. This degradation can be primarily attributed to its failure to include a critical minority of nodes within its candidate set. Consequently, the local

construction model struggles to effectively select the optimal next-visit node. These critical nodes are exemplified by those to lead the sub-problem into a suboptimal state (see Figure 7c−Figure 7d) or manifest as anomalous single-node visits (see Figure 7e−Figure 7f). In contrast, the enhanced candidate selection capability of L2R allows it to capture more accurate candidate nodes, which facilitates better sequential decision-making and ultimately yields superior overall performance.

### N.2.4 SOLUTION VISUALIZATIONS

To further support our argument, we provide additional solution visualizations comparing L2R and D-SSR. As illustrated in Figure 8, L2R tends to follow the optimal route more closely, but the cumulative effect of over-pruning during the solution construction of D-SSR leads to significant deviations in subsequent node selections. These deviations are a direct consequence of critical nodes not being adequately included in the candidate set, ultimately resulting in a significant degradation of D-SSR's solution quality compared to L2R.

### N.3 COMPLEXITY ANALYSIS

We present the time and space complexity of three different constructive models: L2R with learning-based SSR, INViT with distance-based SSR, and LEHD without SSR. For the L2R method, we decompose the computational complexity into three key components: (1) the complexity of search space reduction (denoted as STEP1), (2) the complexity of local solution construction (denoted as STEP2), and (3) the overall framework complexity.

Table 18: Complexity analysis between L2R and existing works. Note that $N$ and $k$ denote the scale and the number of local neighbors, respectively. All results are obtained using greedy decoding. "Time" denotes the total inference time in the construction process, and "$\#m$" is the number of tested instances.

| Method | Time Complexity | Space Complexity | TSP5K(#16) Gap | TSP5K(#16) Time | TSP10K(#16) Gap | TSP10K(#16) Time | CVRP5K(#100) Gap | CVRP5K(#100) Time | CVRP10K(#16) Gap | CVRP10K(#16) Time |
|---|---|---|---|---|---|---|---|---|---|---|
| LEHD | $\mathcal{O}\left(N^3\right)$ | $\mathcal{O}\left(N^2\right)$ | 15.46% | 24m | 27.24% | 3.1h | 9.51% | 2.3h | 13.25% | 3.2h |
| INViT-3V† | $\mathcal{O}\left(N(k_1^2+k_2^2+k_3^2+k_1)\right)$ | $\mathcal{O}\left(k_1^2+k_2^2+k_3^2+k_1\right)$ | 6.84% | 3.4m | 7.09% | 9.3m | 15.87% | 11.6m | 15.53% | 19.2m |
| L2R-STEP1‡ | $\mathcal{O}\left(N^2\right)$ | $\mathcal{O}\left(N\right)$ | − | 8.0s | − | 21.4s | − | 17.5s | − | 18.1s |
| L2R-STEP2‡ | $\mathcal{O}\left(Nk^2\right)$ | $\mathcal{O}\left(k^2\right)$ | − | 21.5s | − | 42.6s | − | 34.1s | − | 52.1s |
| L2R | $\mathcal{O}\left(N^2+Nk^2\right)$ | $\mathcal{O}\left(N+k^2\right)$ | 4.69% | 29.5s | 4.82% | 1.1m | 7.74% | 51.6s | 4.18% | 1.2m |

† For INViT, $k_1, k_2, k_3$ represent different view, respectively.
‡ L2R-STEP1 and L2R-STEP2 represent the learning-based reduction model and local solution construction model, respectively.

Our comparative results in Table 18 demonstrate that L2R-STEP2 achieves significantly lower computational complexity than both LEHD and INViT. This efficiency advantage translates into substantially reduced runtime compared to existing representative models. Since distance-based SSR may compromise solution optimality, we introduce a learning-based reduction model to refine candidate node selection, thereby mitigating its negative impact. Although the reduction model incurs linear time complexity growth with problem size, experimental results demonstrate that the additional computational cost is justified, and L2R can perform better than D-SSR in solution quality. For more results and analyses, please see Appendix N.2.

### N.4 TRAINING AND INFERENCE COSTS

Table 19: Details of L2R training for different problems.

| Problem | (#) Params | Device | Epochs | Batch size | Training Time |
|---|---|---|---|---|---|
| TSP | 1.19M | A RTX 3090 | 100 | 180 | 6 days |
| CVRP | 1.17M | (24GB Memory) | 100 | 60 | 7 days |

We have conducted a detailed analysis of training and inference costs. As shown in Table 19, our L2R model requires approximately 6–7 days of training on a single NVIDIA RTX 3090 GPU. We believe this computational cost is reasonable and competitive with other learning-based methods. For example, mainstream NCO approaches (e.g., POMO (Kwon et al., 2020), LEHD (Luo et al.,

Table 20: Details of L2R Inference for different problems. "Time" indicates the total runtime.

| Problem | 1K | | | 5K | | | 10K | | |
|---|---|---|---|---|---|---|---|---|---|
| | Batch size | Time | Avg.Memory | Batch size | Time | Avg.Memory | Batch size | Time | Avg.Memory |
| TSP | 128 | 6s | 3.39MB | 16 | 30s | 17.65MB | 16 | 64s | 33.80MB |
| CVRP | 100 | 8s | 3.44MB | 100 | 52s | 16.69MB | 16 | 71s | 33.89MB |

2023), and BQ (Drakulic et al., 2023)) typically also require over 7 days of training to achieve full convergence. This demonstrates that L2R achieves strong performance without introducing excessive computational overhead. Meanwhile, the inference costs are provided in Table 20. We report the inference time and average memory usage for solving batches of 1K, 5K, and 10K instances. The results indicate modest resource requirements and efficient scaling with problem size. For example, solving TSP10K instances takes slightly over a minute, with each instance consuming less than 34MB of memory. This highlights the practicality of L2R for large-scale applications.

### N.5 Incorporating Supervised Signals

To validate the effects of our adopted joint training, we have conducted a new experiment that incorporates supervised signals into the loss function of the reduction model. Specifically, using TSP as a test case, we utilize a dataset of 1 million near-optimal solutions for TSP100 instances (provided by LEHD (Luo et al., 2023)) as an oracle. We then incorporate a supervised penalty term into the reward function for the reduction model ($\theta_R$), which penalizes the model whenever the oracle's next node is not included in the predicted candidate set. Following the formulation in PIP (Bi et al., 2024), the penalty term $\mathcal{J}_{OUT}$ is calculated as the number of penalized nodes, and the modified loss function is:

$$\nabla_{\boldsymbol{\theta}_R} \mathcal{L}_{R_{SS}}(\boldsymbol{\theta}_R) = \mathbb{E}_{o(\tau|S,\boldsymbol{\theta}_R)} \left[ (\mathcal{R}(\pi|S,\boldsymbol{\theta}_L) - \lambda * \mathcal{J}_{OUT} - b(S)) \nabla_{\boldsymbol{\theta}_R} \log o(\tau|S) \right], \quad (39)$$

We have tested two supervised versions with penalty coefficients $\lambda = 1.0$ and $\lambda = 0.1$.

Table 21: Comparison of different reduction approaches on TSPLIB instances. Here, D-SSR denotes distance-based search space reduction, and SS indicates the addition of supervised signals to the reduction model in training. "Optimality Ratio" measures the proportion of nodes where the optimal next-visit node is located within the $k$ candidate nodes of the current node.

| Instance | Scale | Optimality Ratio ↑ | | | | Gap ↓ | | | |
|---|---|---|---|---|---|---|---|---|---|
| | | D-SSR | L2R | L2R-SS ($\lambda = 1$) | L2R-SS ($\lambda = 0.1$) | D-SSR | L2R | L2R-SS ($\lambda = 1$) | L2R-SS ($\lambda = 0.1$) |
| pcb1173 | 1173 | 90.54% | **90.96%** | 90.37% | 90.45% | 5.17% | **4.02%** | 4.76% | 5.25% |
| rl1304 | 1304 | 93.94% | 94.63% | **95.09%** | **95.09%** | 9.88% | 8.14% | 7.82% | **7.50%** |
| rl1323 | 1323 | 93.88% | 94.63% | **95.01%** | 94.86% | 16.44% | **9.34%** | 17.81% | 19.77% |
| u1817 | 1817 | 88.83% | 89.71% | 89.49% | **90.59%** | 17.60% | **9.54%** | 12.22% | 18.93% |
| d2103 | 2103 | 94.53% | 95.10% | 95.63% | **95.67%** | 20.50% | **8.09%** | 17.78% | 18.75% |
| pr2392 | 2392 | 88.00% | 89.01% | 89.05% | **90.34%** | 10.72% | 8.08% | 8.26% | **7.50%** |
| fnl4461 | 4461 | 86.77% | **87.34%** | 87.13% | 87.20% | 6.85% | **4.22%** | 6.00% | 5.75% |
| rl11849 | 11849 | 91.58% | 91.69% | **92.06%** | 91.97% | 8.13% | **6.96%** | 8.00% | 7.71% |
| usa13509 | 13509 | 87.99% | 88.44% | 87.97% | **88.74%** | 8.97% | **6.99%** | 9.80% | 8.47% |
| pla33810 | 33810 | 89.51% | **90.10%** | 89.63% | 89.83% | 7.57% | 6.37% | **5.82%** | 6.44% |
| Average | | 90.56% | 91.16% | 91.14% | **91.47%** | 11.18% | **7.18%** | 9.83% | 10.61% |

As shown in Table 21, compared to the original L2R, incorporating supervised signals can indeed lead to a higher average optimality ratio (i.e., the proportion of construction steps where the optimal next-visit node resides within the $k$-nearest candidates), which confirms your hypothesis. However, this improvement in candidate selection does not consistently yield better final solution quality (Optimality Gap). On average, the original L2R significantly outperforms both supervised variants (L2R-SS). We speculate that although supervised signals encourage the reduction model to generate candidate sets that better align with the oracle's choices, the local construction model may not fully capitalize on this move for subsequent steps. A theoretically optimal move can become a local trap if the lower policy cannot maintain that optimality. In contrast, the joint RL framework fosters co-adaptation: the reduction model learns to provide candidates that are compatible with the construction model's behavior, prioritizing feasible high-reward trajectories over rigid adherence to an oracle.

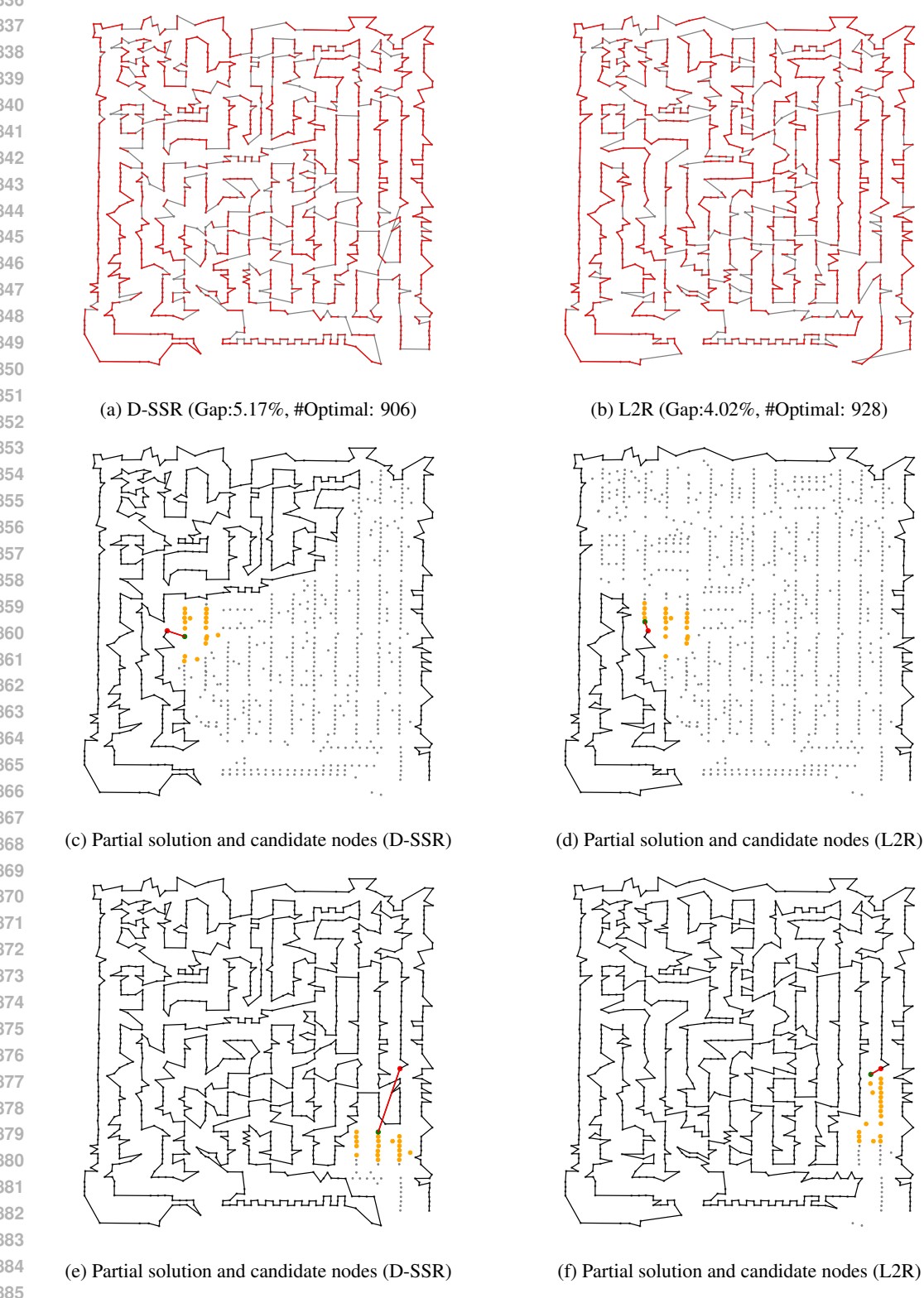

(a) D-SSR (Gap:5.17%, #Optimal: 906)

(b) L2R (Gap:4.02%, #Optimal: 928)

(c) Partial solution and candidate nodes (D-SSR)

(d) Partial solution and candidate nodes (L2R)

(e) Partial solution and candidate nodes (D-SSR)

(f) Partial solution and candidate nodes (L2R)

Figure 7: **The distributions of node selection strategy between different node selection approaches in TSPLIB, using pcb1173 as an example.** The solutions are all generated by greedy decoding. In (a)-(b), the red connections and "#Optimal" indicate optimal node connections and the number of optimal connections, respectively. For the remaining figures, the red, green, and orange nodes indicate the current node, the next visiting node, and candidate nodes, respectively.

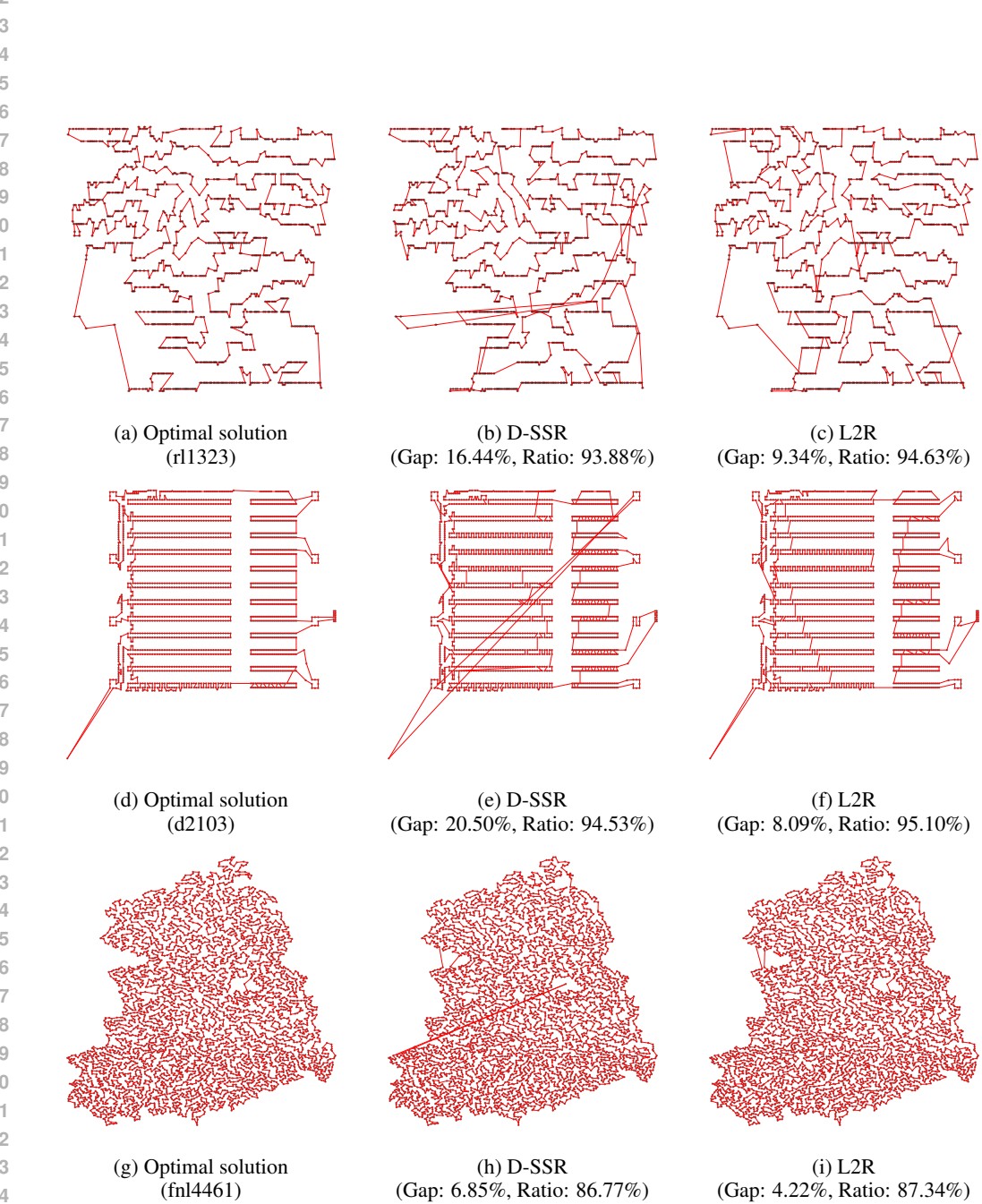

(a) Optimal solution
(rl1323)

(b) D-SSR
(Gap: 16.44%, Ratio: 93.88%)

(c) L2R
(Gap: 9.34%, Ratio: 94.63%)

(d) Optimal solution
(d2103)

(e) D-SSR
(Gap: 20.50%, Ratio: 94.53%)

(f) L2R
(Gap: 8.09%, Ratio: 95.10%)

(g) Optimal solution
(fnl4461)

(h) D-SSR
(Gap: 6.85%, Ratio: 86.77%)

(i) L2R
(Gap: 4.22%, Ratio: 87.34%)

Figure 8: **The solution visualizations of TSPLIB instances with different scales.** The solutions are all generated by greedy decoding. "Ratio" represents the optimality ratio, i.e., the proportion of nodes where the optimal next-visit node is located within the candidate nodes of the current node.

## O ABLATION STUDY

In this section, we conduct a detailed ablation study and analysis to demonstrate the effectiveness and robustness of L2R. Please note that, unless stated otherwise, the results presented in the ablation study are obtained with greedy decoding, and the performance of TSP instances is used as the primary criterion for evaluation.

### O.1 L2R VS. DISTANCE-BASED SSR WITH LARGER SEARCH SPACE

We conduct comparative experiments between four D-SSR models with varying search space sizes ($k \in \{20, 50, 75, 100\}$) and our proposed L2R model ($k = 20$) on TSP100K instances. We report the results obtained from greedy decoding and a PRC improvement procedure with different iterations.

Table 22: Comparison between learning-based small space and distance-based large space on TSP100K instances with uniform distribution.

| Method | Greedy | PRC100 | PRC500 | PRC1000 |
|---|---|---|---|---|
| D-SSR ($k = 20$) | 5.09% | 4.02% | 3.49% | 3.26% |
| D-SSR ($k = 50$) | 6.62% | 4.98% | 4.23% | 3.92% |
| D-SSR ($k = 75$) | 7.33% | 5.05% | 4.20% | 3.87% |
| D-SSR ($k = 100$) | 5.74% | 4.91% | 4.20% | 3.87% |
| L2R ($k = 20$) | **4.79%** | **3.70%** | **3.21%** | **3.00%** |

The results in Table 22 show that our L2R model consistently achieves superior performance compared to distance-based SSR models, despite operating with a significantly smaller search space (L2R with $k = 20$ vs. D-SSR with $k \geq 20$). Furthermore, when combined with PRC, L2R also maintains this performance advantage over all D-SSR variants.

### O.2 EFFECTS OF ADAPTATION BIAS IN COMPATIBILITY

Table 23: Comparison between different component settings of the compatibility module. Here, $a_{t-1,i}^R$ and $a_{t-1,i}^L$ indicate the adaptation bias adopted in Equation (6) and Equation (8), respectively.

| $a_{t-1,i}^R$ | $a_{t-1,i}^L$ | TSP1K | TSP5K | TSP10K | TSP50K | TSP100K |
|---|---|---|---|---|---|---|
| $\times$ | $\times$ | 53.91% | 120.91% | 157.46% | 255.16% | 305.95% |
| $\times$ | $\checkmark$ | 56.64% | 129.57% | 169.68% | 279.63% | 335.51% |
| $\checkmark$ | $\times$ | 5.22% | 5.42% | 5.66% | 5.72% | 5.68% |
| $\checkmark$ | $\checkmark$ | **4.49%** | **4.69%** | **4.82%** | **4.86%** | **4.79%** |

To evaluate the effectiveness of two adaptation biases, we conduct an ablation study with different component configurations. The results in Table 23 demonstrate that accurately identifying candidate nodes within a large search space remains challenging when relying solely on a lightweight network architecture. Unlike existing distance-based reduction methods, our approach addresses the trade-off between learning difficulty and solution quality by incorporating a distance-assisted reduction model to enhance neighborhood selection performance. Both $a_{t-1,i}^R$ and $a_{t-1,i}^L$ contribute significantly to the promising model performance.

### O.3 COMPARISON BETWEEN DIFFERENT ATTENTION MECHANISMS

Following Zhou et al. (2024a), we employ AAFM to implement $\text{Attention}(\cdot)$, enhancing geometric pattern recognition for routing problems (see Appendix F for implementation details). To evaluate the effectiveness of AAFM compared to the standard MHA in search space reduction and local construction, we train a variant of our L2R, denoted as L2R-MHA, which replaces AAFM with MHA. Furthermore, we introduce an adaptation version of L2R-MHA, termed L2R-MHA*, which incorporates an adaptation bias. Due to the special design of MHA, the integration of the adaptation

bias with self-attention is formulated as follows:

$$\text{Attention}^*(Q, K, V, A) = \text{softmax}\left(\frac{QK^{\text{T}}}{\sqrt{d_k}} + A\right) V, \tag{40}$$

where $A = \{\mathbf{a}_{ij}\}$ denotes the pair-wise adaptation bias computed by our adaptation function $f(N, d_{ij})$ in Appendix E. L2R-MHA and L2R-MHA* are trained in exactly the same training settings. The only difference between the three models is the attention mechanism.

Table 24: Comparison of different attention mechanisms on TSP instances with different scales. Here, "Time" denotes the total inference time.

| Method | TSP1K | | TSP5K | | TSP10K | | TSP50K | | TSP100K | |
|---|---|---|---|---|---|---|---|---|---|---|
| | Gap | Time | Gap | Time | Gap | Time | Gap | Time | Gap | Time |
| L2R-MHA | 6.90% | 6.7s | 7.36% | 29.6s | 7.54% | 1.1m | 7.79% | 9.6m | 7.72% | 31.7m |
| L2R-MHA* | 5.36% | 7.0s | 5.57% | 31.0s | 5.83% | 1.1m | 5.92% | 9.8m | 6.23% | 32.2m |
| L2R | **4.49%** | 6.5s | **4.69%** | 29.5s | **4.82%** | 1.1m | **4.86%** | 9.5m | **4.79%** | 29.4m |

As shown in Table 24, the L2R-MHA and L2R-MHA* models demonstrate robust large-scale generalization capabilities, further validating the effectiveness of the proposed L2R framework. Notably, substituting the MHA mechanism with AAFM yields additional performance improvements with shorter runtime.

### O.4 COMPARISON OF DIFFERENT INITIAL SOLUTIONS

Table 25: The impact of PRC under different initial solution generation methods. Note that "NN-Initial PRCm" denotes generating initial solutions using the nearest neighbor method and employing L2R as the repair operator in the PRC process.

| Method | TSP5K | | TSP10K | | TSP50K | | TSP100K | |
|---|---|---|---|---|---|---|---|---|
| | Gap | Avg.time | Gap | Avg.time | Gap | Avg.time | Gap | Avg.time |
| Nearest Neighbor(NN) | 24.45% | 0.06s | 23.92% | 0.2s | 22.99% | 2.7s | 22.83% | 11s |
| NN-Initial PRC100 | 10.42% | 18.1s | 12.36% | 22.8s | 13.85% | 1.2m | 15.09% | 1.4m |
| NN-Initial PRC500 | 9.77% | 1.6m | 11.61% | 1.9m | 13.15% | 6.0m | 13.93% | 7.3m |
| NN-Initial PRC1000 | 9.57% | 3.1m | 11.49% | 3.7m | 12.95% | 11.9m | 13.56% | 14.2m |
| L2R Greedy | 4.69% | 1.8s | 4.82% | 4.1s | 4.86% | 35.5s | 4.79% | 1.8m |
| L2R-Initial PRC100 | 2.82% | 20.1s | 3.03% | 26.3s | 3.27% | 1.8m | 3.70% | 3.2m |
| L2R-Initial PRC500 | 2.50% | 1.6m | 2.72% | 2.0m | 2.92% | 6.6m | 3.21% | 8.8m |
| L2R-Initial PRC1000 | **2.40%** | 3.1m | **2.62%** | 3.8m | **2.80%** | 12.4m | **3.00%** | 15.5m |

To explicitly quantify the contribution of our core L2R model, we conducted a new ablation study to isolate the impact of the initial solution on the final post-processing performance. We compare two scenarios: 1) the initial solution is generated using the commonly adopted nearest neighbor technique from classical heuristics (denoted as NN-Initial), and 2) the initial solution is generated by greedy decoding of our L2R model (denoted as L2R-Initial). Note that all results using PRC are obtained by employing L2R as the repair operator.

As shown in Table 25, after applying the same PRC refinement, L2R-Initial consistently outperforms NN-Initial across problem instances of varying scales. This ablation study confirms that our core L2R model plays a critical role by generating high-quality initial solutions, which are the crucial drivers of the final strong results.

### O.5 EFFECTS OF STATIC REDUCTION

To verify the effect of static reduction on different methods, we have performed two new experiments. Since strong baselines like BQ (Drakulic et al., 2023) and INViT (Fang et al., 2024) already incorporate distance-based dynamic reduction, we selected LEHD (Luo et al., 2023) as a representative baseline for comparison. As shown in Table 26, when applying the same static reduction to LEHD, we observe a significant time reduction while maintaining solution quality. On the other hand,

Table 26: Comparison with and without static reduction for LEHD and our L2R. Note that "SR" is the static reduction technique we employed, which removes the farthest 10% of connections based on Euclidean distance.

| Method | TSP1K Gap | TSP1K Time | TSP10K Gap | TSP10K Time | TSP100K Gap | TSP100K Time |
|---|---|---|---|---|---|---|
| LEHD-Greedy w/o SR | 3.11% | 98s | 27.24% | 3.1h | OOM | |
| LEHD-Greedy w/ SR | 3.38% | **73s** | 26.76% | **2.3h** | OOM | |
| L2R-Greedy w/o SR | 4.49% | 7s | 4.82% | 68s | 4.79% | 29.6m |
| L2R-Greedy w/ SR | 4.49% | **6s** | 4.82% | **64s** | 4.79% | **29.1m** |

when removing SR from our L2R, performance remains unchanged, but the solving time increases, especially for TSP100K instances.

This ablation study confirms that the static reduction serves as an effective, model-agnostic pre-processing step that improves efficiency without sacrificing solution quality. It also demonstrates that the superior performance of our L2R framework stems from its core architecture, not just the preprocessing step. While the 10% reduction ratio may appear small, pruning these edges reduces the solving time during the construction process, which is particularly beneficial when scaling to very large instances (e.g., 1 million nodes).

## O.6 Effects of Different Training Scale

Table 27: Performance comparison across different training scales. Note that all models are trained for 50 epochs.

| Training Scale | Training time per epoch | TSP1K Obj. | TSP1K Gap | TSP10K Obj. | TSP10K Gap |
|---|---|---|---|---|---|
| $N \in \text{Uniform}([100, 200])$ | 2.3h | **24.2718** | **4.99%** | **75.6892** | **5.45%** |
| $N \in \text{Uniform}([100, 500])$ | 4h | 28.9717 | 25.32% | 88.872 | 23.82% |
| $N = 100$ | **1.3h** | 24.2995 | 5.11% | 75.7319 | 5.51% |

Our proposed approach does not require any labeled data, so in theory, we can use any problem size during the training. However, our decision to train the model exclusively on 100-node instances is driven by two primary considerations. First, this setting aligns with standard protocols in the NCO community, where the majority of state-of-the-art baselines (e.g., INViT (Fang et al., 2024), BQ-NCO (Drakulic et al., 2023), LEHD (Luo et al., 2023)) are trained solely on 100-node instances. Adhering to this convention ensures a fair and direct comparison, isolating the efficacy of our proposed framework. Second, we aim to achieve a good trade-off between training efficiency and generalization performance.

We have conducted additional experiments using varying-scale training strategies, following the protocols established in Omni-VRP (Zhou et al., 2023) and ICAM (Zhou et al., 2024a), respectively. We trained two new model variants: one on instances with sizes uniformly sampled from $[100, 200]$ and another from $[100, 500]$. It is essential to note that, due to the time limit, the model trained on $N \in \text{Uniform}([100, 500])$ is still currently in the training phase (requiring approximately 4 hours per epoch). Therefore, to ensure a fair comparison, we evaluate the performance of all three training strategies at the same checkpoint (50 epochs). As shown in Table 10, we can observe: (1) $N \in \text{Uniform}([100, 200])$: This model achieves only marginal performance improvements compared to the 100-node baseline. However, the training time increased substantially; (2) $N \in \text{Uniform}([100, 500])$: The model fails to converge within a reasonable time. The exponential growth of the search space and the sparsity of rewards in RL make it extremely difficult for the model to learn effective policies from scratch on instances at this scale. These findings validate our choice of the fixed 100-node training setting. It enables the model to learn robust and generalizable local patterns efficiently, which transfer effectively to large-scale problems (up to 10 million nodes), while avoiding the instability and prohibitive computational costs associated with training on larger instances.

# P  LICENSES FOR USED RESOURCES

Table 28: List of licenses for the codes and datasets we used in this work

| Resource | Type | Link | License |
|---|---|---|---|
| Concorde (Applegate et al., 2006) | Code | https://github.com/jvkersch/pyconcorde | BSD 3-Clause License |
| LKH3 (Helsgaun, 2017) | Code | http://webhotel4.ruc.dk/~keld/research/LKH-3/ | Available for academic research use |
| HGS (Vidal, 2022) | Code | https://github.com/chkwon/PyHygese | MIT License |
| GLOP (Ye et al., 2024) | Code | https://github.com/henry-yeh/GLOP | MIT License |
| POMO (Kwon et al., 2020) | Code | https://github.com/yd-kwon/POMO/tree/master/NEW_py_ver | MIT License |
| ELG (Gao et al., 2024) | Code | https://github.com/gaocrr/ELG | MIT License |
| Omni_VRP (Zhou et al., 2023) | Code | https://github.com/RoyalSkye/Omni-VRP | MIT License |
| INViT (Fang et al., 2024) | Code | https://github.com/Kasumigaoka-Utaha/INViT | Available for academic research use |
| LEHD (Luo et al., 2023) | Code | https://github.com/CIAM-Group/NCO_code/tree/main/single_objective/LEHD | Available for any non-commercial use |
| BQ (Drakulic et al., 2023) | Code | https://github.com/naver/bq-nco | CC BY-NC-SA 4.0 license |
| Cross-distribution TSPs(Fang et al., 2024) | Dataset | https://github.com/Kasumigaoka-Utaha/INViT | MIT License |
| Cross-distribution CVRPs(Fang et al., 2024) | Dataset | https://github.com/Kasumigaoka-Utaha/INViT | MIT License |
| TSPLIB (Reinelt, 1991) | Dataset | http://comopt.ifi.uni-heidelberg.de/software/TSPLIB95/ | Available for any non-commercial use |
| CVRPLIB Set-XXL (Arnold et al., 2019) | Dataset | http://vrp.galgos.inf.puc-rio.br/index.php/en/ | Available for academic research use |

We list the used existing codes and datasets in Table 28, and all of them are open-sourced resources for academic usage.

# Q  LLM USAGE

We use a large language model (LLM) solely as a general-purpose writing assistant to improve clarity and readability. Specifically: (1) Rephrasing sentences for fluency and style, improving word choice, and enhancing readability, and (2) Correcting grammar, punctuation, and typographical errors.

The LLM does not contribute to any creation of original technical content. All scientific content and claims are produced and validated by the authors, who bear full responsibility. The LLM is not eligible for authorship.

