# OpenReview forum: "Learning to Reduce Search Space for Generalizable Neural Routing Solver"
_ICLR.cc/2026/Conference — Submitted to ICLR 2026_

### Official Review · Reviewer_NvVt · 2025-10-20

**Soundness:** 3
**Presentation:** 3
**Contribution:** 2
**Rating:** 2
**Confidence:** 4

**Summary:**

This paper presents an enhancement of several state-of-the-art models for Neural Combinatorial Optimization, improving their generalization on larger-scale instances. Specifically, it builds upon the Attention Model by [Kool et al. 2019.], replacing the computation-heavy attention mechanism with a two-step approach. In the first step, the search space is reduced using a learnable policy. In the second step, the solution is constructed by considering only the nodes retained from the first step.

**Strengths:**

1. The two-step approach for reducing the search space and selecting nodes is elegant and straightforward.

2. The proposed method achieves better running time than existing solutions for large-scale neural routing solvers.

**Weaknesses:**

1. This paper begins by claiming that distance-based search space pruning can eliminate globally optimal nodes, making it unsuitable for this domain. Then, the proposed method relied on this distance-based pruning approach.

2. Some important baselines are missing, particularly current state-of-the-art NCO solvers for large-scale routing, such as [1]. Comparing the raw results from that paper with the experiments in this work, it appears that [1] performs better on larger problems.

3. The proposed method is quite naive. It represents only an incremental improvement over existing state-of-the-art models. All the "ingredients" used in this work are already well known. Even the idea of avoiding an attention bottleneck by computing scores between the [first,last] and the remaining nodes has already been explored in [1].

4. This approach is limited to (Euclidean) routing problems only. For non-Euclidean problems, it is not possible to apply coordinate normalization, and likely not feasible to use the initial pruning or candidate node reduction strategies. Additionally, the idea of using the first and the last node in attention computation makes this approach applicable only to the routing problems. This paper claims to present a "Generalizable neural routing solver", but TSP and CVRP alone are not enough to support such a broad claim, and it seems this approach will not work beyond these problems.



[1] Luo et al. Boosting Neural Combinatorial Optimization for Large-Scale Vehicle Routing Problems, ICLR 2025

**Questions:**

In addition to the weaknesses, I would like to raise the following questions:

1. Equation (1), which describes static reduction, is not clear. It introduces a threshold as $1−\gamma$, and later $\gamma$ is defined as 10%. Do you remove the 10% or the 90% of the furthest nodes? If you remove only 10%, that seems like a minimal, almost negligible, amount. On the other hand, if you remove 90% of the furthest nodes, that contradicts your earlier discussion stating that distance-based search space reduction is not an acceptable approach because it prunes many promising nodes. Additionally, what does the parameter $\alpha$ represent in this equation?

2. It seems that after the initial step of static graph reduction, the remaining sparse graph may become disconnected. How do you ensure that the remaining graph stays connected and can still produce a feasible solution?

3. Regarding the experiments, the CVRP results are unclear. There is no reported optimality gap - only objective values are given. Moreover, when comparing with other works, it seems that different datasets are used, making any direct comparison impossible. Additionally, why are the TSP results reported for 1K, 5K, 10K, 50K, and 100K instances, while the CVRP results are limited to 1K, 5K, 7K, and 10K? Where does this large gap come from? Does your model perform poorly on CVRP instances larger than 10K?

4. In Section 4, you explicitly explain that your "model is trained solely on 100-node TSP instances from uniform distribution". Do you train only one model on TSP and test it both on TSP and CVRP, or train two models, one for TSP and another for CVRP?

5. What is the motivation for limiting your training instances to only 100 nodes? Your approach does not require any labeled data, so in theory, you could use any problem size during the training. Does this limitation actually benefit training time or performance? Did you try training on larger instances?

---

> ### Author Response · Authors · 2025-11-24
> **Response to Reviewer NvVt [1/7]**
>
> Thank you very much for your time and effort in reviewing our work. Below, we address each concern point by point. All revisions and additions made to the manuscript have been marked in blue to aid in your review.
>
> > **W1. The proposed method relied on this distance-based pruning approach.**
>
> Thank you very much for this thoughtful observation. We acknowledge that the distinction between "solely distance-based pruning" and our "distance-assisted learning" approach could be articulated more clearly. We are grateful for the opportunity to further clarify our design:
>
> - **Key Distinction:** Our critique in this paper targets methods that rely exclusively on fixed distance heuristics (e.g., static k-Nearest Neighbors) to eliminate nodes. Existing dynamic search space reduction methods are fully implemented based on distance, which can severely harm solution quality if an incorrect reduction is made, especially when facing more complex routing problems (e.g., CVRP and CVRPTW).
>
> In contrast, **as stated in the original manuscript (Page 2), our L2R framework "adaptively combines distance-based priors with context-dependent features learned by the network"**. By incorporating distance as an auxiliary structural prior, we can leverage the efficiency of local search while perceiving and utilizing node attributes beyond simple spatial proximity (e.g., demands). This allows L2R to be effectively applied to more complex routing problems, where the optimal solution depends not only on geometric distance but also heavily on customer demands and time window information.
>
> - **Capability Beyond Spatial Proximity:** We would like to emphasize the core contribution of the proposed L2R is that **L2R is the first framework for learning-based dynamic search space reduction, rather than relying solely on fixed heuristics**. The most compelling evidence that our method does not regress to simple distance-based pruning is its performance on complex constraints. Extensive experiments demonstrate that L2R significantly outperforms methods that rely solely on distance rules (such as INViT[1]). Moreover, L2R can also demonstrate unprecedented generalization capabilities. To the best of our knowledge, it is **the first constructive neural solver to surpass HGS on large-scale CVRP instances ($N \ge 50K$) when trained solely on 100-node uniform instances without any post-processing operations**. (as detailed in our response to W2 and Q3).
>
> In summary, we utilize distance to guide the learning process, rather than rigidly constraining the search scope. We will revise the relevant sections to make this distinction explicit and avoid potential confusion. We sincerely appreciate your thorough reading, which has helped improve the clarity and accuracy of our paper.
>
> > **W2. Some important baselines are missing**
>
> Thank you for pointing out this important baseline. We acknowledge that SIL (Self-Improved Learning[2]) is a powerful neural routing solver for large-scale routing problems. We would like to provide further clarification on this baseline:
>
> - **Key Reason for Initial Exclusion of SIL:** The primary reason for excluding SIL from the initial comparison is the fundamental difference in training strategy. **SIL trains separate models for each target scale** (e.g., a model trained on 10K-node instances is used to solve the 10K-node problem). The better "raw results" you mentioned are achieved by models trained directly on those large-scale instances. In contrast, our L2R is trained exclusively on small-scale instances (100 nodes) and generalizes in zero-shot to large-scale instances (up to 10M nodes). Comparing a model trained on 100 nodes against a model trained on 50K/100K nodes would be an unfair comparison, as it evaluates training capability rather than generalization capability (the focus of our work).
>
> - **Comparison with SIL:** To further address your concern and provide a fair assessment of the generalization performance, we have conducted additional experiments comparing L2R (trained on 100 nodes) with SIL (trained on 1,000 nodes). We selected the SIL-1K model because it is the smallest official model provided by SIL, yet **it still benefits from a training size 10x larger than L2R's**. We evaluate both L2R and SIL on large-scale TSP and CVRP instances (50K and 100K nodes). As shown in **Table 1**, despite a disadvantage in training data scale (100 vs. 1,000), L2R significantly outperforms SIL in both solution quality (Gap) and inference speed, highlighting the efficiency of our search space reduction strategy. These results further confirm that the proposed L2R framework possesses robust generalization capabilities and practical efficiency.
>
> We hope the above clarification and additional experimental results can adequately address your concerns. We will carefully incorporate them into the revised paper.

---

> > ### Comment · Reviewer_NvVt · 2025-11-27
> >
> > Thank you for your detailed and inspiring answers.
> >
> > Many things are clearer to me now.
> >
> > To summarise your responses: you emphasise that (i) this work represents the first learning-based framework for dynamic search-space reduction, and (ii) your model is trained solely on 100-node instances, in contrast to existing SOTA models.
> >
> > Regarding the first claim, I fully agree. As I mentioned, I find the overall approach elegant and straightforward. However, the method is essentially composed of three steps: a naïve distance-based pruning stage, followed by two stages that are already well studied in the NCO community (e.g., Kool et al. (2019) and Kwon et al. (2020)). Splitting the process into two sequential steps and using the top-k items from the first as the input to the second is original. Still, I am not convinced that this is enough for acceptance at such a type of top-tier ML conference.
> >
> > Regarding the second point, as stated in my review, I do not see how training exclusively on 100-node instances constitutes a major advantage. This aspect would be more relevant in a supervised setting (requiring labelled data). Still, under your training setup - as well as in SIL - the size of the training instances is not a critical factor, aside from possibly having a minor effect on training efficiency.
> >
> > A further remark, which does not influence my overall decision: it appears to me that your method is highly tailored to routing-type problems and cannot easily be applied beyond that domain. The reason is the need to represent the current partial solution. Equations (3) and (7) explicitly require the initial node and the latest node; for many other problems (including the mentioned FJSP), such a representation is not feasible. The same holds for SIL paper.
> >
> > Given all of this, I cannot clearly support acceptance of the paper. However, I will not oppose acceptance if other reviewers take a different view. Therefore, I will increase my score to 4.

---

> > > ### Author Response · Authors · 2025-12-01
> > > **Response to C1: One Model for All Scales: From 100 to 10M Nodes**
> > >
> > > We sincerely thank you for recognizing the elegance of our proposed framework and for your willingness to reconsider your score. We greatly appreciate your constructive feedback regarding the novelty and the implications of our training strategy.
> > >
> > > > **C1. One Model for All Scales: The first unified and label-free framework that achieves strong cross-scale generalization performance (from 100 nodes to 10 million nodes) using a single model**
> > >
> > > We fully agree with you that our method builds upon established components in the NCO literature. However, we would like to emphasize that the core contribution of L2R is that **it is the first framework for learning-based dynamic search space reduction, which enables unprecedented robust cross-scale generalization performance (from 100 nodes to 10 million nodes)**. This remains an **open challenge** that existing methods, including SOTA methods like SIL, have not overcome.
> > >
> > > While many existing models (e.g., BQ and LEHD) perform well on small-scale instances (hundreds of nodes), they often fail on large-scale problems (N > 10,000) due to memory limitations. The recent SOTA method, SIL, uses linear attention to scale and performs strongly when trained directly on instances with larger scales (up to 100K nodes). However, our systematic evaluation reveals that **SIL lacks consistent robustness across varying instance scales**:
> > >
> > > - **Experimental Setting:** We evaluated all publicly available SIL models (trained on 1K to 100K nodes) against L2R models trained solely on 100-node instances. Comparisons are conducted on the standard CVRPLib benchmark datasets (Set-X and Set-XXL), covering scales from 100 ≤ N ≤ 30,000.
> > >
> > > - **One Model for All Scales:** As shown in the table below, SIL models exhibit severe performance degradation on small-scale instances (N < 1,000), with optimal gaps often exceeding 40\% or even failing to find feasible solutions (gap > 100\%). In contrast, L2R maintains strong and robust performance across all scales using a single model trained only on 100-node instances.
> > >
> > > - **Label-Free Training:** Another advantage of L2R is that it requires absolutely no labeled data. Unlike SIL, which relies on computationally expensive search strategies (e.g., RRC) to generate high-quality pseudo-labels, L2R operates in a fully label-free manner, substantially reducing computational overhead for model training.
> > >
> > > In summary, we believe L2R offers a **novel and practically impactful contribution** by introducing **a unified and label-free framework that achieves strong cross-scale generalization performance (from 100 nodes to 10 million nodes) using a single model**, which is a key step toward making NCO methods more scalable and widely applicable.
> > >
> > > Table 1. Performance comparison of L2R and different versions of SIL on CVRPLib benchmark at various scales. Please note that we consider instances with a gap greater than 100\% as unsolvable.
> > > |Method|Training|0≤N＜1K||1K≤N＜10K||10K≤N＜100K||All||
> > > |:---:|:---:|:---:|:---:|:---:|:---:|:---:|:---:|:---:|:---:|
> > > ||Scale|Gap|\#Solved|Gap|\#Solved|Gap|\#Solved|Gap|\#Solved|
> > > |SIL-1K greedy|1K|43.23\%|58/99|21.73\%|5/5|15.39\%|6/6|39.26\%|69/110|
> > > |SIL-5K greedy|5K|42.70\%|62/99|18.34\%|5/5|12.38\%|6/6|38.54\%|73/110|
> > > |SIL-10K greedy|10K|46.94\%|62/99|20.78\%|5/5|12.62\%|6/6|42.32\%|73/110|
> > > |SIL-50K greedy|50K|40.04\%|65/99|16.09\%|5/5|**10.81\%**|6/6|36.16\%|76/110|
> > > |SIL-100K greedy|100K|40.44\%|60/99|18.32\%|5/5|10.95\%|6/6|36.39\%|71/110|
> > > |L2R greedy|100|**8.16\%**|**99/99**|**11.62\%**|**5/5**|11.08\%|**6/6**|**8.48\%**|**110/110**|

---

> > > ### Author Response · Authors · 2025-12-01
> > > **Response to C2: Fundamental Challenge in Reinforcement Learning for NCO**
> > >
> > > > **C2. Fundamental challenge in reinforcement learning for NCO**
> > >
> > > Thank you very much for raising this important point. We would like to further clarify the motivation behind our choice to train exclusively on 100-node instances. This design is indeed a **strategic response to a fundamental challenge in RL-based training for NCO**, rather than a minor efficiency consideration.
> > >
> > > - **Model Training with Supervised Learning for NCO:** In SL methods like BQ and LEHD, training benefits from **single-step gradient backpropagation**, where the model is teacher-forced to match the ground-truth action at each step independently. This effectively decouples the optimization from the full decision sequence, eliminating the need to traverse the entire constructive sequence during training and avoiding issues of long-horizon credit assignment and gradient variance accumulation.
> > >
> > > - **Fundamental Challenge in Reinforcement Learning for NCO:** In contrast, RL-based constructive solvers must generate the complete solutions to receive a reward signal after the full construction process. Each training step, therefore, requires rolling out the entire constructive trajectory. Consequently, when training on large-scale instances, the number of decision steps increases dramatically, leading to gradient estimation with high variance. Therefore, an extremely large batch size is required to stabilize the training process, which is typically infeasible due to practical hardware constraints.
> > >
> > > - **The Advantage of L2R:** Our choice to train solely on 100-node instances directly addresses this limitation, which allows us to use **practical batch sizes** to **ensure stable convergence and efficient training**. Crucially, the key advantage demonstrated by L2R is that this small-scale training strategy successfully generalizes to instances of unprecedented scale (up to 10 million nodes). This massive cross-scale generalization ability also validates the **core contribution of our learning-to-reduce framework: a model that learns to iteratively reduce problem size can transfer this capability across vastly different scales**.
> > >
> > > Therefore, the significant advantage lies **not merely in the small training size itself, but in the demonstrated ability** of our framework to **bridge a generalization gap that existing RL-based methods have not overcome**, all while maintaining training stability under realistic computational budgets.

---

> > > ### Author Response · Authors · 2025-12-01
> > > **Response to C3: Domain Specificity**
> > >
> > > > **C3. Domain specificity**
> > >
> > > We appreciate your insightful comment regarding the applicability of our method.
> > >
> > > We agree with you that our current framework, like other autoregressive constructive solvers (e.g, SIL, LEHD, BQ-NCO), leverages the sequential structure inherent in routing problems. Our primary contribution, as reflected in our title "Neural Routing Solver," focuses specifically on introducing a novel learning-to-reduce framework to address the **critical challenge of cross-scale generalization** within this well-established domain. We believe that the demonstrated capability to **scale from 100 to 10M nodes using a single model trained on small instances** represents a significant advancement and substantial contribution in its own.
> > >
> > > We acknowledge that extending this framework directly to problems with fundamentally different structures (e.g., FJSP) would require significant architectural adaptations. However, we believe the **core principle of hierarchical, iterative search space reduction** could inspire future research into scalable solutions for other combinatorial domains. We view this as a promising and exciting direction for further work.
> > >
> > > Thank you again for your thoughtful feedback.
> > >
> > > ---
> > >
> > > We sincerely thank you for your constructive engagement throughout this discussion and for your willingness to raise your score. Your insightful comments and feedback have undoubtedly strengthened our work.
> > >
> > > We hope our responses have satisfactorily addressed your remaining concerns, and believe **L2R represents a meaningful step forward in addressing scalability and generalization challenges in neural combinatorial optimization**.

---

> ### Author Response · Authors · 2025-11-24
> **Response to Reviewer NvVt [2/7]**
>
> Table 1. Performance comparison with SIL on large-scale TSP and CVRP instances. Note that "Time" is the average inference time. Each dataset includes 16 instances.
> |Method|TSP50K||TSP100K||CVRP50K||CVRP100K||
> |:---:|:---:|:---:|:---:|:---:|:---:|:---:|:---:|:---:|
> ||Obj.(Gap)|Time|Obj.(Gap)|Time|Obj.(Gap)|Time|Obj.(Gap)|Time|
> |Near-optimal|159.93(0.00\%)|10h|225.99(0.00\%)|25h|1081.00(0.00\%)|4h|2087.51(0.00\%)|6h|
> |SIL greedy|178.99(11.92\%)|8.6m|266.63(17.98\%)|36.2m|1108.28(2.52\%)|9.2m|2153.78(3.17\%)|45.8m|
> |L2R greedy|**167.70(4.86\%)**|**35.5s**|**236.81(4.79\%)**|**1.8m**|**1074.98(-0.56\%)**|**37s**|**2055.14(-1.55\%)**|**2m**|
> |SIL PRC1000|167.01(4.43\%)|1.6h|241.68(6.94\%)|3.0h|1094.63(1.26\%)|1.6h|2140.02(2.52\%)|3.5h|
> |L2R PRC1000|**164.41(2.80\%)**|**12.4m**|**232.77(3.00\%)**|**15.5m**|**1070.49(-0.98\%)**|**33m**|**2050.91(-1.75\%)**|**1.1h**|
>
> > **W3. the novelty of our work and its distinction from existing literature**
>
> Thank you for your insightful comments regarding the novelty of our work. We are grateful for the opportunity to further clarify the novelty of our work and its distinction from existing literature, particularly reference SIL[2]:
>
> - **Our Contributions**: We agree with you that the architectural components (e.g., attention mechanisms, context embeddings) in L2R are widely used in the NCO community. This common foundation allows us to focus our contribution on the novel L2R mechanism. However, we respectfully disagree with the characterization of L2R as "quite naive."  We would like to emphasize that the key contribution of L2R  lies in introducing **the first learning-based framework for dynamic search space reduction**, which goes beyond the existing purely heuristic-based approaches.
>
> Existing search space reduction methods rely entirely on distance-based rules, which can severely harm the solution quality if an incorrect reduction is made. In contrast, L2R learns reduction policies dynamically, which enables effective reduction and leads to promising performance on complex routing problems such as CVRP and CVRPTW (see the response for W4), where optimal solutions depend not only on geometric distance but also heavily on customer demands and time window information. As a result, L2R can significantly outperform methods that rely solely on distance-based rules such as INViT[1].
>
> - **Different Roles of [First, Last] Nodes**:  For the role of first and last node, we would like to clarify a potential misunderstanding regarding our model design and its fundamental difference from SIL:
>
> (1) In SIL, these two nodes are employed as representative nodes to construct a specific cross-attention module, reducing the computational complexity of the model from $\mathcal{O}(N^2)$ to linear complexity $\mathcal{O}(N)$. This is an architectural solution to the high cost of self-attention.
>
> (2) In L2R, the first and last nodes are used solely to form the context embedding, which is mathematically justified by the structure of the routing problem, and it is naturally modeled as a **Markov Decision Process**. For a routing problem, the transition probability depends only on the current state and the remaining unvisited nodes, rather than the full set of visited nodes along the route. Given the Hamiltonian cycle constraint, the current state is sufficiently defined by the current node ($\pi_{t-1}$) and the origin/target node ($\pi_{1}$). This formulation is a standard practice in current representative neural routing solvers (e.g., INViT[1] and BQ-NCO[5]), rather than to alter the attention complexity itself.
>
> - **Different Approaches to Scalability:** We respectfully disagree that our method is an incremental improvement for avoiding attention bottlenecks. Our approach to handling large-scale complexity is different from SIL: SIL achieves scalability by designing an advanced linear attention mechanism that efficiently processes all nodes. In contrast, L2R achieves scalability by learning-based search space reduction. Instead of trying to compute attention over all nodes more cheaply, we learn to dynamically prune the search space to a small and manageable size $k$ (e.g., $k=20$ in TSP) before the heavy computation occurs.
>
> In summary, by learning to dynamically reduce the search space, **L2R avoids the quadratic complexity issue at the decision level, rather than the architectural level**. This is the core contribution and novelty of our work, which enables us to solve massive-scale instances (up to 10M nodes). Remarkably, L2R can also demonstrate unprecedented generalization capabilities. To the best of our knowledge, it is **the first constructive neural solver to surpass HGS on large-scale CVRP instances ($N \ge 50K$) when trained solely on 100-node uniform instances without any post-processing operations**.

---

> ### Author Response · Authors · 2025-11-24
> **Response to Reviewer NvVt [3/7]**
>
> > **W4. The scope and applicability of our approach**
>
> Thank you for your thoughtful analysis regarding the scope and applicability of our approach. We acknowledge that our current implementation focuses on 2D Euclidean problems, but we respectfully disagree that the core methodology is inherently limited to them. We would like to clarify the definition of "Generalizable" in our context and the adaptability of our framework.
>
> - **Applicability to Non-Euclidean Problems:** While we demonstrate results on Euclidean graphs, the L2R framework can be adapted to non-Euclidean settings:
>
> (1) Coordinate normalization is a specific preprocessing step for 2D data, primarily used to homogenize distance distributions within each extracted sub-graph, thereby enhancing the stability and consistency of the model's perception across different sub-graphs. For non-Euclidean problems (e.g., those with Manhattan Distance), standard techniques such as distance matrix normalization can replace coordinate normalization without altering the core L2R logic;
>
> (2) Both static and dynamic reduction rely on cost values ($d_{ij}$), not necessarily geometric coordinates. **As long as a cost metric exists (even if asymmetric), the principle of filtering high-cost candidates remains valid.**
>
> - **Architecture Specificity:** The use of first and last node embeddings is designed to capture the state of a sequential decision process, based on the Markov decision process nature inherent in routing problems. This design is indeed specific to Routing Problems, which aligns perfectly with the stated scope of our paper. On the other hand, search space reduction is a general strategy applicable to many combinatorial optimization problems. Analogous to L2R, the reduction strategy employed in [3] effectively solves the Flexible Job-Shop Scheduling Problem by restricting the action space of dispatching rules by selecting only the top-$k$ operations based on specific criteria (e.g., earliest start or earliest completion times). We believe our framework has potential for a broader range of applications, and extending L2R to non-routing problems is an important direction for future work. We sincerely appreciate your valuable comment, which has greatly inspired meaningful future directions for enhancing the L2R framework.
>
>
> - **Clarification on Generalizable:** In the context of this work (and consistent with NCO literature[4]), "generalizable" primarily refers to cross-scale generalization (e.g., model trained on 100-node instances to solve problems with up to 1M nodes ) and cross-distribution generalization (e.g., model trained on uniformly distributed instances to solve instances from varied distributions like cluster/explosion ). Our results demonstrate unprecedented success in these aspects, which strongly supports our claim. We do not intend to claim that this specific architecture can solve non-routing problems, as our title explicitly describes L2R as a "neural routing solver."
>
> - **Verification on Complex Constraints:** To further validate the applicability of L2R on problems with complex non-spatial constraints, we have conducted additional experiments on the Capacitated Vehicle Routing Problem with Time Windows (CVRPTW). Unlike TSP or CVRP, CVRPTW involves complex time window constraints alongside spatial and demand constraints. This makes it particularly challenging for the reduction methods, since they must learn the latent relationship between multiple non-spatial attributes (demand and time windows) and the suitability of candidate nodes.
>
> According to the results in **Table 2**, our proposed L2R consistently achieves the best solution quality, complemented by remarkably fast inference times, across various problem scales. The advantages of L2R over existing neural solvers are significant. This confirms that our learning-based reduction approach can successfully capture complex non-spatial constraints that simple distance-based heuristics cannot, proving the broad applicability of the L2R framework beyond simple routing tasks.
>
> Table 2. Comparison of CVRPTW instances. Note that "Time" indicates the total running time. We limit the test scale to 10,000 nodes, as existing NCO methods consistently run out of Memory (OOM) on instances where $N \ge 5,000$.
> |Method|CVRPTW1K||CVRPTW5K||CVRPTW10K||
> |:---:|:---:|:---:|:---:|:---:|:---:|:---:|
> ||Obj.(Gap)|Time|Obj.(Gap)|Time|Obj.(Gap)|Time|
> |HGS-PyVRP|159.35(0.00\%)|20m|787.25(0.00\%)|3.3h|1375.84(0.00\%)|5h|
> |MTPOMO aug*8|219.22(37.57\%)|1.7m|OOM||OOM||
> |MVMoE aug*8|233.53(46.55\%)|1.7m|OOM||OOM||
> |ReLD aug*8|183.31(15.036\%)|1.6m|OOM||OOM||
> |L2R greedy|**174.31(9.39\%)**|**8s**|**849.83(7.95\%)**|**40s**|**1497.30(8.83\%)**|**1.4m**|

---

> ### Author Response · Authors · 2025-11-24
> **Response to Reviewer NvVt [4/7]**
>
> > **Q1. Equation (1), which describes static reduction, is not clear.**
>
> Thank you for the careful examination of Equation (1). We apologize for the confusion caused by the notation and the typo regarding the parameter $\alpha$.
>
> - **Clarification on Reduction Ratio:** In Equation (1), the condition $\mathrm{rank}(d_{ij}) \leq (1 - \gamma)|\Omega|, \quad \gamma \in [0,1]$ means that we retain the closest 90\% of the nodes and remove the farthest 10\%. As you noted, this is indeed a minimal reduction. The static reduction is designed as a conservative preprocessing step, rather than a primary method for reducing the search space. Its goal is to filter out clearly redundant long-distance connections, thereby improving memory efficiency for very large graphs (e.g., 10M nodes) without the risk of pruning optimal edges.
>
> - **No Contradiction:** Our criticism in the introduction targets aggressive distance-based pruning (e.g., keeping only the top-$k$ nearest neighbors, where $k$ is small, like 20 or 50). Such aggressive pruning can discard globally optimal nodes. In contrast, retaining 90\% of the neighbors is a safe strategy that preserves the solution space while acting as a preliminary filter. The main reduction step is subsequently performed by our learning-based reduction model.
>
>
> - **Necessity of Static Reduction:** We conduct an ablation study to verify the necessity of this static reduction phase, as detailed in **Table 3**. Since strong baselines like BQ[9] and INViT[1] already incorporate distance-based dynamic reduction, we selected LEHD[10] as a representative baseline for comparison.
>
> According to the results, on the LEHD baseline, enabling static reduction reduces the inference time on TSP10K from 3.1 hours to 2.3 hours, which is a significant speedup while maintaining solution quality. On the other hand, when removing SR from our L2R, performance remains unchanged, but the solving time increases, especially for TSP100K instances. This result confirms that the static reduction serves as an effective, model-agnostic preprocessing step that improves efficiency without sacrificing solution quality. It also demonstrates that the superior performance of our L2R framework stems from its core architecture, not just the preprocessing step. While a 10\% reduction may appear small, pruning these edges can meaningfully reduce the solving time during the construction process, which is beneficial for scaling to massive instances (e.g., 1 million nodes).
>
> - **Clarification on Parameter:** We apologize for the oversight. The symbol $\alpha$ in Equation (1) is a typo that was unintentionally left over from a previous version of the manuscript. It does not represent any active parameter in the current formulation and should be removed. The correct threshold parameter is $\gamma$, as defined in the text.
>
> We will carefully revise our paper to incorporate the points discussed above. Thank you again for your constructive comment.
>
> Table 3. Comparison with and without static reduction for LEHD and our L2R. Note that "SR" is the static reduction technique we employed, which removes the farthest 10\% of connections based on Euclidean distance.
> |Method|TSP1K||TSP10K||TSP100K||
> |:---:|:---:|:---:|:---:|:---:|:---:|:---:|
> ||Gap|Time|Gap|Time|Gap|Time|
> |LEHD-Greedy w/o SR|3.11\%|98s|27.24\%|3.1h|OOM||
> |LEHD-Greedy w/ SR|3.38\%|**73s**|26.76\%|**2.3h**|OOM||
> |L2R-Greedy w/o SR|4.49\%|7s|4.82\%|68s|4.79\%|29.6m|
> |L2R-Greedy w/ SR|4.49\%|**6s**|4.82\%|**64s**|4.79\%|**29.1m**|

---

> ### Author Response · Authors · 2025-11-24
> **Response to Reviewer NvVt [5/7]**
>
> > **Q2. How do you ensure that the remaining graph stays connected and can still produce a feasible solution?**
>
> Thank you for raising the concern regarding graph connectivity. We appreciate the opportunity to clarify the specific configuration of our static reduction approach and explain why connectivity is theoretically and empirically guaranteed in our framework.
>
> - **Conservative Pruning Strategy:** As we responded regarding Q1, our static reduction strategy is designed to be conservative. We remove only the 10\% connections of the farthest nodes, which means 90\% of the edges are retained for every node. Unlike aggressive sparsification methods (e.g., k-NN with $k=20$), our static reduction results in a dense graph $G'$ where each node maintains a high degree ($degree \approx 0.9N$).
>
> - **Connectivity Guarantee:** From a graph theory perspective, retaining 90\% of the edges ensures that the graph remains highly connected. According to Dirac's theorem[7], for a graph with $N$ nodes, if the minimum degree of each node is at least $\lceil N/2 \rceil$, the graph is guaranteed to be Hamiltonian (and thus connected). Since our retained degree is approximately $0.9N$ (which is $\gg 0.5N$), the remaining sparse graph $G'$ is mathematically guaranteed to stay connected and contain feasible solutions.
>
> - **Role of Static Reduction:** We emphasise that the primary goal of this static phase is merely to act as a preprocessing step to filter out obviously non-optimal long-distance edges to save memory, rather than performing aggressive search space reduction. Empirically, we have never encountered a disconnected graph or feasibility issue with this setting across all tested scales (up to 10M nodes).
>
> > **Q3. The experiments of CVRPs**
>
> Thank you for the rigorous examination of our experimental results. We apologize for the oversight regarding the missing optimality gaps and the limited range of reported CVRP scales. We have updated the manuscript to include all optimality gaps.
>
> - **Performance on Ultra-Large Scale CVRP:** In the original paper, we reported the CVRP results on instances with 1K, 5K, 7K and 10K nodes following the setting in TAM[8]. Following your suggestion, we have conducted new experiments on CVRP50K and CVRP100K. To ensure a strong baseline, we evaluate HGS on a high-performance server equipped with an Intel(R) Xeon(R) Platinum 8352V CPU at 2.10 GHz and 80GB of RAM. Due to the excessive memory consumption of HGS on these large-scale instances, parallel processing is infeasible on CVRP100K instances. We allocate sufficient time budgets based on the instance size to execute HGS (4 hours for CVRP50K instances and 6 hours for CVRP100K instances).
>
> As shown in **Table 4**, the results are remarkably promising, L2R achieves unprecedented performance. To the best of our knowledge, **L2R is the first constructive neural solver trained solely on small-scale instances (100 nodes, uniform) to outperform the leading heuristic HGS on large-scale instances ($N \ge 50K$) using only greedy decoding (without any post-processing)**. This confirms that our model maintains strong generalization performance on large-scale CVRPs, and its advantage over current neural routing solvers indeed amplifies as the scale increases.
>
> - **Clarification on Dataset and Capacity Settings:** Regarding the dataset consistency, we strictly followed the capacity constraints and generation protocols established in TAM[8]. This setting is the current mainstream standard for large-scale CVRP comparisons and has been adopted by recent representative works such as GLOP[9] and UDC[10]. Therefore, the comparisons in our paper are conducted under consistent and fair conditions.
>
> - **Validation on CVRPLib:** To further address the raised concern regarding dataset variations and ensure a fair benchmark, we evaluate L2R on the widely-used CVRPLib (Set-XXL) benchmark ($3,000 \leq N \leq 30,000$). We also include SIL[2] (as requested in W2) in this comparison. As detailed in **Table 5**, L2R consistently achieves SOTA performance on these standardized benchmarks. Notably, **L2R-greedy alone surpasses all comparable baselines, including SIL-PRC1000**. This further validates the superiority of L2R on complex routing problems beyond synthetic distributions.

---

> ### Author Response · Authors · 2025-11-24
> **Response to Reviewer NvVt [6/7]**
>
> Table 4. Comparison of CVRP instances with uniform distribution.
> |Method|CVRP1K||CVRP5K||CVRP10K||CVRP50K||CVRP100K||
> |---|:---:|:---:|:---:|:---:|:---:|:---:|:---:|:---:|:---:|:---:|
> ||Obj.(Gap)|Time|Obj.(Gap)|Time|Obj.(Gap)|Time|Obj.(Gap)|Time|Obj.(Gap)|Time|
> |HGS|41.2(0.00\%)|5m|126.2(0.00\%)|5m|227.3(0.00)|5m|1081.0(0.00\%)|4h|2087.5(0.00\%)|6h|
> |GLOP-G(LKH-3)|45.9(11.4\%)|1.1s|140.6(11.4\%)|4.0s|256.4(11.1\%)|6.2s|OOM||OOM||
> |LEHD greedy|44(6.80\%)|0.8s|138.2(9.51\%)|1.4m|256.3(12.76\%)|12m|OOM||OOM||
> |LEHD RRC1,000|**42.4(2.91\%)**|3.4m|132.7(5.15\%)|10m|243.8(7.28\%)|51.6m|OOM||OOM||
> |BQ greedy|44.2(7.28\%)|1s|139.9(10.86\%)|18.5s|262.2(15.35\%)|2m|OOM||OOM||
> |BQ bs16|43.1(4.61\%)|14s|136.4(8.08\%)|2.4m|OOM||OOM||OOM||
> |POMO aug*8|101(145.15\%)|4.6s|632.9(401.51\%)|11m|OOM||OOM||OOM||
> |ELG aug*8|46.4(12.62\%)|10.3s|OOM||OOM||OOM||OOM||
> |INViT-3V greedy|48.3(17.23\%)|1s|146.2(15.85\%)|7s|262.5(15.54\%)|1.2m|1331.1(23.1\%)|5.6m|2683.4(28.55\%)|22m|
> |L2R greedy|45.8(11.27\%)|0.1s|136.0(7.74\%)|0.5s|236.7(4.18\%)|4.4s|1075.0(-0.56\%)|37s|2055.1(-1.55\%)|2m|
> |L2R PRC100|44.7(8.53\%)|4.0s|132.7(5.14\%)|14.4s|233.2(2.62\%)|39.9s|1072.6(-0.78\%)|3.5m|2051.5(-1.73\%)|7m|
> |L2R PRC500|44.4(7.75\%)|20.0s|131.2(3.97\%)|1.2m|230.9(1.65\%)|3.0m|1071.2(-0.90\%)|17m|2051.1(-1.74\%)|33m|
> |L2R PRC1,000|44.3(7.42\%)|40.3s|**130.6(3.52\%)**|2.3m|**230.1(1.26\%)**|6.0m|**1070.5(-0.98\%)**|33m|**2050.9(-1.75\%)**|1.1h|
>
> Table 5. Performance comparison on large-scale CVRPLIB Set XXL benchmark ($3,000 \leq N \leq 30,000$). The symbol * indicates that some instances are skipped due to the OOM issue.
> |Method|3K $ \leq N \leq $7K|7K < $ N \leq $ 30K|All|Solved\#|
> |---|:---:|:---:|:---:|:---:|
> ||(4 instances)|(6 instances)|(10 instances)||
> |GLOP|17.07\%|21.32\%|19.62\%|10/10|
> |TAM-LKH3|20.44\%|$-$|$-$|$-$|
> |BQ bs16|20.20\%|OOM|$-$|4/10|
> |LEHD greedy|22.22\%|32.80\%*|$-$|6/10|
> |LEHD RRC1000|14.06\%|21.52\%*|$-$|6/10|
> |POMO aug*8|331.24\%*|OOM|$-$|2/10|
> |ELG aug*8|16.82\%*|OOM|$-$|2/10|
> |INViT-3V greedy|20.74\%|26.64\%|24.28\%|10/10|
> |SIL greedy|23.54\%|15.41\%|18.66\%|10/10|
> |SIL PRC1000|13.47\%|11.35\%|12.20\%|10/10|
> |L2R greedy|12.05\%|11.22\%|11.55\%|10/10|
> |L2R PRC100|8.62\%|9.34\%|9.05\%|10/10|
> |L2R PRC1000|**7.81\%**|**8.37\%**|**8.14\%**|10/10|
>
> > **Q4. Do you train only one model on TSP and test it both on TSP and CVRP, or train two models, one for TSP and another for CVRP?**
>
> Thank you very much for pointing out this lack of clarity. We apologize for the confusion caused by the ambiguity in our phrasing. To clarify, we trained two separate models: one specifically for the TSP and another specifically for the CVRP, both of which are trained on 100-node instances drawn from a uniform distribution. The statement "model is trained solely on 100-node TSP instances" is intended to emphasize the small scale of the training data (i.e., we did not train on 1K or 10K instances), rather than implying that the TSP model is used to solve CVRP.
>
> In the revised paper, we will rephrase this sentence to explicitly state: **"For each problem type (TSP, CVRP, and CVRPTW), the corresponding model is trained solely on 100-node instances from the uniform distribution"**.

---

> ### Author Response · Authors · 2025-11-24
> **Response to Reviewer NvVt [7/7]**
>
> > **Q5. What is the motivation for limiting your training instances to only 100 nodes?**
>
> Thank you for raising this insightful question regarding our training scale selection. Our decision to train the model exclusively on 100-node instances is driven by two primary considerations. First, this setting aligns with standard protocols in the NCO community, where the majority of representative baselines (e.g., INViT[1], BQ-NCO[5], LEHD[6]) are trained solely on 100-node instances. Adhering to this convention **ensures a fair and direct comparison, isolating the efficacy of our proposed framework**. Second, we aim to achieve a **good trade-off between training efficiency and generalization performance**.
>
> To address your concern, we conducted additional experiments using varying-scale training strategies, following the protocols established in Omni-VRP[4] and ICAM[11], respectively. We trained two new model variants: one on instances with sizes uniformly sampled from $[100, 200]$ and another from $[100, 500]$. It is essential to note that, due to the time limit, the model trained on $N \in \mathrm{Uniform}([100, 500])$ is still currently in the training phase (requiring approximately 4 hours per epoch). Therefore, to ensure a fair comparison, we evaluate the performance of all three training strategies at the same checkpoint (50 epochs).
>
> The results in **Table 6** offer the following insights:
>
> (1) $N \in \mathrm{Uniform}([100, 200])$: This model achieves only marginal performance improvements compared to the 100-node baseline. However, the training time increased substantially.
>
> (2) $N \in \mathrm{Uniform}([100, 500])$: The model fails to converge within a reasonable time. The exponential growth of the search space and the sparsity of rewards in RL make it extremely difficult for the model to learn effective policies from scratch on instances at this scale.
>
> These findings validate our choice of the fixed 100-node training setting. It enables the model to learn robust and generalizable local patterns efficiently, which transfer effectively to large-scale problems (up to 10 million nodes), while avoiding the instability and prohibitive computational costs associated with training on larger instances.
>
> We sincerely thank you for this professional comment. We will incorporate the above discussion and experimental results into the revised paper.
>
> Table 6. Performance comparison across different training scales. Note that all models are trained for 50 epochs.
> |Training scale|Training time|TSP1K||TSP10K||
> |:---:|:---:|:---:|:---:|:---:|:---:|
> ||per epoch|Obj.|Gap|Obj.|Gap|
> |$N\in\mathrm{Uniform}([100,200])$|2.3h|**24.2718**|**4.99\%**|**75.6892**|**5.45\%**|
> |$N\in\mathrm{Uniform}([100,500])$|4h|28.9717|25.32\%|88.872|23.82\%|
> |$N=100$|**1.3h**|24.2995|5.11\%|75.7319|5.51\%|
>
> > **References**
>
> [1] Invit: A generalizable routing problem solver with invariant nested view transformer, ICML 2024.
>
> [2] Boosting Neural Combinatorial Optimization for Large-Scale Vehicle Routing Problems, ICLR 2025
>
> [3] Diverse policy generation for the flexible job-shop scheduling problem via deep reinforcement learning with a novel graph representation. Engineering Applications of Artificial Intelligence, 2025, 139: 109488.
>
> [4] Towards omni-generalizable neural methods for vehicle routing problems, ICML 2023.
>
> [5] Bq-nco: Bisimulation quotienting for efficient neural combinatorial optimization, NeurIPS 2023.
>
> [6] Neural combinatorial optimization with heavy decoder: Toward large scale generalization, NeurIPS 2023.
>
> [7] Dirac G A. Some theorems on abstract graphs. Proceedings of the London Mathematical Society, 1952, 3(1): 69-81.
>
> [8] Generalize Learned Heuristics to Solve Large-scale Vehicle Routing Problems in Real-time, ICLR 2023.
>
> [9] Glop: Learning global partition and local construction for solving large-scale routing problems in real-time, AAAI 2024.
>
> [10] UDC: A unified neural divide-and-conquer framework for large-scale combinatorial optimization problems, NeurIPS 2024.
>
> [11] Instance-Conditioned Adaptation for Large-scale Generalization of Neural Routing Solver, arXiv preprint arXiv:2405.01906, 2024.

---

### Official Review · Reviewer_LmUY · 2025-10-31

**Soundness:** 3
**Presentation:** 3
**Contribution:** 2
**Rating:** 6
**Confidence:** 4

**Summary:**

This paper proposes L2R, a hierarchical search space reduction framework for neural routing solvers that combines static sparsification, a lightweight learned dynamic reducer, and a local construction policy trained jointly via REINFORCE. Trained only on 100-node uniform instances, L2R shows strong cross-size generalization up to 1M nodes and robustness across distributions, achieving favorable speed–quality trade-offs versus NCO baselines and competitive gaps with classic heuristics at large scales.

**Strengths:**

1.	The idea of learning a policy for dynamic search space reduction is valuable, especially for scaling constructive NCO to very large routing instances.

2.	The proposed approach is coherent and empirically effective. The reported cross-size generalization (up to 1M nodes) and cross-distribution robustness are noteworthy, with favorable speed–quality trade-offs.

**Weaknesses:**

1. Experiments for N < 1000 are missing. Most local models without global awareness degrade on small-scale problems; thus, performance on small- and medium-scale instances should be included to assess performance comprehensively. The L2R model is trained on N = 100, but results on the training scale are not reported. Omitting these makes it difficult to judge overfitting and optimality on the training distribution.

2. Given the hierarchical nature of the decision process, established hierarchical RL methods could be directly applied or adapted to improve credit assignment and training stability for L2R. In this context, the simple joint REINFORCE used in the paper may not be the most suitable or advanced choice.

**Questions:**

As shown in Table 15, the improvement in SSR accuracy is marginal, raising concerns about whether the RL-based reduction policy sufficiently explores and learns optimal strategies. It is possible that the solver policy rarely visits a substantial portion of optimal actions during training, limiting the reduction policy’s opportunity to learn them. Would incorporating supervised signals (e.g., oracle candidate sets) plausibly increase candidate-set recall and improve overall accuracy?

---

> ### Author Response · Authors · 2025-11-24
> **Response to Reviewer LmUY [1/2]**
>
> Thank you very much for your time and effort in reviewing our work. Below, we address each concern point by point. All revisions and additions made to the manuscript have been marked in blue to aid in your review.
>
> > **W1. Experiments for N < 1000 are missing.**
>
> Thank you for raising this important concern. We agree that assessing performance on small-scale instances is crucial for a comprehensive evaluation, especially in terms of the training distribution (N = 100). We apologize for not including these results in the initial submission. To address this, we have conducted new experiments on both TSP and CVRP instances with sizes $N = 100$ and $N = 500$. We directly compare L2R with current representative NCO solvers. For small-scale CVRP, the capacity settings follow the setup in LEHD[1].
>
> As shown in the table below, our L2R achieves **highly competitive performance on these small-scale instances** compared to mainstream NCO solvers, such as BQ, LEHD, and ELG. Furthermore, the results also highlight L2R's superior scalability. These comprehensive experiments confirm that L2R's promising large-scale generalization ability does not come at the cost of performance degradation on small-scale instances.
>
> We would like to emphasize that the core contribution of this work lies in pushing the frontier of NCO in terms of generalization and scalability. Comprehensive experiments demonstrate that our proposed L2R method, trained only on 100-node instances, can generalize well to handle instances with up to 10 million nodes. As instance size increases, most NCO solvers (BQ, LEHD, POMO, ELG) fail due to Out-of-Memory (OOM) errors, while L2R's performance advantage becomes more significant. It is particularly noteworthy that on large-scale CVRP (e.g., CVRP100K), **our L2R finds a higher-quality solution than the powerful heuristic HGS (which has been refined for decades)**, and does so with a dramatic speedup (**31 minutes vs. 4 days on CVRP100K instances**). This represents a significant milestone in NCO: **the first constructive neural solver to achieve superior performance over HGS on large-scale CVRP instances ($N \ge 50K$) that only trained on 100-node uniform instances without any post-processing operations**.
>
> The new experiments you suggested helpfully validate that L2R is also a competitive solver on small-scale instances. We sincerely appreciate your valuable suggestion and will include these new results and corresponding discussions in the revised paper.
>
> Table 1. Experimental results on varying-scale TSP and CVRP instances with uniform distribution. All models are trained on instances of size N = 100. Note that "Time" indicates the total running time, and all methods employ greedy decoding, without utilizing instance augmentation or additional search strategies (e.g., RRC or PRC).
> |||TSP100(128 ins.)||TSP500(128 ins.)||TSP50K(16 ins.)||TSP100K(16 ins.)||
> |---|---|:---:|:---:|:---:|:---:|:---:|:---:|:---:|:---:|
> |Method||Obj.|Time|Obj.|Time|Obj.|Time|Obj.|Time|
> |LKH3||7.73|1m|16.52|32m|159.93|10h|225.99|25h|
> |BQ||7.76|2s|**16.72**|46s|OOM||OOM||
> |LEHD||7.78|1s|16.78|16s|OOM||OOM||
> |POMO||**7.76**|1s|20.50|9s|OOM||OOM||
> |ELG||7.78|1s|17.82|17s|OOM||OOM||
> |INViT||8.10|2s|17.55|12s|171.42|1.3h|242.26|5h|
> |L2R||7.84|2s|17.03|3s|**167.70**|9.5m|**236.81**|29.4m|
> |||**CVRP100(128 ins.)**||**CVRP500(128 ins.)**||**CVRP50K(16 ins.)**||**CVRP100K(16 ins.)**||
> |Method||Obj.|Time|Obj.|Time|Obj.|Time|Obj.|Time|
> |HGS||15.52|3.5m|36.56|4h|1081.00|1.3d|2087.51|4d|
> |BQ||16.03|2s|38.43|47s|OOM||OOM||
> |LEHD||16.16|1s|**38.41**|17s|OOM||OOM||
> |POMO||**15.80**|1s|48.22|10s|OOM||OOM||
> |ELG||15.94|1s|39.67|23s|OOM||OOM||
> |INViT||16.77|2s|43.44|12s|1331.10|1.5h|2683.40|5.8h|
> |L2R||16.12|2s|39.31|4s|**1074.98**|10m|**2055.14**|31m|
>
> > **W2. The simple joint REINFORCE used in the paper may not be the most suitable or advanced choice.**
>
> Thank you for this valuable suggestion. We fully agree with your expert assessment that applying established hierarchical RL methods could further improve the training stability and performance of L2R, particularly in terms of credit assignment between the reduction model and construction models.
>
> In this regard, we would like to clarify the research focus and unique contribution of L2R. In this work, our primary goal is to address the critical scalability challenge in NCO. The choice of a joint REINFORCE algorithm is intended to provide a straightforward training objective that clearly demonstrates the core architectural innovations of our proposed learning-to-reduce framework. Notably, the strong performance achieved even with this simple combined loss validates the effectiveness of our hierarchical framework, which integrates reduction and construction phases.
>
> As mentioned in the limitations section, developing more efficient and stable training strategies remains an important direction for future work. Your insightful suggestion offers a clear and promising path for such improvements.

---

> ### Author Response · Authors · 2025-11-24
> **Response to Reviewer LmUY [2/2]**
>
> > **Q1. Incorporating supervised signals:**
>
> Thank you for this insightful comment. We appreciate your suggestion to investigate the potential benefits of incorporating supervised signals. Your concern that the RL-based reduction policy might not sufficiently explore optimal candidates is also reasonable. To address this, we have conducted a new experiment following your suggestion precisely.
>
> - **Experimental Setup:** Taking TSP as a test case, we utilize a dataset of 1 million near-optimal solutions for TSP100 instances (provided by LEHD[1]) to serve as an oracle. We then incorporate a supervised penalty term into the reward function for the reduction model ($\theta_R$), which penalizes the model whenever the oracle's next node is not included in the predicted candidate set. Following the formulation in PIP [2], the penalty term $J_{OUT}$ is calculated as the number of penalized nodes, and the modified loss function is:
> $$ \nabla_{\theta_{R}} L_{R_{SS}} (\theta_{R}) = E_{o(\tau| S,{\theta_{R}})} [(R(\pi| S,{\theta_{L}})- \lambda * J_{OUT}-b(S)) \nabla_{\theta_{R}} \log o(\tau| S)]$$
> We have tested two supervised versions with penalty coefficients $\lambda=1.0$ and $\lambda=0.1$.
>
> - **Results and Analysis:** As shown in **Table 2**, compared to the original L2R, incorporating supervised signals can indeed lead to a higher average optimality ratio (i.e., the proportion of construction steps where the optimal next-visit node resides within the $k$-nearest candidates), which confirms your hypothesis. However, this improvement in candidate selection does not consistently yield better final solution quality (Optimality Gap). On average, the original L2R significantly outperforms both supervised variants (L2R-SS).
> - **Further Discussion:** We speculate that although supervised signals encourage the reduction model to generate candidate sets that better align with the oracle's choices, the local construction model may not fully capitalize on this move for subsequent steps. A theoretically optimal move can become a local trap if the lower policy cannot maintain that optimality. In contrast, the joint RL framework fosters co-adaptation: the reduction model learns to provide candidates that are compatible with the construction model’s behavior, prioritizing feasible high-reward trajectories over rigid adherence to an oracle.
>
> We sincerely thank you for your valuable feedback and suggestions, which provide important guidance for future improvements to the L2R framework.
>
> Table 2.Comparison of different reduction approaches on TSPLIB instances. Here, D-SSR denotes distance-based search space reduction, and SS indicates the addition of supervised signals to the reduction model in training. "Optimality Ratio" measures the proportion of nodes where the optimal next-visit node is located within the $k$ candidate nodes of the current node.
> |Instance|Scale|OptimalityRatio$\uparrow$||||Gap$\downarrow$||||
> |---|:---:|:---:|:---:|:---:|:---:|:---:|:---:|:---:|:---:|
> |||D-SSR|L2R|L2R-SS ($\lambda=1$)|L2R-SS ($\lambda=0.1$)|D-SSR|L2R|L2R-SS($\lambda=1$)|L2R-SS($\lambda=0.1$)|
> |pcb1173|1173|90.54\%|**90.96\%**|90.37\%|90.45\%|5.17\%|**4.02\%**|4.76\%|5.25\%|
> |rl1304|1304|93.94\%|94.63\%|**95.09\%**|**95.09\%**|9.88\%|8.14\%|7.82\%|**7.50\%**|
> |rl1323|1323|93.88\%|94.63\%|**95.01\%**|94.86\%|16.44\%|**9.34\%**|17.81\%|19.77\%|
> |u1817|1817|88.83\%|89.71\%|89.49\%|**90.59\%**|17.60\%|**9.54\%**|12.22\%|18.93\%|
> |d2103|2103|94.53\%|95.10\%|95.63\%|**95.67\%**|20.50\%|**8.09\%**|17.78\%|18.75\%|
> |pr2392|2392|88.00\%|89.01\%|89.05\%|**90.34\%**|10.72\%|8.08\%|8.26\%|**7.50\%**|
> |fnl4461|4461|86.77\%|**87.34\%**|87.13\%|87.20\%|6.85\%|**4.22\%**|6.00\%|5.75\%|
> |rl11849|11849|91.58\%|91.69\%|**92.06\%**|91.97\%|8.13\%|**6.96\%**|8.00\%|7.71\%|
> |usa13509|13509|87.99\%|88.44\%|87.97\%|**88.74\%**|8.97\%|**6.99\%**|9.80\%|8.47\%|
> |pla33810|33810|89.51\%|**90.10\%**|89.63\%|89.83\%|7.57\%|6.37\%|**5.82\%**|6.44\%|
> |Average||90.56\%|91.16\%|91.14\%|**91.47\%**|11.18\%|**7.18\%**|9.83\%|10.61\%|
>
> > **References**
>
> [1] Neural combinatorial optimization with heavy decoder: Toward large scale generalization, NeurIPS 2023.
>
> [2] Learning to handle complex constraints for vehicle routing problems, NeurIPS 2024.

---

> > ### Comment · Reviewer_LmUY · 2025-11-27
> >
> > Thank you for your response. I am satisfied with the additional experiments you have provided. While I believe that integrating hierarchical RL remains a worthwhile direction for exploration, it is somewhat beyond the current scope of this work and can reasonably be considered as future work. Overall, the core idea of this paper is compelling and has the potential to address some fundamental challenges in NCO. Therefore, I am raising my score to a clear accept.

---

> > > ### Author Response · Authors · 2025-11-27
> > >
> > > We sincerely thank you for the positive evaluation and for raising the score to a clear accept. We are particularly encouraged by your comments on the potential of our method to address fundamental challenges in NCO. We also fully agree with your insight regarding hierarchical RL. It is indeed a promising direction, and we will definitely consider it as a priority for our future work.

---

### Official Review · Reviewer_nzMG · 2025-10-31

**Soundness:** 2
**Presentation:** 3
**Contribution:** 2
**Rating:** 4
**Confidence:** 4

**Summary:**

This paper proposes to reduce the search space of constructive neural routing solvers with an additional learning to reduce (L2R) module, which consists of an embedding layer and a self-attention layer. Experiments demonstrate impressive performance when generalizing to large-scale instances.

**Strengths:**

1.	The idea of learning to reduce the search space is clear and reasonable. The writing of this paper is clear and easy to follow.
2.	The experimental results on generalizing to large-scale instances are significant and impressive.

**Weaknesses:**

1.	The methodology that utilizing an embedding layer and a self-attention layer for the purpose of learning to reduce search space with thousands of nodes seems brute. Meanwhile, such architectures are prevailing in constructive neural routing solvers when identifying the importance scores of nodes, which, to some extent, limits the novelty and potential contribution of this work.

**Questions:**

1.	As described in the paper, the static reduction phase only reduces 10% of candidate nodes, which is a very small part. How do you determine the ratio? Is there any experimental evidence to show the necessarily of static reduction?
2.	What about the performance on small-scale instances like 100-node instances? Does the generalization ability come at the cost of degradation on small-scale instances?

---

> ### Author Response · Authors · 2025-11-24
> **Response to Reviewer nzMG [1/3]**
>
> Thank you very much for your time and effort in reviewing our work. Below, we address each concern point by point. All revisions and additions made to the manuscript have been marked in blue to aid in your review.
>
> > **W1. The novelty and potential contribution of this work.**
>
> Thank you for the insightful comment regarding the model architecture. We fully agree with you that utilizing embedding layers and an attention layer to identify node importance is indeed prevailing in constructive neural routing solvers, and our approach builds upon this established concept. However, we would like to offer further clarification regarding the specific design choices, novelty, and contributions of this work:
>
> - **Contribution of L2R:** Standard attention mechanisms typically require interactions between the current state and every feasible node at each construction step. While effective at small scales, this approach becomes computationally prohibitive for large-scale problems (i.e., $N \ge 10K$). To address this, we employ a lightweight reduction model. As you rightly noted, identifying promising candidates within a large search space using only a lightweight network remains a challenging task. Therefore, we designed a distance-assisted reduction model, which can adaptively combine distance-based priors with context-dependent features learned by the network to generate potential scores for unvisited nodes.
>
> While the scoring mechanism shares conceptual similarities with existing works, a key contribution of L2R is its ability to perceive and utilize node attributes beyond simple spatial proximity (e.g., demands and time windows), enabling it to effectively apply to more complex routing problems, such as CVRP and supplementary CVRPTW, where the optimal solution depends not only on geometric distance but also heavily on customer demands and time window information. Extensive experiments demonstrate that L2R significantly outperforms methods that rely solely on distance rules (such as INViT[1]).
>
> Moreover, L2R can also demonstrate unprecedented generalization capabilities. To the best of our knowledge, it is **the first constructive neural solver to surpass HGS on large-scale CVRP instances ($N \ge 50K$) when trained solely on 100-node uniform instances without any post-processing operations** (as detailed in our response to Q2).
>
> - **Verification on Complex Constraints:** To further validate the applicability of L2R on problems with complex non-spatial constraints, we have conducted additional experiments on the Capacitated Vehicle Routing Problem with Time Windows (CVRPTW). Unlike TSP or CVRP, CVRPTW involves complex time window constraints alongside spatial and demand constraints. This makes it particularly challenging for the reduction methods, since they must learn the latent relationship between multiple non-spatial attributes (demand and time windows) and the suitability of candidate nodes.
>
> According to the results in **Table 1**, our proposed L2R consistently achieves the best solution quality on CVRPTW instances, complemented by remarkably fast inference times, across various problem scales. The advantages of L2R over existing neural solvers are significant. This confirms that our learning-based reduction approach can successfully capture complex non-spatial constraints that simple distance-based heuristics cannot, proving the broad applicability of the L2R framework beyond simple routing tasks.
>
> We hope these discussions can highlight both the novelty, contribution, and potential practical impact of our approach, and we will add them to the revised paper.
>
> Table 1. Comparison of CVRPTW instances. Note that "Time" indicates the total running time. We limit the test scale to 10,000 nodes, as existing NCO methods consistently run out of Memory (OOM) on instances where $N \ge 5,000$.
> |Method|CVRPTW1K||CVRPTW5K||CVRPTW10K||
> |:---:|:---:|:---:|:---:|:---:|:---:|:---:|
> ||Obj.(Gap)|Time|Obj.(Gap)|Time|Obj.(Gap)|Time|
> |HGS-PyVRP|159.35(0.00\%)|20m|787.25(0.00\%)|3.3h|1375.84(0.00\%)|5h|
> |MTPOMO aug*8|219.22(37.57\%)|1.7m|OOM||OOM||
> |MVMoE aug*8|233.53(46.55\%)|1.7m|OOM||OOM||
> |ReLD aug*8|183.31(15.036\%)|1.6m|OOM||OOM||
> |L2R greedy|**174.31(9.39\%)**|**8s**|**849.83(7.95\%)**|**40s**|**1497.30(8.83\%)**|**1.4m**|

---

> ### Author Response · Authors · 2025-11-24
> **Response to Reviewer nzMG [2/3]**
>
> > **Q1. How do you determine the ratio of the static reduction phase? Is there any experimental evidence to show the necessity of static reduction?**
>
> Thank you for raising this question regarding the static reduction ratio. Static search space reduction is a practical technique that consistently improves computational efficiency, as validated in related studies. Our approach builds upon prior work, such as MLPR[4,5], which employs model predictions to eliminate unpromising edges in large-scale COPs. We also draw inspiration from recent advancements, such as DIMES[6], DIFUSCO[7], and T2T[8], which utilise graph sparsification to enhance solving efficiency. In our L2R framework, the static search space reduction serves as a conservative, one-time preprocessing step that filters out clearly non-optimal edges (typically the longest edges) without compromising the search space for subsequent learning-based dynamic reduction.
>
> - **Determination of the Ratio (Empirical Rationale)** To determine an appropriate threshold $\gamma$, we have conducted a statistical analysis of the optimality ratio, defined as the proportion of construction steps where the (near-)optimal next-visit node resides within the $k$ -nearest candidates. We generate TSP instances at three scales ($N=100, N=1,000, N=10,000$) across four distinct distributions (uniform, clustered, explosion, implosion). Specifically, the dataset comprises 10,000 instances per setting for $N=100$, and 1,000 instances per setting for the larger scales ($N=1,000$ and $10,000$). We use LKH3 to generate near-optimal solutions as ground truth.
>
> As illustrated in **Appendix G** of the revised paper, the optimality ratio increases monotonically with the neighbour size $k$, and each distribution exhibits a critical threshold $k^{+}$ such that all optimal nodes are preserved when $k \leq k^{+}$. We observe that the edges belonging to the farthest 10\% (long-distance connections) almost never appear in the optimal solutions across all tested distributions. Therefore, we set $\gamma$=10% (retaining the top 90\% of edges) as a safe boundary. This ensures that it preserves the potential for global optimality while reducing memory and computational overhead for the subsequent steps.
>
> - **Necessity of Static Reduction (Experimental Evidence):** We conduct an ablation study to verify the necessity of this static reduction phase, as detailed in **Table 2**. Since strong baselines like BQ[9] and INViT[1] already incorporate distance-based dynamic reduction, we selected LEHD[10] as a representative baseline for comparison.
>
> According to the results in **Table 2**, on the LEHD baseline, enabling static reduction reduces the inference time on TSP10K from 3.1 hours to 2.3 hours, which is a significant speedup while maintaining solution quality. On the other hand, when removing SR from our L2R, performance remains unchanged, but the solving time increases, especially for TSP100K instances. This result confirms that the static reduction serves as an effective, model-agnostic preprocessing step that improves efficiency without sacrificing solution quality. It also demonstrates that the superior performance of our L2R framework stems from its core architecture, not just the preprocessing step.
>
> While the 10\% reduction ratio may appear small, pruning these edges reduces the solving time during the construction process, which is particularly beneficial when scaling to very large instances (e.g., 1 million nodes).
>
> Table 2. Comparison with and without static reduction for LEHD and our L2R. Note that "SR" refers to the static reduction technique we employed, which removes the farthest 10% of connections based on Euclidean distance.
> |Method|TSP1K||TSP10K||TSP100K||
> |:---:|:---:|:---:|:---:|:---:|:---:|:---:|
> ||Gap|Time|Gap|Time|Gap|Time|
> |LEHD-Greedy w/o SR|3.11\%|98s|27.24\%|3.1h|OOM||
> |LEHD-Greedy w/ SR|3.38\%|**73s**|26.76\%|**2.3h**|OOM||
> |L2R-Greedy w/o SR|4.49\%|7s|4.82\%|68s|4.79\%|29.6m|
> |L2R-Greedy w/ SR|4.49\%|**6s**|4.82\%|**64s**|4.79\%|**29.1m**|

---

> ### Author Response · Authors · 2025-11-24
> **Response to Reviewer nzMG [3/3]**
>
> > **Q2. What about the performance on small-scale instances like 100-node instances? Does the generalization ability come at the cost of degradation on small-scale instances?**
>
> Thank you for raising this important concern. We agree that assessing performance on small-scale instances is crucial for a comprehensive evaluation, especially in terms of the training distribution (N = 100). We apologize for not including these results in the initial submission. To address this, we have conducted new experiments on both TSP and CVRP instances with sizes $N = 100$ and $N = 500$. We directly compare L2R with current representative NCO solvers. For small-scale CVRP, the capacity settings follow the setup in LEHD[1].
>
> As shown in **Table 3**, our L2R achieves **highly competitive performance on these small-scale instances** compared to mainstream NCO solvers, such as BQ, LEHD, and ELG. Furthermore, the results also highlight L2R's superior scalability. These comprehensive experiments confirm that L2R's promising large-scale generalization ability does not come at the cost of performance degradation on small-scale instances.
>
> We would like to emphasize that the core contribution of this work lies in pushing the frontier of NCO in terms of generalization and scalability. Comprehensive experiments demonstrate that our proposed L2R method, trained only on 100-node instances, can generalize well to handle instances with up to 10 million nodes. As instance size increases, most NCO solvers (BQ, LEHD, POMO, ELG) fail due to Out-of-Memory (OOM) errors, while L2R's performance advantage becomes more significant. It is particularly noteworthy that on large-scale CVRP (e.g., CVRP100K), **our L2R finds a higher-quality solution than the powerful heuristic HGS (which has been refined for decades)**, and does so with a dramatic speedup (**31 minutes vs. 4 days on CVRP100K instances**). This represents a significant milestone in NCO: **the first constructive neural solver to achieve superior performance over HGS on large-scale CVRP instances ($N \ge 50K$) that only trained on 100-node uniform instances without any post-processing operations**.
>
> The new experiments you suggested helpfully validate that L2R is also a competitive solver on small-scale instances. We sincerely appreciate your valuable suggestion and will include these new results and corresponding discussions in the revised paper.
>
> Table 3. Experimental results on varying-scale TSP and CVRP instances with uniform distribution. All models are trained on instances of size N = 100. Note that "Time" indicates the total running time, and all methods employ greedy decoding, without utilizing instance augmentation or additional search strategies (e.g., RRC or PRC).
> |||TSP100(128 ins.)||TSP500(128 ins.)||TSP50K(16 ins.)||TSP100K(16 ins.)||
> |---|---|:---:|:---:|:---:|:---:|:---:|:---:|:---:|:---:|
> |Method||Obj.|Time|Obj.|Time|Obj.|Time|Obj.|Time|
> |LKH3||7.73|1m|16.52|32m|159.93|10h|225.99|25h|
> |BQ||7.76|2s|**16.72**|46s|OOM||OOM||
> |LEHD||7.78|1s|16.78|16s|OOM||OOM||
> |POMO||**7.76**|1s|20.50|9s|OOM||OOM||
> |ELG||7.78|1s|17.82|17s|OOM||OOM||
> |INViT||8.10|2s|17.55|12s|171.42|1.3h|242.26|5h|
> |L2R||7.84|2s|17.03|3s|**167.70**|9.5m|**236.81**|29.4m|
> |||**CVRP100(128 ins.)**||**CVRP500(128 ins.)**||**CVRP50K(16 ins.)**||**CVRP100K(16 ins.)**||
> |Method||Obj.|Time|Obj.|Time|Obj.|Time|Obj.|Time|
> |HGS||15.52|3.5m|36.56|4h|1081.00|1.3d|2087.51|4d|
> |BQ||16.03|2s|38.43|47s|OOM||OOM||
> |LEHD||16.16|1s|**38.41**|17s|OOM||OOM||
> |POMO||**15.80**|1s|48.22|10s|OOM||OOM||
> |ELG||15.94|1s|39.67|23s|OOM||OOM||
> |INViT||16.77|2s|43.44|12s|1331.10|1.5h|2683.40|5.8h|
> |L2R||16.12|2s|39.31|4s|**1074.98**|10m|**2055.14**|31m|
>
> > **References**
>
> [1] Invit: A generalizable routing problem solver with invariant nested view transformer, ICML 2024.
>
> [2] MVMoE: Multi-Task Vehicle Routing Solver with Mixture-of-Experts. ICML 2024.
>
> [3] Generalize Learned Heuristics to Solve Large-scale Vehicle Routing Problems in Real-time, ICLR 2023.
>
> [4] Using statistical measures and machine learning for graph reduction to solve maximum weight clique problems. IEEE TPAMI 2019.
>
> [5] Generalization of machine learning for problem reduction: a case study on travelling salesman problems. Or Spectrum, 2021, 43(3): 607-633.
>
> [6] Qiu R, Sun Z, Yang Y. Dimes: A differentiable meta solver for combinatorial optimization problems.NeurIPS 2022.
>
> [7] Sun Z, Yang Y. Difusco: Graph-based diffusion solvers for combinatorial optimization. NeurIPS 2023.
>
> [8] T2t: From distribution learning in training to gradient search in testing for combinatorial optimization. NeurIPS 2023.
>
> [9] Bq-nco: Bisimulation quotienting for efficient neural combinatorial optimization, NeurIPS 2023.
>
> [10] Neural combinatorial optimization with heavy decoder: Toward large scale generalization, NeurIPS 2023.

---

### Official Review · Reviewer_8Tiw · 2025-10-31

**Soundness:** 3
**Presentation:** 2
**Contribution:** 2
**Rating:** 4
**Confidence:** 4

**Summary:**

The paper applies RL to the search space reduction problem in NCO. The proposed algorithm is simple to implement and exhibits good generalization capability.

**Strengths:**

- The proposed algorithm is simple to implement and generalizes well to larger instances and different distributions.

**Weaknesses:**

- The writing can be improved. The related work section, a large part of section 2, and many experiments were moved to the appendix. The "Impact on Constructive NCOs" paragraph in section 2 provides no useful information, as the main content has been moved to the appendix. Many experiments only have their names mentioned in the main text. The author should at least describe the results and implications of the experiments in the main text. I suggest the authors revise their manuscript to include the necessary information in the main text.
- There is little algorithmic novelty. The paper simply applies RL to the search space reduction problem. The network architecture was reused or derived from prior works with little justification. For example, following previous works, the context embedding is just the sum of the first and last node embeddings. What is the justification for this? Can a better context embedding improve the performance?
- The proposed algorithm addresses a small and specific problem in NCO. It is a nice engineering trick that unlikely applies to other problems.

**Questions:**

- Why use RL instead of other learning paradigms? A clearer motivation for using RL and a comparison between RL and SL (and possibly SSL) would improve the quality of this paper.

---

> ### Author Response · Authors · 2025-11-24
> **Response to Reviewer 8Tiw [1/3]**
>
> Thank you very much for your time and effort in reviewing our work. Below, we address each concern point by point. All revisions and additions made to the manuscript have been marked in blue to aid in your review.
>
> > **W1. The writing can be improved.**
>
> Thank you for the constructive feedback regarding the organization and presentation of our manuscript. We fully agree that essential analyses and experimental implications should be presented within the main text to ensure a coherent and self-contained narrative. We will carefully revise the manuscript to address this issue. Specifically:
>
> - **Reintegrate Related Work:** We will move the core content of the Related Work section back to the main text. This will ensure that the background of existing Neural Combinatorial Optimization (NCO) methods and the distinction between Static and Dynamic Search Space Reduction are clearly established before introducing our methodology.
>
> - **Revise Section 2:** We will rewrite the "Impact on Constructive NCOs" paragraph to explicitly summarize the key findings regarding how distance-based reduction affects different NCO architectures (e.g., Heavy Encoder vs. Heavy Decoder) within the main text, rather than solely referring to the Appendix.
>
> Thank you again for helping us refine these important details. We believe these revisions will significantly improve the readability and completeness of our paper.
>
> > **W2.1. Algorithmic novelty**
>
> Thank you for your insightful comments regarding the algorithmic novelty of our work. We would like to clarify our contributions and address the specific points raised.
>
> - **Algorithmic Novelty**: We agree with you that the network architecture in L2R builds upon existing designs, which enables us to focus our contribution on the novel L2R mechanism. However, we respectfully disagree with the assessment regarding novelty. As noted in the related work section of our manuscript, search space reduction is an established concept that appears in many existing works (e.g., INViT[1]). Nevertheless, we would like to emphasize that the key contribution of L2R  lies in introducing **the first learning-based framework for dynamic search space reduction**, which goes beyond the existing purely heuristic-based approaches.
>
> Existing dynamic search space reduction methods are implemented fully based on distance, which can severely harm the solution quality if an incorrect reduction is made. In contrast, L2R learns reduction policies dynamically, which enables effective reduction and leads to promising performance on complex routing problems such as CVRP and CVRPTW (see the response for W3), where optimal solutions depend not only on geometric distance but also heavily on customer demands and time window information. As a result, L2R can significantly outperform methods that rely solely on distance-based rules, such as INViT [1].
>
> Moreover, L2R can also demonstrate unprecedented generalization capabilities. As shown in **Table 1**, to the best of our knowledge, it is **the first constructive neural solver to surpass HGS on large-scale CVRP instances ($N \ge 50K$) when trained solely on 100-node uniform instances without any post-processing operations**. To ensure a strong baseline, we evaluate HGS on a high-performance server equipped with an Intel(R) Xeon(R) Platinum 8352V CPU at 2.10 GHz and 80GB of RAM. Due to the excessive memory consumption of HGS on these large-scale instances, parallel processing is infeasible on CVRP100K instances. We allocate sufficient time budgets based on the instance size to execute HGS (4 hours for CVRP50K instances and 6 hours for CVRP100K instances).
>
> Table 1. Experimental results on large-scale CVRP instances with uniform distribution. All models are trained on instances of size N = 100. Note that "Time" indicates the total running time, and HGS is running under the given 80 GB of memory.
> |||CVRP50K(16 ins.)||CVRP100K(16 ins.)||CVRP1M(16 ins.)||
> |---|---|:---:|:---:|:---:|:---:|:---:|:---:|
> |Method||Obj.(Gap)|Time|Obj.(Gap)|Time|Obj.(Gap)|Time|
> |HGS||1081.00(0.00\%)|1.3d|2087.51(0.00\%)|4d|OOM||
> |BQ greedy||OOM||OOM||OOM||
> |LEHD greedy||OOM||OOM||OOM||
> |POMO aug*8||OOM||OOM||OOM||
> |ELG aug*8||OOM||OOM||OOM||
> |INViT greedy||1331.10(23.1\%)|1.5h|2683.40(28.55\%)|5.8h|N/A||
> |L2R greedy||**1074.98(-0.56\%)**|10m|**2055.14(-1.55\%)**|31m|**17231.85($-$)**|22h|

---

> ### Author Response · Authors · 2025-11-24
> **Response to Reviewer 8Tiw [2/3]**
>
> > **W2.2. Context embedding**
>
> The context embedding used in our model is mathematically justified by the structure of the routing problem, which is naturally modeled as a **Markov Decision Process**. For a routing problem, the transition probability depends only on the current state and the remaining unvisited nodes, rather than the full set of visited nodes along the route. Given the Hamiltonian cycle constraint, the current state is sufficiently defined by the current node ($\pi_{t-1}$) and the origin/target node ($\pi_{1}$). This formulation is widely adopted in current representative neural routing solvers (e.g., INViT[1] and BQ-NCO[2]). A general context embedding is often defined as $h_{C_{D}}^{t}=W_{c}[ h_{\pi_{1}}^{D}, h_{\pi_{t-1}}^{D}]$, where $W_{c} \in \mathbb{R}^{d \times 2d}$ is a learnable matrix.
>
> The approach used in our paper is **mathematically equivalent** to the above standard practice. For computational efficiency, we decompose $W_{c}$ into $W_{\text{first}} \in \mathbb{R}^{d \times d}$ and $W_{\text{last}} \in \mathbb{R}^{d \times d}$, resulting in our formulation $h_{C_{D}}^{t} = W_{\text{first}} h_{\pi_{1}}^{D} + W_{\text{last}} h_{\pi_{t-1}}^{D}$. Thank you again for this valuable comment. We will add the justification to the revised paper.
>
> We agree that more advanced context embeddings (e.g., incorporating positional encodings as suggested in [3]) could potentially further boost performance. We appreciate this valuable suggestion and will explore designing better context embeddings in our future work.
>
> > **W3. The proposed algorithm addresses a small and specific problem in NCO. It is a nice engineering trick that unlikely applies to other problems.**
>
> Thank you for raising the important point regarding the applicability and scope of our method. We fully agree that evaluating our method on more complex routing problems is essential to demonstrate its generalization and practical value.
>
> To demonstrate that L2R is not merely a specific engineering trick but a robust framework for a broad class of routing problems, we extended our experiments to the Capacitated Vehicle Routing Problem with Time Windows (CVRPTW). Unlike TSP or CVRP, CVRPTW involves complex time window constraints in addition to spatial and demand constraints. This makes it particularly challenging for the reduction methods, since they must learn the latent relationship between multiple non-spatial attributes (demand and time windows) and the suitability of candidate nodes.
>
> - **Experimental Setup**: We have generated CVRPTW instances following the MVMoE setting[4] and limited the test scale to 10,000 nodes since existing NCO methods consistently run out of Memory (OOM) on instances where $N \ge 5,000$. Each dataset contains 16 instances. For all instances, we adhere to the capacity constraints specified in [5].
>
> - **Results:** According to the results in **Table 2**, our proposed L2R consistently achieves the best solution quality, complemented by remarkably fast inference times, across various problem scales. The advantages of L2R over existing neural solvers are significant. These results confirm that our learning-based reduction successfully captures complex, non-spatial constraints that simple distance-based heuristics cannot, proving the broad applicability of the L2R framework beyond simple routing tasks.
>
> - **Potential for Broader Combinatorial Optimization:** Beyond routing problems, we argue that the principle of dynamic search space reduction is applicable to a wide range of combinatorial optimization problems, where the search space is often defined by decision actions rather than graph nodes. For example, in the Flexible Job-Shop Scheduling Problem, the action space typically consists of all valid job-machine pairs. Analogous to L2R, the reduction strategy employed in [6] effectively limits the action space of dispatching rules by selecting only the top-$k$ operations based on specific criteria (e.g., earliest start or earliest completion times). This parallel provides compelling evidence that L2R can serve as a fundamental and adaptable methodological framework, rather than a narrow engineering trick restricted to specific tasks. We believe our framework holds significant potential for such domains, and extending L2R to non-routing problems is a key direction for our future research.
>
> We will include these new CVRPTW results in the revised manuscript to better highlight the generalization of our proposed method. Thank you very much again for this suggestion, which has substantially strengthened our paper's contribution.

---

> ### Author Response · Authors · 2025-11-24
> **Response to Reviewer 8Tiw [3/3]**
>
> Table 2. Comparison on CVRPTW instances. Note that "Time" indicates the total running time. We limit the test scale to 10,000 nodes, as existing NCO methods consistently run out of Memory (OOM) on instances where $N \ge 5,000$.
> |Method|CVRPTW1K||CVRPTW5K||CVRPTW10K||
> |:---:|:---:|:---:|:---:|:---:|:---:|:---:|
> ||Obj.(Gap)|Time|Obj.(Gap)|Time|Obj.(Gap)|Time|
> |HGS-PyVRP|159.35(0.00\%)|20m|787.25(0.00\%)|3.3h|1375.84(0.00\%)|5h|
> |MTPOMO aug*8|219.22(37.57\%)|1.7m|OOM||OOM||
> |MVMoE aug*8|233.53(46.55\%)|1.7m|OOM||OOM||
> |ReLD aug*8|183.31(15.036\%)|1.6m|OOM||OOM||
> |L2R greedy|**174.31(9.39\%)**|**8s**|**849.83(7.95\%)**|**40s**|**1497.30(8.83\%)**|**1.4m**|
>
> > **Q1. Why use RL instead of other learning paradigms?**
>
> Thank you for this insightful question. A clear comparison of learning paradigms would indeed help strengthen the motivation of our work. We chose Reinforcement Learning (RL) based on the specific challenges of broad routing problems and the core philosophy of Neural Combinatorial Optimization (NCO).
>
> - **Comparison with Supervised Learning (SL):** SL relies heavily on high-quality ground truth labels. However, for complex routing variants, obtaining a large amount of near-optimal solutions is computationally prohibitive. This often requires (computationally) expensive exact solvers or extensive runs of leading heuristics, which severely limits the practical applicability of SL-based methods.
>
> - **Comparison with Self-Supervised Learning (SSL):** While SSL can reduce the need for extensive (nearly) optimal solutions, effective SSL in the NCO community typically requires designing advanced search strategies to generate high-quality pseudo-labels for self-training. Developing these strategies demands substantial domain-specific expert knowledge and careful fine-tuning.
>
> - **Motivation for RL:** Our primary objective aligns with the fundamental goal of NCO: to minimize reliance on domain expertise. RL enables the model to learn rules by repeatedly interacting with the environment, without requiring any labeled data or manually well-designed search strategies. This enables L2R to be effectively applied across multiple routing problems, such as TSP, CVRP, and CVRPTW, with outstanding large-scale generalization performance, which significantly facilitates the practical application of the model.
>
> We will incorporate this discussion into the revised paper to provide a clearer explanation of our design choice.
>
> > **References**
>
> [1] Invit: A generalizable routing problem solver with invariant nested view transformer, ICML 2024.
>
> [2] Bq-nco: Bisimulation quotienting for efficient neural combinatorial optimization, NeurIPS 2023.
>
> [3] The Transformer Network for the Traveling Salesman Problem. arXiv preprint arXiv:2103.03012, 2021.
>
> [4] MVMoE: Multi-Task Vehicle Routing Solver with Mixture-of-Experts. ICML 2024.
>
> [5] Generalize Learned Heuristics to Solve Large-scale Vehicle Routing Problems in Real-time, ICLR 2023.
>
> [6] Diverse policy generation for the flexible job-shop scheduling problem via deep reinforcement learning with a novel graph representation. Engineering Applications of Artificial Intelligence, 2025, 139: 109488.

---

### Meta-Review · Area_Chair_Dj4b · 2026-01-06

**Summary:**

Reviewers' concerns are mainly around the following points:

W1. Writing needs improvement

W2. Limited novelty, and the proposed method is quite naive

W3. The proposed trick unlikely applies to other problems

W4. The reduction method seems brute, and only a small portion of search space is reduced

W5. Experiments for N < 1000 are missing

W6. While claim distance-based pruning is unsuitable, the paper actually relies on the distance-based pruning

W7. Missing important baselines

W8. This approach is limited to (Euclidean) routing problems only

W9. The “Generalizable neural routing solver” is an overstatement

**Reviewer Concerns:**

Among the above concerns, I feel that the limited technical novelty (W2), the limited scope (W8 and W9) are still outstanding. I agree with reviewers that the architecture largely relies on existing works, and authors did not demonstrate the ability of applying to non-Euclidean routing problems and other types of COPs.

**Reviewer Scores:**

According to authors' summary, two reviewers improved their ratings. However, I found it hard for reviewers to further improve scores considering the insufficient authors' rebuttal mentioned above.

---

### Decision · Program_Chairs · 2026-01-26

Reject